# Parallel wavelength-division-multiplexed signal transmission and dispersion compensation enabled by soliton microcombs and microrings

Yuanbin Liu [1,4], Hongyi Zhang[1,4], Jiacheng Liu[1,4], Liangjun Lu[1,2] ✉, Jiangbing Du [1] ✉, Yu Li[1,2], Zuyuan He [1], Jianping Chen[1,2], Linjie Zhou [1,2] & Andrew W. Poon [3]

The proliferation of computation-intensive technologies has led to a significant rise in the number of datacenters, posing challenges for high-speed and power-efficient datacenter interconnects (DCIs). Although inter-DCIs based on intensity modulation and direct detection (IM-DD) along with wavelength-division multiplexing technologies exhibit power-efficient and large-capacity properties, the requirement of multiple laser sources leads to high costs and limited scalability, and the chromatic dispersion (CD) restricts the transmission length of optical signals. Here we propose a scalable on-chip parallel IM-DD data transmission system enabled by a single-soliton Kerr microcomb and a reconfigurable microring resonator-based CD compensator. We experimentally demonstrate an aggregate line rate of 1.68 Tbit/s over a 20-km-long SMF. The extrapolated energy consumption for CD compensation of 40-km-SMFs is ~0.3 pJ/bit, which is calculated as being around 6 times less than that of the commercial 400G-ZR coherent transceivers. Our approach holds significant promise for achieving data rates exceeding 10 terabits.

According to the International Data Corporation (IDC), the exponential growth of digital data generation is projected to reach 175 zettabytes by 2025[1] due to the surgent growth of computation-intensive technologies such as artificial intelligence, the Internet of Things (IoT), and autonomous vehicles. Although the total data capacity grows 80 times from 640 Gbit/s in 2010 to 51.2 Tbit/s in 2022 in datacenter interconnects (DCIs), the total power consumption also grows dramatically with a 22-fold increase[2]. As a result, there is a critical need for energy-efficient and cost-effective DCIs that can accommodate a substantial capacity. There are mainly two technical approaches to DCIs, namely intensity modulation and direct detection (IM-DD) and

coherent schemes. Until recently, the IM-DD technology has been the clear choice for intra-DCIs (<2 km), owing to its low energy consumption and easy implementation[3–5]. However, the scaling of data rate beyond 1.6 Tbit/s is challenging even for the O-band intra-datacenter scenarios as the IM-DD system is less tolerant to optical impairments such as the fiber chromatic dispersion (CD)[3]. The system suffers from an even larger CD in the C-band, which also limits the scaling of the data rate of IM-DD systems working in the C-band. For the coherent technology, it has been widely used in long-haul and metro optical networks owing to its high receiver power sensitivity and high spectral efficiency[3]. Due to the considerable reduction of cost and

[1]State Key Laboratory of Advanced Optical Communication Systems and Networks, Department of Electronic Engineering, Shanghai Jiao Tong University, Shanghai 200240, China. [2]SJTU-Pinghu Institute of Intelligent Optoelectronics, Pinghu 314200, China. [3]Photonic Device Laboratory, Department of Electronic and Computer Engineering, The Hong Kong University of Science and Technology, Clear Water Bay, Hong Kong. [4]These authors contributed equally: Yuanbin Liu, Hongyi Zhang, Jiacheng Liu. ✉e-mail: luliangjun@sjtu.edu.cn; dujiangbing@sjtu.edu.cn

power consumption, the coherent technology has shown competitive performances for transmission within 100 km single-mode fiber (SMF)[3]. Recently, data transmission using coherent technology has already reached the fiber length of less than 10 km[6–8], some of which have even been successfully demonstrated for intra-datacenter applications[7,8].

As the typical distance between two data centers is within 80 km and the majority (~83%) of the DCIs are within 40 km[9], we mainly focus on the short-reach inter-DCIs with a fiber length of up to 40 km and the utilization of IM-DD scheme for this scenario due to the inherent advantages IM-DD offers, including cost-effectiveness, low power consumption, and a compact physical footprint. Due to the lower data rate of the IM-DD system for a single wavelength channel than the coherent scheme, wavelength-division multiplexing (WDM) technology is commonly employed to economically enhance the data capacity. However, multiple laser sources are required to provide multiple operation wavelengths in the WDM systems, which leads to a higher cost. Moreover, in IM-DD systems operating within the C-band, the transmission distance is primarily limited by CD. The CD-induced frequency-selective fading (FSF) significantly impacts transmission performance[10,11].

To address the CD effect, various electronic and optical approaches have been employed, which can be generally categorized into three types[12], the pre-distortion compensation[13–19], the inline compensation[20,21], and the post compensation[22–27], depending on the position of CD compensator (CDC). Digital signal processing (DSP)-based electronic CD compensation is usually applied at the receiver side as post compensation, which has been used in IM-DD[22] and coherent[23,24] transmission systems. Particularly, it has been commonly used in the commercially available 400 G ZR coherent transceiver modules[25–27]. However, the DSP-based coherent transceiver module suffers from a relatively high energy consumption of about 0.5 W/100 G (5 pJ/bit) for 120-km-SMF CD compensation[25]. Dispersion-compensating fibers (DCFs) are also commonly used to compensate for the CD of SMF. Nevertheless, using DCFs is burdened with a relatively high associated cost and low reconfigurability for flexible CD compensation of different fiber lengths. The tunability of CDCs is important, as the amount of CD may vary in time due to some possible impairments, such as the optical power variations and environmental-variation-induced transmission condition changes[28]. Besides, one reconfigurable CDC capable of compensating the CD of different lengths of SMFs up to 40 km can meet the demand of CD compensation for ~83% of DCI links, which can greatly reduce the overall costs. In addition, reconfigurable optical networks that may be deployed in next-generation DCI networks to improve the system performance and efficiency[29] also need tunable CDCs.

In recent years, the adoption of silicon photonics facilitated the integration of various silicon-based photonic devices, including chirped Bragg gratings[30–34], Mach-Zehnder interferometers (MZIs)[35,36], and microring resonators (MRRs)[37–43], to effectively mitigate the impact of CD in SMFs. These integrated devices offer several advantages, including a compact form factor, cost efficiency, low power consumption, and high reconfigurability. While imbalanced MZIs and MRRs demonstrate desirable characteristics for parallel signal processing due to their periodic spectral properties, their potential in WDM systems is currently limited. As a result, existing studies have reported only a modest maximum data transmission line rate of 240 Gbit/s[41].

Moreover, chip-scaled soliton microcombs, known for their wide spectral range, low noise, and high repetition rate, have emerged as a promising light source for WDM applications[44–53]. Recent advancements have demonstrated massively parallel data transmission using various types of microcomb sources, including bright single-solitons[54], soliton crystals with defects[55], and dark solitons[56]. For example, the utilization of two interleaved single-soliton microcombs in a silicon

nitride (Si$_3$N$_4$) microresonator enabled coherent data transmission with 179 carriers in the C- and L-bands, achieving a total line rate of 55 Tbit/s[54]. Similarly, the soliton crystal with defects, generated in a doped silica glass microresonator, facilitated a line rate of 44.2 Tbit/s and a spectral efficiency of 10.2 bit/s/Hz[55]. However, it is critical to note that the majority of WDM transmission systems utilizing soliton microcombs rely on expensive coherent schemes, necessitating costly transmitters and receivers[54–59]. Recently, massively parallel optical interconnects based on a Si$_3$N$_4$ Kerr dark soliton comb source and a silicon microdisk modulator array with a data rate of 512 Gbit/s for IM-DD transmission have been demonstrated[60]. In addition, a parallel optical data link driven by an AlGaAs Kerr microcomb has achieved a data rate of 2 Tbit/s in the IM-DD system[61]. However, we note that these transmission techniques are limited in terms of transmission lengths and are primarily suitable for intra-DCI applications.

In this paper, we present an approach to short-reach inter-DCIs based on an IM-DD optical communications architecture. Our proposed system leverages a single-soliton microcomb source generated by a Si$_3$N$_4$ MRR as a WDM light source. Furthermore, we introduce a fully reconfigurable optical CDC based on cascaded silicon MRRs for parallel CD compensation. This WDM-assisted IM-DD scheme offers numerous advantages, including high scalability, compactness, energy efficiency, and cost-effectiveness. To evaluate the performance of our proposed system, we conducted experiments demonstrating parallel signal transmission using up to 15 wavelength channels within the C-band. By employing pulse-amplitude four-level (PAM4) modulation signals at a rate of 80 Gbit/s, we achieved an aggregate data line rate of 1.2 Tbit/s. Additionally, by employing discrete multi-tone (DMT) modulation signals at a rate of 112 Gbit/s, we achieved an aggregate data line rate of 1.68 Tbit/s. These transmission rates were achieved over a 20-km-length SMF. Importantly, we assessed the power consumption for the CD compensation to be as low as ~0.3 pJ/bit for 40-km SMFs. This remarkably low power consumption is estimated to be around six times less than that of the commercially available 400 G ZR coherent transceiver modules. By aligning the free spectral range (FSR) of the Kerr MRR and the CDC and further increasing the number of wavelength channels, we expect the transmission capacity to exceed 10 Tbit/s. As a result, the estimated power consumption can even be reduced to ~0.1 pJ/bit for the 40-km SMF transmission with an optimized coupling loss and insertion loss of our CDC. This capability holds significant promise for meeting the demands of future hyperscale DCIs.

## Results

### Parallel IM-DD data transmission and chromatic dispersion compensation architecture

Figure 1a shows the schematics of the integrated parallel data communications system for IM-DD applications. At the transmitter, a bright single-soliton microcomb with a smooth spectrum, which is generated by a Si$_3$N$_4$ microresonator, is utilized as the multi-wavelength laser source with evenly distributed and low-noise frequency tones. The comb source is then demultiplexed, independently encoded with electrical data, and multiplexed by silicon microring modulator (MRM) arrays working at several individual wavelength channels. The modulated data are transmitted through an SMF before being sent to the receiver. The MRMs have the advantages of a compact size and a high modulation bandwidth, which have been demonstrated with a modulation bandwidth of 110 GHz[62]. To increase the channels and to reduce the crosstalk, de-/multiplexers based on MRRs[63–65], asymmetric MZIs[66,67], and MRR-coupled MZIs[68,69] can be inserted before and after the silicon MRR modulator arrays. At the receiver end, we employ a silicon-based CDC utilizing cascaded MRRs to simultaneously compensate for the CD induced by SMF transmission across all the wavelength channels. The FSR of the cascaded MRRs is designed to match the wavelength spacing of the frequency combs. Moreover, the

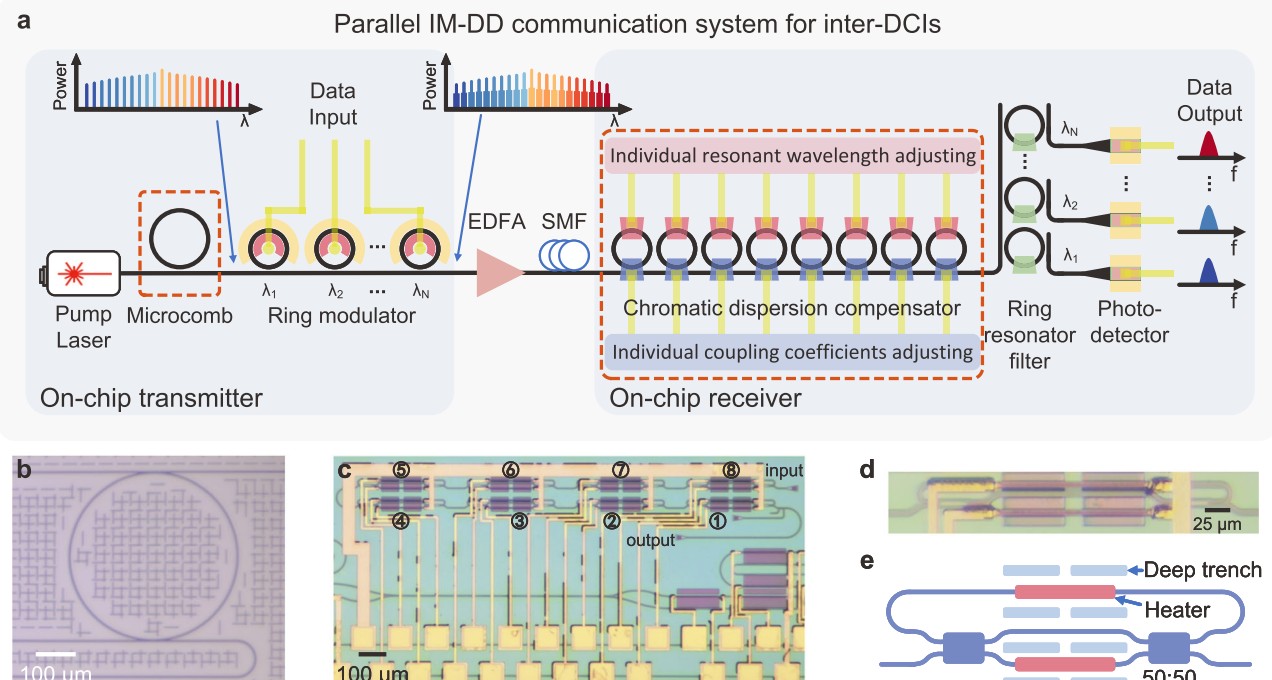

**Fig. 1 | Chip-based parallel data communication architecture for short-reach inter-DCIs. a** Schematics of the data transmission system. At the transmitter, a multi-wavelength laser source with an equal channel spacing is generated by a $Si_3N_4$ Kerr MRR, and then individually intensity-modulated by integrated MRR modulator arrays. After that, the modulated multi-wavelength optical signal transmits through an SMF. At the receiver side, the parallel WDM signals are first processed by a cascaded-MRRs-based optical signal processor for CD compensation, and then demultiplexed by MRR filter arrays and received by photodetectors separately. The red dashed boxes show the on-chip components that have been demonstrated as a proof of concept in this work. **b** Microscope image of the $Si_3N_4$ micro-resonator for single-soliton generation. **c** Microscope image of the silicon-MRRs-based CDC. The numbers label the self-defined sequence of each ring, and the input and output ports are labeled. **d** Enlarged view of a silicon MRR in the CDC. **e** Schematics of the silicon MRR.

resonant wavelengths and the coupling coefficients of each MRR can be independently adjusted, offering a fully reconfigurable configuration that caters to various transmission ranges and operation bandwidths. Following the compensation process, the signal is then separated into distinct wavelength channels with MRR-based de-multiplexers for photodetection. This comprehensive system enables parallel data transmission and CD compensation through the integration of photonic devices, featuring a simple arrangement and remarkable scalability. Consequently, our approach holds great promise for the development of fully integrated photonic circuits, especially for high-capacity inter-DCIs.

In this proof-of-concept study, we used a $Si_3N_4$ microresonator with a bending radius of 232 μm and an average quality (Q) factor of $7 \times 10^5$ for single-soliton generation. Figure 1b shows the microscope image of the fabricated $Si_3N_4$ microresonator. The waveguide cross-section is $1800 \times 800$ nm$^2$ and the measured second-order group-velocity dispersion ($D_2/2\pi$) is 601.2 kHz[70]. The FSR is about 97.7 GHz. We utilized a program-controlled scheme based on thermal auxiliary compensation to generate and stabilize the microcomb in the whole experiments[71]. See Supplementary Note 1 for the detailed program-controlled scheme for the single-soliton generation and stabilization. At the receiver side, the fully reconfigurable silicon CDC comprises 8 identical MRRs. Figure 1c shows the microscope image of the device. Figure 1d, e demonstrate the enlarged microscope image and the schematics of one MRR, respectively. The FSR of these MRRs is around 99.5 GHz for dense WDM (DWDM) scenarios in the C-band. The coupling region of each MRR is replaced by an MZI-based tunable coupler. Both the ring and the coupling region are integrated with a titanium micro-heater for independent thermal adjustment of the resonant wavelength and the coupling coefficient. A pair of grating couplers are utilized to couple the optical signal in and out of this chip. The

footprint of the CDC is 2 mm × 1 mm. See Methods for the detailed design, fabrication, and packaging of the $Si_3N_4$ Kerr comb and of the CDC.

## Characterizations of the CDC and data transmission using a CW laser

To achieve reconfiguration of the CDC for varying lengths of SMF, we devised an optimization algorithm that combines the transfer matrix method with the sequential quadratic programming (SQP) algorithm. This algorithm allowed us to determine the target parameters for the MRR-based CDC. We successfully demonstrated CD compensation for SMF transmission up to 40 km within a bandwidth of 32 GHz. See Supplementary Note 2 for a detailed description of the optimization methodology and of the obtained results. Then, to facilitate an efficient CD compensation for diverse CD values, the optimized resonant wavelengths and coupling coefficients of all the MRRs for different CD conditions were documented in a look-up table. Accounting for the inherent fabrication deviations in silicon photonic devices, we deem it necessary the calibration of the random states for each MRR. Furthermore, we investigated the impact of thermal crosstalk on the tuning of the chip to their optimal configurations, considering the intra- and inter-thermal crosstalk within and between MRRs, respectively. By characterizing the tuning efficiencies influenced by the intra- and inter-thermal crosstalk and adjusting the applied voltages of MRRs correspondingly according to these efficiencies, we successfully compensated for the thermal crosstalk-induced resonant wavelength shifts.

Figure 2a, b shows the measured transmission and group delay responses of CD compensation for a 40-km-long SMF with and without thermal crosstalk compensation (TCC). The implementation of TCC significantly alleviated these shifts, leading to a smoother and more

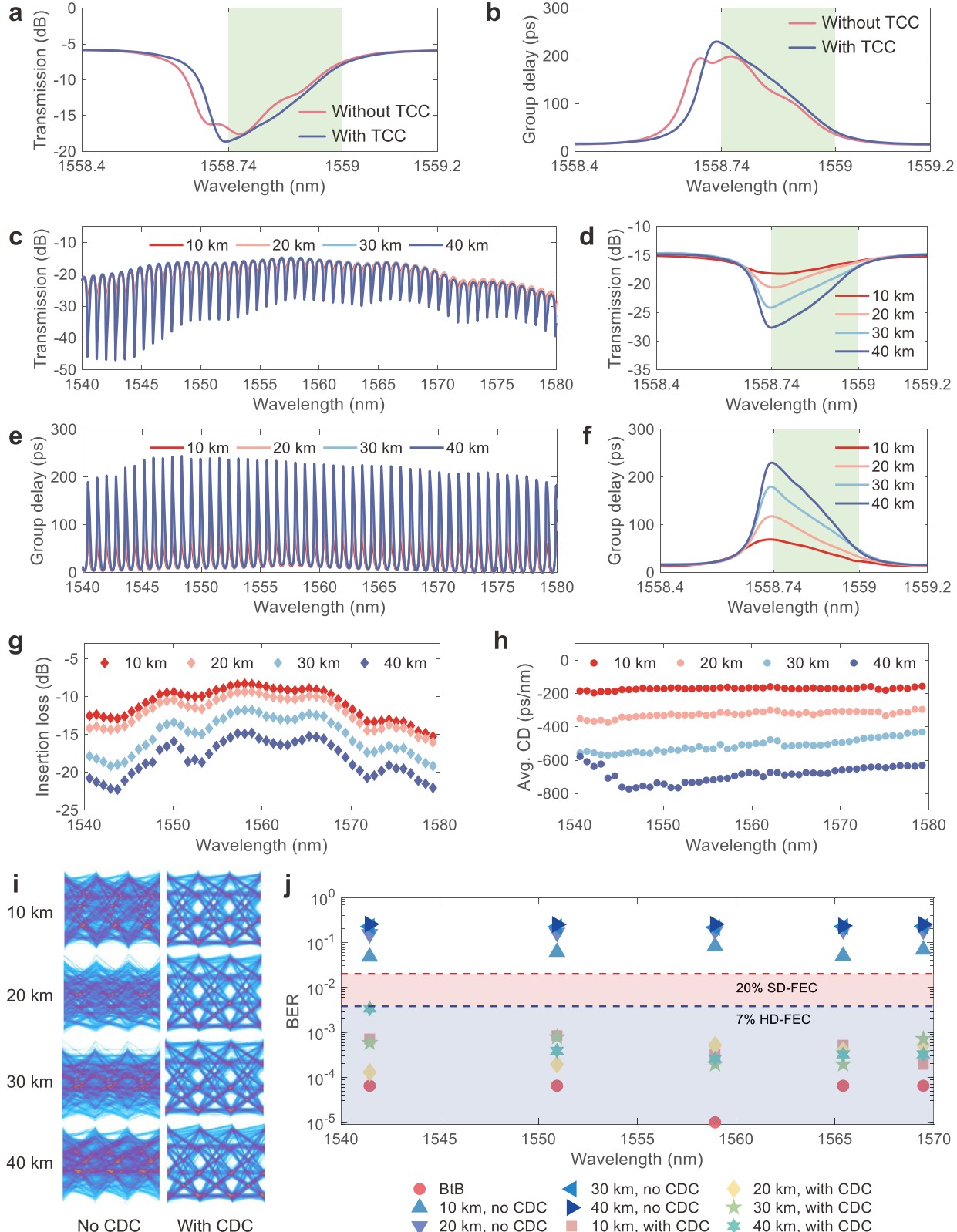

linearly varied response within the operation bandwidth. For detailed information on the calibration process, see Supplementary Note 3. We note that the proposed TCC method is applicable to other photonic devices comprising multiple thermal phase shifters, which can significantly simplify the adjustment of the working states.

Figure 2c, e shows the measured transmission spectra and the corresponding group delay responses covering a wavelength span

from 1540 nm to 1580 nm after the calibration. The MRRs were calibrated at the wavelength of 1558.7 nm with a 32-GHz operation bandwidth for 10-km to 40-km SMFs. The enlarged views at the wavelength around 1558.7 nm are shown in Fig. 2d, f. The transmission spectra include the coupling loss of the grating couplers. The grating coupler has a maximum loss non-uniformity of ~3.5 dB over the 40 nm wavelength, which can be improved by using an edge coupler. The

**Fig. 2 | Characterizations of the CDC and data transmission using a CW-laser.** **a**, **b** Measured (**a**) transmission and (**b**) group delay responses with and without thermal crosstalk compensation for the CD compensation of 40-km-long SMF. TCC: thermal crosstalk compensation. **c** Measured transmission spectra for CD compensation of 10-km-, 20-km-, 30-km-, and 40-km-long SMFs in the wavelength range of 1540 nm to 1580 nm. **d** Enlarged view of the transmission spectra at the wavelength around 1558.7 nm. The operation bandwidth is calibrated to 32 GHz. **e** Measured group delay responses in the 40-nm wavelength span. **f** Enlarged view of the group delay responses at the wavelength of about 1558.7 nm. **g** Extracted insertion loss of each channel for CD compensation of different lengths of SMFs. The insertion loss is obtained at the center of the operation bandwidth. **h** Average CD of each channel for the 4 kinds of CD compensation. **i** Measured PAM4 eye diagrams for the data transmission using a CW-laser as the light source at the wavelength of around 1558.96 nm for the four lengths of SMFs with and without CD compensation. 64-Gbit/s PAM4 signal is transmitted through SMFs. CDC: chromatic dispersion compensator. **j** Measured BERs for data transmission under BtB transmission and SMFs transmissions with and without CD compensation at 5 different wavelengths in the 30-nm wavelength span ranging from 1540 nm to 1570 nm. BERs are much higher than the 20% SD-FEC threshold ($2 \times 10^{-2}$) without the CD compensation, while they are all within the 7% HD-FEC threshold ($3.8 \times 10^{-3}$) after the CD compensation.

increased on-resonance transmission loss at the shorter wavelength side is due to the wavelength-dependent loss and the coupling coefficient of the MRRs. Figure 2g, h shows the extracted insertion loss and the average CD of the CDC. The insertion loss is obtained at the center of the operation bandwidth of each channel, and the average CD is calculated as $D_{avg} = (\tau_1 - \tau_2)/(\lambda_1 - \lambda_2)$, where $\lambda_1$ and $\lambda_2$ are the shorter and longer wavelength ends across the operation bandwidth of each channel, $\tau_1$ and $\tau_2$ are the corresponding group delays, respectively. The insertion loss increases from -8.4 dB to -14.8 dB at the wavelength around 1558.7 nm for CD compensation of 10-km to 40-km SMFs, respectively. We can reduce the insertion loss by lowering the transmission loss of the silicon MRR waveguides. The average CD is −169.5 ps/nm, −325.4 ps/nm, −506.6 ps/nm, and −682.9 ps/nm for CD compensation of 10-km-, 20-km-, 30-km-, and 40-km-long SMFs at the same wavelength. The average CD increases slightly with the wavelength, which we attribute to the coupling coefficient variations of MRRs across wavelengths. See Supplementary Note 2 for the simulated variation of the average CD. The group delay degradation around 1540 nm shown in Fig. 2e is limited by the dynamic range of our measurement equipment, the Photonic Dispersion and Loss Analyzer (PDLA, Agilent 86038B).

Next, we conducted a 64 Gbit/s PAM4 signal transmission experiment using a continuous-wave (CW) laser as the light source to demonstrate the reconfigurability and the wide-band signal processing capability of our CDC in a wide wavelength span. We utilized an offline digital signal processing (DSP) algorithm with a time-domain feed-forward equalization (FFE) at the receiver side to equalize the impairments of the back-to-back (BtB, without the SMFs and the MRR-based CDC) transmission of each wavelength channel. We note that no CD compensation algorithm was used in the signal-receiving end. See Supplementary Note 4 for the detailed experimental setup. We measured 5 different wavelengths distributed from 1540 nm to 1570 nm under the 10-, 20-, 30-, and 40-km-long SMFs with and without CD compensation. The FFE taps were identical to the BtB transmission. The wavelength span was limited by our tunable bandpass filter. Figure 2i shows the measured eye diagrams for the transmissions with and without CD compensation at the wavelength of 1558.96 nm under the four lengths of SMFs. In the absence of CD compensation, the eye diagrams exhibit significant blurring. In contrast, with our CDC, the eye diagrams are successfully restored. The rest of the measured eye diagrams at the other wavelengths are included in Supplementary Note 4. The bit-error ratios (BERs) were also measured and shown in Fig. 2j. The BERs without CDC are all far beyond the 20% soft-decision forward-error correction (SD-FEC) threshold while all of them are below the 7% hard-decision forward-error correction (HD-FEC) threshold after CD compensation, demonstrating that our CDC exhibits a fairly good continuous CD compensation tunability and is capable of parallelly processing the data of at least 37 channels over the 30-nm wavelength range. The slightly higher BERs at wavelengths around 1540 nm and 1570 nm are due to the grating coupler-induced loss non-uniformity. The successful data transmission at the wavelength around 1540 nm indicates the degradation of the average CD of the 40-km-long SMF shown in Fig. 2h is limited by our instrument.

Besides, even with a -10-dB loss variation over the signal bandwidth as shown in Fig. 2a, the BER deviation is negligible compared with the impairments from the CD of the SMFs.

## Parallel data transmission using a comb light source

We demonstrated parallel data transmission and CD compensation using the $Si_3N_4$ soliton comb lines as the light source. In this demonstration, we utilized a 20-km-long SMF in the transmission, and we calibrated our CDC for a 50-GHz operation bandwidth. See Supplementary Note 3 for the measured transmission spectra and group delay responses. The 80-Gbit/s PAM4 and 112-Gbit/s discrete multitone (DMT) signals were transmitted through the SMF, respectively.

Figure 3a shows the experimental setup of the PAM4 signal transmission. The spectrum of the generated single-soliton microcombs is depicted in Fig. 3b. The 10-dB spectral bandwidth of the single soliton is 62.6 nm, with 80 comb lines ranging from 1527 nm to 1590 nm included. Here the 10-dB spectral bandwidth describes the optical wavelength span of the spectrum with the optical power of the comb lines decreased by 10 dB from the comb line with the maximum power. As there is a slight FSR discrepancy between the comb lines ( ~97.7 GHz) and the CDC ( ~99.5 GHz), we only selected 15 comb lines from 1555 nm to 1563 nm by a programmable optical filter for the following data transmission. In principle, by carefully designing the FSRs of both devices with a smaller or even no discrepancy, we can select more wavelength channels for parallel data transmission. We utilized a single modulator to simultaneously modulate all the wavelength channels for simplicity. We detail the experimental settings, and the DSP flows in Methods. The optical spectra of the reshaped and of the modulated comb lines are depicted in Fig. 3c, d, respectively.

Figure 3e shows the measured eye diagrams of the 4th, 6th, 8th, 10th, and 12th channels after transmission and CD compensation. See Supplementary Note 5 for the eye diagrams of all the 15 comb lines. All the 15 wavelength channels have good and comparable eye diagrams for the BtB and the 20-km-long SMF transmission. Figure 3f shows the measured BERs after CD compensation. Although a minor FSR difference between the comb lines and the channels of the CDC leads to a slightly higher BER level around the left and right sides of the wavelength span, all the BERs are below the 7% HD-FEC threshold, enabling a transmitted line rate of 1.2 Tbit/s in total (1.12 Tbit/s net rate after FEC overhead subtraction, if the FEC overhead is encoded). The BERs under different received optical powers for the CW-laser and the microcombs after CD compensation are illustrated in Fig. 3g. The 3rd, 6th, and 9th channels were chosen in the experiments, and the wavelengths of the CW-laser were kept the same as the ones of the selected comb lines. A trivial deterioration is observed between the BERs of the microcomb and the CW-laser transmission, demonstrating a comparable performance between these two light sources. The power penalty is -1 dB at the 7% HD-FEC threshold.

DMT is an IM-DD modulation technique that enables higher transmission rates to be achieved. In addition, DMT can also be used for frequency domain analysis and characterization of CD performance. Therefore, we further carried out high-speed data transmission using DMT modulation format. To achieve a 112 Gbit/s data rate per

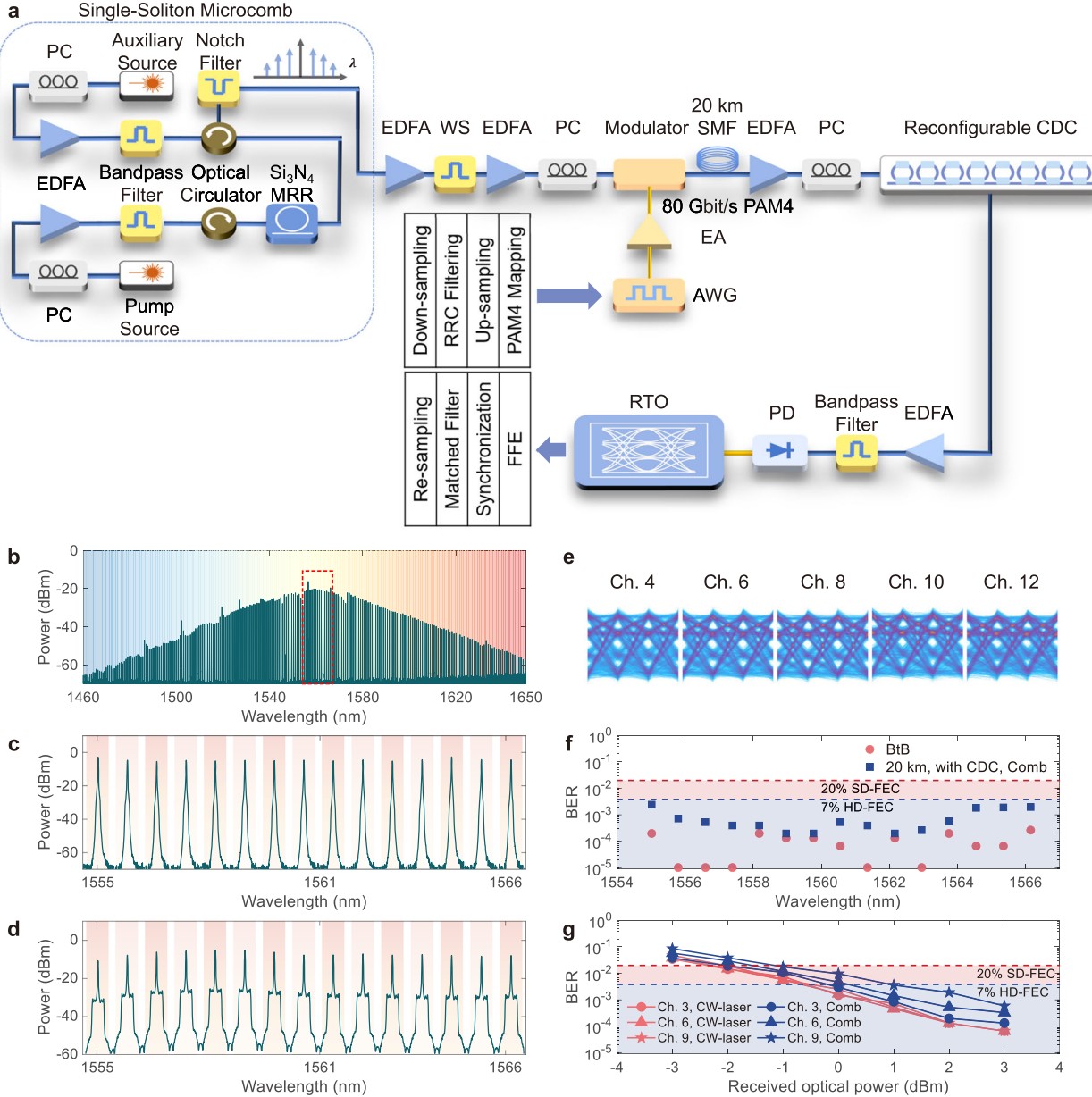

**Fig. 3 | 80-Gbit/s PAM4-based WDM data transmission with microcombs.**
**a** Schematic experimental setup of a single-soliton microcomb-driven parallel sig-nal transmission system. PC: polarization controller; EDFA: erbium-doped fiber amplifier; WS: waveshaper; EA: electrical amplifier; AWG: arbitrary waveform gen-erator; CDC: chromatic dispersion compensator; RTO: real-time oscilloscope; RRC: root-raised cosine; FFE: feed-forward equalization. **b** Measured spectrum of the generated single-soliton microcomb, which shows a sech²-like spectral shape. The utilized 15 comb lines are indicated by a dashed box. **c** Measured spectrum of the chosen 15 comb lines after the waveshaper. **d** Measured spectrum of the 15 comb

lines after the 40-Gbaud PAM4 modulation. **e** Measured eye diagrams of the 4th, 6th, 8th, 10th, and 12th channels after the CD compensation. **f** Measured BERs for the 15 WDM channels under the BtB transmission and the 20-km-long SMF transmission with CD compensation. All BERs are below the 7% HD-FEC threshold after CD compensation. CDC: chromatic dispersion compensator. **g** BERs versus received optical power at the photodiode for the 20-km-long SMF transmissions of the 3rd, 6th, and 9th channels with CD compensation using the CW-laser and the microcomb as the light source, respectively.

lane, we implemented the DMT modulation format to increase the total data capacity. Compared with the PAM4 modulation format, DMT signals use a higher-order modulation format and can achieve higher spectral efficiency for the microcomb-based WDM transmission sys-tem. The experimental setup of the DMT transmission is the same as that for the PAM4 transmission, except that the DSP procedure is different, and the sampling rate of the arbitrary waveform generator (AWG) is set to be 64 GSa/s because of the limitation of the maximum length of the data sequence. The bandwidth of the CDC is still 50 GHz, and the DMT signal was mapped to a 32-GHz bandwidth with 160 sub-

carriers. The details of the experimental setup are discussed in Meth-ods. Before the data transmission, training of the DMT is essential to obtain the bit allocation of each tone. The training was carried out at the 9th channel under the BtB condition. The signal-to-noise-ratio (SNR) response and the bit allocation were obtained, which are shown in Fig. 4a, b. The maximum bit allocation is 4, corresponding to the modulation format of 16 quadratic amplitude modulation (16-QAM). Each sub-carrier was modulated with its own allocated modulation format. Then, we applied the identical bit allocation to all the 15 comb lines to ensure a uniform DMT setting.

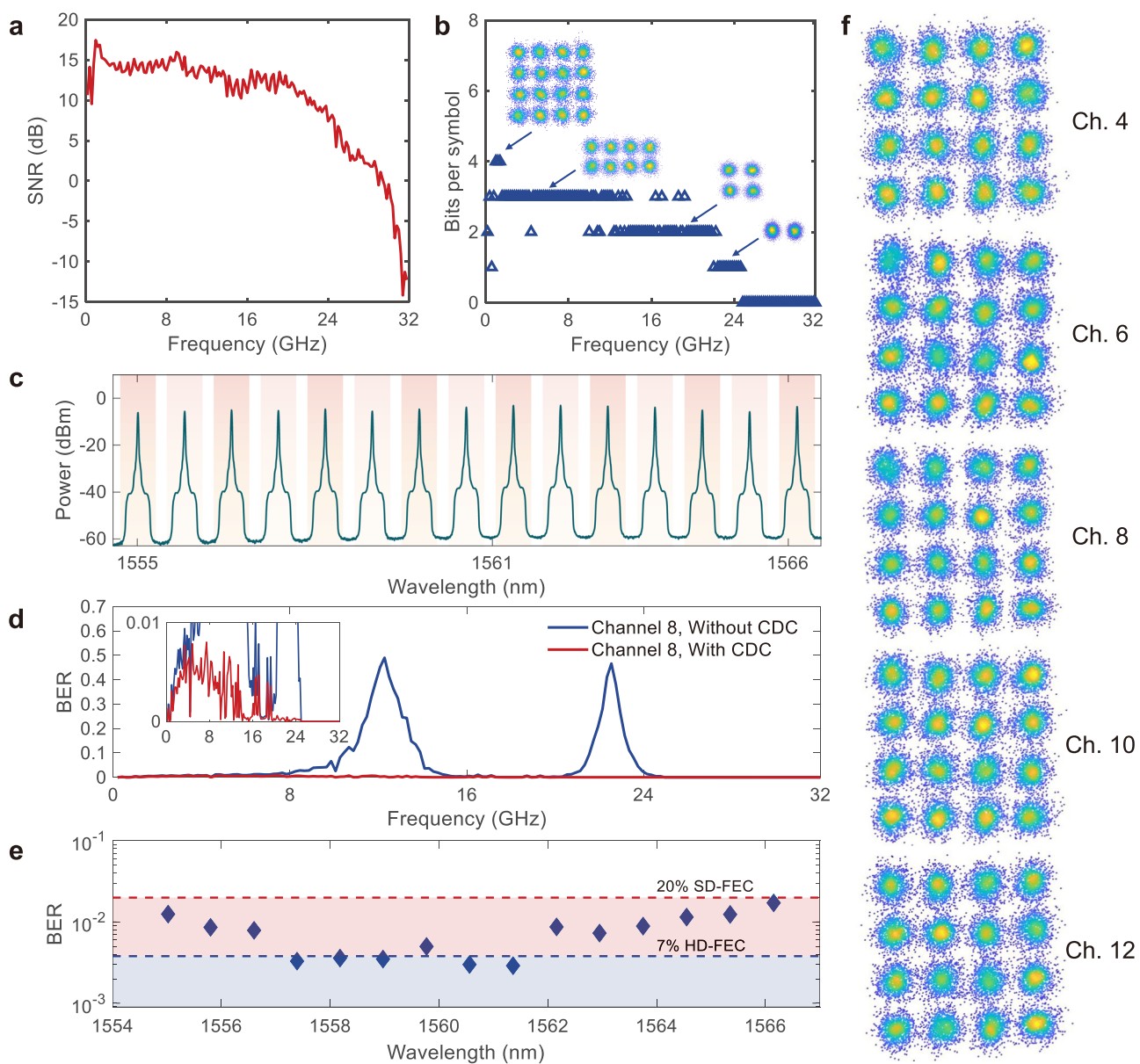

**Fig. 4 | 112-Gbit/s DMT-based WDM transmission with microcombs. a** Measured signal-to-noise-ratio (SNR) response of the transmission system after the DMT training under the BtB transmission. **b** Bit allocations for DMT transmission within 32-GHz bandwidth. **c** Measured spectrum of the 15 comb lines after 112-Gbit/s DMT modulation. **d** Measured total BERs for the 8th channel with and without the CD compensation. Inset: The enlarged view of BER for transmission with CD compensation. CDC: chromatic dispersion compensator. **e** BERs for WDM transmissions with CD compensation. All BERs are within the 20% SD-FEC threshold after CD compensation. **f** Measured 16-QAM constellations for the 4th, 6th, 8th, 10th, and 12th comb line channels at the 7th subcarrier.

Figure 4c illustrates the spectrum of the 112-Gbit/s DMT-modulated comb lines. Figure 4f depicts the measured 16-QAM constellations of the 7th sub-carrier for the 4th, 6th, 8th, 10th, and 12th channels. See Supplementary Note 6 for the constellations of all the 15 comb lines. Figure 4d shows the measured BERs for the 8th channel with and without CD compensation. The inset illustrates the enlarged view of the BER for transmission with CD compensation. At the frequency around 13 GHz and 23 GHz, the BERs reach as high as 0.49 and 0.47 before the CD compensation, respectively. We attribute this to the FSF of the 20-km SMF. See Supplementary Note 6 for the measured $S_{21}$ response of the 20-km SMF. The BERs can be suppressed below 0.01 after CD compensation. The BERs of all the 15 channels after CD compensation are all below the 20% SD-FEC threshold, as shown in Fig. 4e. No FFE was introduced in the DMT transmission. In this case, an aggregate data line rate of 1.68 Tbit/s (1.46 Tbit/s net rate, if FEC overhead is encoded) was achieved.

## Discussion

To our knowledge, we present the first demonstration of on-chip parallel signal transmission and CD compensation using a single-soliton microcomb and a CDC based on MRRs. While several commercially available optical CDCs, including DCFs[72] and tunable dispersion compensation modules (TDCMs)[73,74], can be utilized for parallel CD compensation in WDM systems, DCFs can only compensate for the CD of a fixed length of SMF, and both DCFs and TDCMs are bulky and lack compactness. In contrast, our approach shows the merits of compact size and reconfigurability, which are suitable for inter-DCI applications. Since the power consumption is of vital importance in the inter-DCI applications, we calculate the power consumption of CD compensation for our CDC and coherent transceiver modules and make a fair comparison by normalizing the calculated ones to pJ/bit for every 40 km SMF transmission. To be noted, the power consumption of our parallel scheme includes both the

tuning power of the MRRs-based CDC and the power consumption of the EDFA used for loss compensation of the CDC. See Supplementary Note 7 for a detailed calculation of the power consumption. The key assumptions we make are a ~16% wall-plug efficiency for EDFAs[75], (https://seminex.com/product/chip-laser-diode-8/), and that CD compensation consumes approximately 12.5% of the power used in a coherent transceiver[25,76]. Table 1 gives a comparison of the performance of various integrated CD compensation methods, including the on-chip optical CDCs, the existing commercially available 400 G ZR coherent transceiver modules, and the next-generation coherent modules. We can see that our CDC outperforms the other integrated optical CDCs in terms of reconfigurability, and the supported total data capacity. Besides, our optical CDC outperforms electronic CD compensation approaches based on DSP algorithms, especially in power efficiency, which necessitates individual processing for each wavelength channel. As seen from Table 1, the normalized power consumption of CD compensation for our CDC is estimated to be ~0.3 pJ/bit, which is about 6 times, 4 to 5 times, and even 3 times less than that of the existing 400 G ZR coherent modules, the next generation 800 G ZR coherent modules, and the next generation 1.6 T ZR coherent modules, respectively, due to the parallel CD compensation capabilities of our optical CDC.

The power consumption for CD compensation of our parallel compensation scheme can be further improved by reducing the loss of our CDC chip and also using more wavelength channels. The coupling loss in our work is ~4 dB per grating coupler. It is plausible to minimize the fiber-to-fiber loss associated with the CDC to ~0.32 dB/facet by introducing low-loss and broadband edge coupler[77]. Additionally, MRRs made of widened waveguides can be incorporated to mitigate on-chip losses. Figure 5a shows the estimated normalized power consumption of our CDC and the Marvell 400 G ZR coherent transceiver with various SMF lengths from 40 km to 120 km. We also calculate the power consumption assuming an optimized coupling loss (CL, ~0.64 dB in total) and an improved insertion loss (IL, 0.5 dB/cm waveguide loss, and ~0.1 dB for each MMI) of our CDC. In the calculation, 7 extra MRRs are needed for the CD compensation of an additional 40 km SMF, and every extra 40 km SMF transmission introduces a 10-dB extra loss. Compared with the 400 G coherent transceiver, the power consumption of our CDC increases much slower with the SMF length. For a 120-km-SMF transmission, the power consumption of our CD compensation scheme is estimated to be less than ~0.8 pJ/bit. By considering an optimized coupling loss and insertion loss of our CDC, we can further get a lower power consumption of ~0.5 pJ/bit, which is almost 1/10 of that of the DSP power consumption. Figure 5b depicts

**Table 1 | Performance comparison of various integrated CD compensation methods**

| Method/Company | CBG[82] | MRR[41] | MZI[36] | Marvell[25] | Lumentum[26] InnoLight[27] | Next generation coherent[83,84] | | | This work |
|---|---|---|---|---|---|---|---|---|---|
| Approach | CBG[a] | MRRs | MZIs | DSP | DSP | DSP | DSP | DSP | MRRs |
| Platform/Node | SOI | Si₃N₄ | Si₃N₄ | 7 nm | 7 nm | 5 nm | 5 nm | 3 or 2 nm | SOI |
| Insertion loss (dB) | ~6 | 0.8 | 20 | / | | | | | 10 |
| CD (ps/nm) | 480 | −170 | −500 | Max. −2400 | | | | | −355 (Max. −720) |
| FSR (GHz) | / | 100 | 100 | / | | | | | 99.5 |
| Bandwidth (GHz) | ~110 | 50 | 15 | / | | | | | 50 |
| SMF length (km) | 28 | 10 | 40 | Max. 120 | | | | | 20 (Max. 40) |
| Max. data line rate (Gbit/s) | 60 | 240 | 53.125 | 400 | 400 | 800 | 800 | 1600 | 1680 |
| Power cons.[b] (pJ/bit) | / | | | ~1.7 | ~1.9 | ~1.3 | ~1.6 | ~0.9 | ~0.3 |
| Footprint[c] | 5 mm | 2.2 × 0.56 mm² | 9.89 × 22.5 mm² | 89.4 × 18.35 × 8.5 mm³ | | | 100.4 × 22.58 × 13.0 mm³ | / | 2 × 1 mm² |

ᵃCBG stands for chirped Bragg grating.

ᵇPower cons. stands for power consumption. It has been normalized to the CD compensation of 40 km SMF.

ᶜFor the on-chip optical CDCs, the footprint refers to that of the chips. For the commercial modules, the footprint is the one for the entire module.

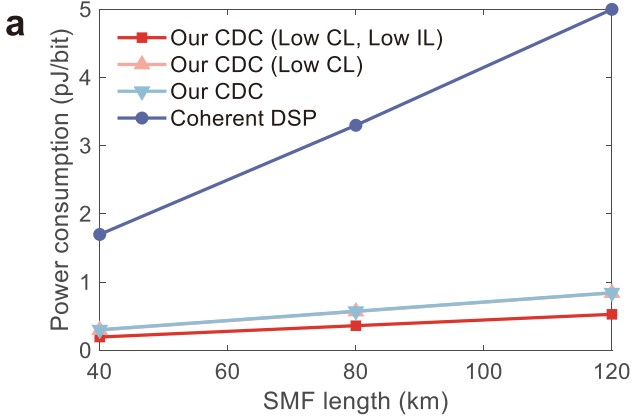

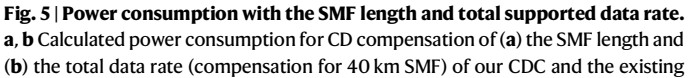

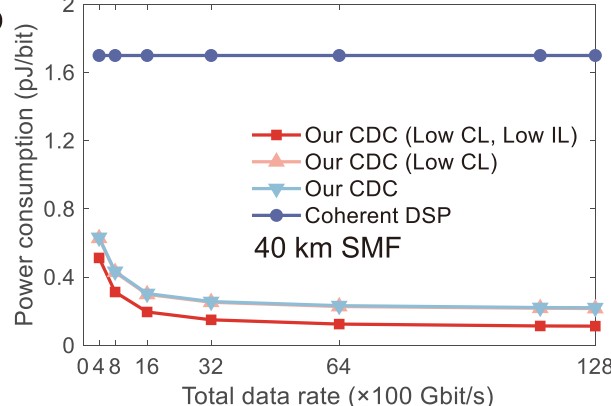

**Fig. 5 | Power consumption with the SMF length and total supported data rate. a, b** Calculated power consumption for CD compensation of (**a**) the SMF length and (**b**) the total data rate (compensation for 40 km SMF) of our CDC and the existing Marvell 400 G ZR coherent transceiver module, respectively. CDC chromatic dispersion compensator, CL coupling loss, IL insertion loss.

the extrapolated power consumption with different total supported data rates from 400 Gbit/s to 12.8 Tbit/s over 40 km SMF transmission. Due to the parallel compensation, the estimated power consumption of our CDC is even lower for higher data rates. It is approximately 15 times less than that of the DSP when optimized CL and IL are considered at a total data rate of 11.424 Tbit/s (112 Gbit/s×102). See Supplementary Note 7 for a detailed calculation.

Under the assumption of a matched FSR between the microcomb and the CDC, we can, in principle, utilize up to 51 channels with an output power of higher than −30 dBm from the microcomb, which encompasses a wavelength span ranging from 1540 nm to 1580 nm, to expand the total data rate to around 6 Tbit/s. To achieve a total bit rate exceeding 10 Tbit/s, the spacing of adjacent comb lines and the channel spacing of the CDC can be further reduced to 50 GHz. We can use the efficient dark soliton microcomb as the light source, which leads to at least 102 available channels of the microcomb and the CDC. We can still transmit 112 Gbit/s data per channel to achieve a total bit rate of larger than 10 Tbit/s. See Supplementary Note 8 for the detailed discussion. Besides, it is worth noting that we can integrate various components such as $Si_3N_4$ microcomb, a silicon MRR-based CDC, MRR modulators, photodetectors, and optical de-/multiplexers onto a single chip. By accomplishing this, we can realize a fully integrated optical communications system tailored specifically for short-reach DCIs.

In the experiment, the single-soliton microcomb in the microresonator with anomalous dispersion demonstrates a conversion energy efficiency of approximately 1%. The power of most of the comb lines is below −20 dBm (considering the coupling and link loss). Investigations highlighting the effectiveness of utilizing a double-microring structure have been proposed with a high conversion energy efficiency of 55% for single soliton generation[78]. Besides, the laser cavity soliton and the dark soliton microcombs have also been demonstrated to have a high conversion efficiency[79–81]. It is emphasized that the efficient soliton microcomb sources can support comb lines with high power, which reduces the gain required by the EDFA. Consequently, this leads to improved noise characteristics and facilitates higher bit rates for each carrier. By implementing these strategies, we believe a higher number of microcomb channels can be effectively utilized for data transmission.

In summary, we have presented parallel data transmission and CD compensation utilizing a $Si_3N_4$ single-soliton microcomb and a silicon MRR-based CDC. With the full reconfigurability of the MRRs, our CDC achieves a continuously tunable CD compensation of up to 40 km SMFs within a 32-GHz bandwidth, and a CD compensation of 20-km-long SMF within a 50-GHz bandwidth. We proposed a method for calibration of the MRR-based CDC, which can eliminate the thermal crosstalk and is applicable to other integrated photonic circuits. Our experimentation with the $Si_3N_4$ single-soliton microcomb revealed the utilization of 15 comb lines for parallel data transmission, resulting in an aggregate total line rate of 1.2 Tbit/s and of 1.68 Tbit/s using 80-Gbit/s PAM4 and 112-Gbit/s DMT modulation signals for 20-km SMF transmission, respectively. The power consumption of the CD compensation of our CDC is estimated to be ~0.3 pJ/bit for 40-km-SMFs, which is ~1/6 and ~1/3 of that of the existing commercially available 400 G ZR coherent transceiver modules and the next generation 1.6 T ZR coherent transceiver modules, respectively. Particularly, when the optimized CL and IL are considered, the estimated power consumption of our CDC can be further reduced to ~0.1 pJ/bit, which is ~15 times less than that of the 400 G ZR coherent transceiver modules for a total data rate of 11.424 Tbit/s. These demonstrations in utilizing the microcomb and the silicon photonic circuits enable a higher data transmission capacity, potentially lower costs, and power-efficient solutions for parallel data processing, especially in WDM-assisted IM-DD systems. And we show the potential of scaling the data rate to tens of terabits based on our proposed system. Our findings represent a

significant advancement toward the pragmatic applications of cost-effective and practical short-reach DCIs.

## Methods

### Design, fabrication, and packaging of the silicon photonic CDC chip

The coupling region of each microring comprises an MZI-based tunable coupler, which comprises two 50:50 multimode interferometers (MMIs), two 165-µm-long straight arms, and waveguide connections between them. The feedback region of each microring comprises a 335-µm-long straight waveguide, and two 10-µm-radius arc bends. The dimensions of the waveguide cross-section are 500 nm (width) × 220 nm (height). Two 160-µm-long and 2-µm-wide titanium micro-heaters, with a resistance of about 1800 Ω, are integrated 2-µm above the center of the 335-µm-long straight waveguide and of the lower arm of the MZI coupler. Deep trenches are positioned beside each micro-heater to suppress the thermal crosstalk within the MRR.

The device was fabricated on a silicon-on-insulator (SOI) platform with a 220-nm-thick silicon top layer and a 2-µm buried oxide (BOX) using electron-beam lithography. All the devices except for the grating couplers were fully etched to a depth of 220 nm. A 70-nm partial etch was implemented on the grating coupler regions. The gold metal connections were deposited above the titanium micro-heaters for electrical contacts. The metal pads were wire-bonded to an external printed circuit board (PCB). The chip and the PCB were placed on a metal shell, and a thermo-electric cooler (TEC) was placed beneath the chip to control the on-chip temperature.

### Design, fabrication, and packaging of the $Si_3N_4$ microcomb chip

The $Si_3N_4$ microresonator has waveguide cross-sectional dimensions of 1800 nm (width) × 800 nm (height), with the fundamental TE mode exhibiting anomalous dispersion. The radius of the microresonator is 232 µm, resulting in an FSR of 97.7 GHz. The ultra-low-loss waveguide was fabricated on an 800-nm-thick $Si_3N_4$ layer, deposited by low-pressure chemical vapor deposition (LPCVD) by Ligentec. The device was patterned using 193-nm photolithography and the quality (Q) factor of the microring resonator is $7 \times 10^5$ to support the generation of the soliton microcomb. The $Si_3N_4$ chip was packaged with a pair of fiber arrays and the coupling loss is ~3 dB/facet after being fixed by an ultra-violet (UV) curable adhesive. We utilized a TEC to control the on-chip temperature for the long-term single-soliton generation.

### Experimental settings and DSP flows of PAM4 parallel data transmission

For the 80-Gbit/s PAM4-based parallel data transmission, the generated comb lines were first amplified to a total power of 20 dBm, and then they were sent into a programmable optical filter (Finisar Wave-shaper 1000 S) to select 15 comb lines and to ensure that the power of each comb line was almost identical to each other. Then, another EDFA was utilized to amplify the 15 comb lines to a total power of 17.5 dBm. Next, the optical comb lines were simultaneously modulated by a commercial 40-GHz intensity modulator (iXblue MXAN-LN-40) driven by a PAM4 signal generated from an AWG (Keysight 8199 A). The modulated comb lines were sent into the 20-km SMF, and the CD was then compensated for by our CDC. Then, one channel of the CD-compensated parallel signals was filtered out by a tunable narrow-bandwidth optical filter and was directly detected by a commercial 50-GHz photodetector (Finisar XPDV2320R). The received electrical signal was finally recorded by a real-time oscilloscope (RTO, Keysight Z592A), and a DSP algorithm, which is identical to the one utilized in the CW-laser transmission, was applied to process offline the received data. At the transmitter side, a high-speed PAM4 signal was transmitted using the root-raised cosine (RRC) filter with a roll-off factor of 0.01 to compress the signal bandwidth. The matched filter and the time-domain FFE were applied at the receiver side to obtain lower

BERs. The instruments and their settings were consistent with those in the CW-laser transmission. The optical power before the photodetector was set to the same power to ensure consistency.

**Experimental setup and DSP flows of DMT parallel data transmission**

Supplementary Fig. 11 shows the schematic experimental setup of the microcomb-based data transmission using the DMT modulation format. A high-speed pseudo-random binary sequence (PRBS) data stream was divided into several parallel low-speed data streams and was mapped to a QAM constellation map. The complex values corresponding to each point in the constellation map were converted to real numbers by an inverse fast Fourier transform (I-FFT). A total number of 160 sub-carriers was mapped to the 32-GHz frequency range. Finally, the signals were converted to a serial data stream after adding the cyclic prefix (CP) to compensate for the multipath delay broadening. At the receiver side, the photocurrent signal was sent to the RTO for data collection. The CP was first removed from the sampled data sequence. Next, the serial data was converted to parallel data and mapped to the QAM constellation map through a fast Fourier transform (FFT). The BERs of the full link were then calculated by comparing the measured data sequence with the transmitter data sequence. We note that we did not encode the FEC overhead in the transmitted data for both the PAM4 and the DMT modulation formats.

## Data availability

The data that support the findings of this study are available from the corresponding authors upon request.

## Code availability

The codes that support the findings of this study are available from the corresponding authors upon request.

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

## Acknowledgements
J.C. acknowledges the funding from the National Natural Science Foundation of China (62120106010). L.Z. acknowledges the funding from the National Natural Science Foundation of China (62090052, 62135010). L.L. acknowledges the funding from the National Natural Science Foundation of China (62075128) and the Shanghai Science and Technology Committee Rising-Star Program (23QA1404500). The authors also thank Prof. Qunbi Zhuge for discussing the power consumption calculation of various CD compensation methods.

## Author contributions
Y.Liu. designed, simulated, and characterized the CDC. Y.Liu. performed the experiments of the CW-laser-based 64-Gbit/s PAM4 data transmission. H.Z. designed, simulated, and characterized the Kerr microcomb. Y.Liu. and H.Z. conceived the link architecture and performed the high-speed data-transmission experiments of the microcomb-based parallel signal processing. J.L. conducted the offline DSP. All authors helped analyze the data. L.L., J.D., Y.Liu., H.Z. and J.L. prepared the manuscript. Y.Liu., H.Z., J.L., L.L., J.D. and L.Z. revised the manuscript. J.C., Y.Li., Z.H. and A.W. Poon provided suggestions and feedbacks during the revisions. L.L. and J.D. co-supervised the research.

## Competing interests
The authors declare no competing interests.
