## [Peer Review File · Nature Communications]

Parallel Wavelength-Division-Multiplexed Signal Transmission and Dispersion Compensation Enabled by Soliton Microcombs and MicroringsEditorial Note: This manuscript has been previously reviewed at another journal that is not operating a transparent peer review scheme. This document only contains reviewer comments and rebuttal letters for versions considered at *Nature Communications*.

REVIEWER COMMENTS:

Reviewer 1: (Remarks to the Author):

The authors claim to demonstrate the first on-chip parallel signal transmission and dispersion compensation using a single-soliton microcomb and a dispersion compensator based on micro-ring resonators (MRRs). The experimental results are impressive. Although the concept of using MRRs for dispersion compensation has been previously reported and there is nothing unique about the design and fabrication of the Soliton source, the integration of both devices in a single experimental setup is novel and highlights the scalability silicon photonics offers.

Main comments:

- Despite the significance of the IMDD results presented in this work, it is well-known that the industry is accepting coherent transmission in a shorter reach. Currently, the 800 CWDM4 IMDD systems are standardized for a maximum of 10 km. Hence, I disagree with the overall claim of the manuscript that C-band IMDD systems are an attractive solution for DCIs. Can the author comment on this with supporting references from the industry? Yet, I still see some potential applications for C-band IMDD, but no way it will go into the DCIs market.
- Regarding the MRR CDC, would it be beneficial to design it for a fixed dispersion of 20 km and tune it for ± 20 km; instead of going all the way from 0 to 40 km? Would it reduce the insertion loss?
- What limited the transmission performance of PAM4 to 80 Gbps? if the spacing between the lines is > 90 GHz, then, you should be able to transmit up to 90 Gbaud (180 Gbps) PAM4. If it is the bandwidth of the MRMs, do not you think it is better to design the comb source with narrower spacing? It might have implications on the reliability of the MRM tunability, please comment on this in the supplementary files.
- In line 287, the authors used the CEI-112G standard to justify employing DMT modulation; however, this standard is based on PAM4. Please revise it.

Minor suggestions:

- I suggest replacing ref. 4 with 10.1109/LPT.2021.3124196, as it is an updated version of the ref. 4.
- Also, I suggest adding ref. 10.1109/LPT.2023.3285881 to the manuscript, as these are the most recent IMDD results to the reviewer's best knowledge.
- In Fig. 2 (i-j), I prefer using CDC for chromatic dispersion compensation instead of just DC (Just for clarity).

Reviewer #2 (Remarks to the Author):

The manuscript "Parallel Wavelength-Division-Multiplexed Signal Transmission and Dispersion Compensation Enabled by Soliton Microcombs and Microrings" by Y. Liu et al. presents a study on chromatic dispersion compensation over inter-DC (i.e. DCI) relevant distances using a microring-based dispersion compensator, leveraging the link between frequency comb line spacing and dispersion compensation band spacing to enable parallel dispersion compensation.

The work is very well done on a technical level, and so I have attempted to try to understand the impact of this work. I am not sure that the potential for impact is well presented by the authors in the current manuscript.

I think the impactful parts of this study are to show that aligned dispersion compensation between a microcomb-supported set of signals and the dispersion compensator can be made to work well over multiple channels, and that the power consumption of this stage can be low. I think both need to be well described and shown in the manuscript for impact.

Given that these are the important aspects, the major gap in this publication is a proper comparison of energy consumption of the presented system and the (ballpark) energy consumption of coherent DCI links. I'll detail this further below.

There is little novelty in the use of a microcomb to support an IM/DD system, it is an open question as to whether IM/DD is actually a compelling option for inter-DC (i.e. DCI) applications, against the coherent 400G ZR (and soon, 800G ZR) products are available now.

As such, for this work to have the potential for impact matching expectations for Nature Communications, I think the power consumption issue needs to be a defensible focus of this work. Otherwise, all this shows is a relatively straightforward proof-of-concept demonstration of a concatenation of known technologies, which I would suggest is not at the level of impact desired for Nature Communications.

I've put down some specific points (moving in order of where points appear in the manuscript) to help the authors revise their manuscript to address these concerns.

- 1) The authors state, in the abstract, that dispersion compensation are of high power consumption or lack reconfigurability. These are then points that need to be quantified, for a quantitative analysis to really prove the point that the authors have solved this issue. How high is the power consumption of current methods? How much reconfigurability is needed for inter-DC (DCI) links?
- 2) Again, in the abstract, the authors state a power consumption of below 160mW. This is then an important point to really analyse in the main manuscript. It also needs to be a wholistic comparison – what bounds do the authors put on their power consumption analysis, to make sure that there is a fair comparison to other dispersion compensation techniques?
- 3) In the abstract, the authors state that they present an aggregate bit rate of 1.68 Tbit/s. They need to be clear at this point that they are presenting a number for the line rate (i.e. without including reduction of data rates due to coding overhead)
- 4) I am unaware of the originality of the authors approach. They state in the abstract that they "present an original scalable on-chip parallel IM/DD data transmission system". Given that all of the building blocks of their system have been demonstrated separately, or in different combinations, and that the authors do not present a fully chip-based demonstration themselves ... I would not agree that this is "original", which suggests that the novelty in their work primarily lies there.
- 5) In the introduction, the authors write:
"The total data capacity grows 80 times from 640 GHz in 2010 to 51.2 THz in 2022 for data switching in datacenter interconnects (DCIs) ..."
This is a problematic sentence in that data carrying capacity is measured in bits/s, while the authors here use units of Hz. This also conflates the concept of switching with data transmission ... the authors should rethink what they are trying to present here and be clear in setting up this key problem statement.
- 6) The authors make the bold claim that:

"For short-reach DCIs spanning distances of less than 40 km, the utilization of intensity modulation and direct detection (IM-DD) has emerged as a practical and preferred solution [3-5]." This is a rather controversial statement, and is not really sufficiently backed up by the papers cited.

I would point the authors to the development of the 400G ZR standards, and the use of coherent 16-QAM within this for DCI applications (i.e. 2km to 80km or so).

Coherent is also starting to be considered by R&D divisions of major datacentre companies for intra-DC links (i.e. <2km). For example, in X. Zhou et al. (Google), JLT, v38 p475, 2020, they show that coherent looks good at >2km reach for 1.6 Tbit/s links. For DCI, then coherent already looks good at the 400G ZR level for > 2km reach (R. Nagarajan (Marvell), JLT, v39 p5221, 2021). I think, that for the authors to really make a fair comparison, they need to consider these relatively recent developments in coherent DCI technologies to understand the potential future impact of their work and the comb-based IM/DD approach. Otherwise, their work is not put in the relevant context, where there dispersion compensator may make a compelling case for the use of IM/DD over inter DC distances.

7) The authors write in their introductions:

"To address the dispersion effect, various electronic and optical approaches have been employed, such as digital signal processing (DSP) [11-13] and dispersion compensating fibers (DCFs). However, these methods are burdened with limitations, particularly in terms of a high energy consumption and associated costs [14]."

They then switch topics to microcombs.

a) The authors need to separate their arguments against DSP and optical methods here. DCFs are clearly not high energy consumption, especially compared to the author's own chip-based approach. DSP is also fairly clearly not comparatively high cost, given its prevalence in both IM/DD and coherent DCIs.

b) The authors should here start to acknowledge the large number of tunable optical chromatic dispersion compensation techniques available, especially before the advent of coherent systems c.a. 2006. This then sets their compact chip-based system up in context.

c) Ref [14] does not really make the point about high energy consumption and cost ...

8) Ref [26] is claimed to be variously based on "perfect soliton crystals" and "soliton crystals with defects" ... which one is it (I think the "with defects", looking at [26])?

9) Is the rate of 55 Tb/s line rate for [25] (capacity before taking FEC into account), or net rate (capacity after subtracting for the FEC overhead)? (I think line ...)

10) Is the rate of 40.1 Tb/s for [21] line or net? (I think net ...)

11) Again, the introduction closes with a paragraph that suggest that the authors system is "original". I would again suggest that this is not really the case, and that that they are looking at one building block in the chain. I would remove this focus, and instead shift to focusing on the power consumption aspect.

12) At the end of the introduction, the authors state that they are using PAM4 and DMT as modulation schemes. To start to understand the power consumption of these schemes, the authors can refer to J. Cheng (Alibaba), Proc. OFC, W1F.2, 2019. Here they present graphs that show the breakdown of power consumption of PAM4 and DMT schemes, including DSP. This shows that DMT DSP consumes about as much power as coherent (noting that coherent explicitly includes chromatic dispersion compensation), while PAM4 has about half the power for DSP as coherent (noting that PAM4 DSP explicitly does not include dispersion compensation, just an FFE as the authors here use). It may be that the authors wish to foreshadow this here, or introduce this information later.

13) Figure 1 is somewhat misleading about what has been achieved in this demonstration. It would be better to clearly outline, in figure 1a, what has been demonstrated in this paper (i.e. the CD compensator), while clearly showing that this has not been implemented here with the ring-based Tx and Rx

14) The authors state around line 184, that:

"The insertion loss increases from ~8.4 dB to ~14.8 dB at the wavelength around 1558.7 nm for dispersion compensation of 10-km to 40-km SMFs, respectively."

Two things to note:

a) The insertion loss changes significantly over the signal bandwidth, as shown in Fig 2a for example. This then means that the FFE (PAM) or 1-tap eq. (DMT) used in the receiver DSP will need to equalize out this loss, potentially increasing DSP complexity somewhat compared to a system without this eq.

b) This insertion loss means that the signal power needs to be boosted by an EDFA. To make a fair comparison with other dispersion compensating techniques, you need to then include EDFA power consumption.

A back of the envelope calculation:

Assuming your top received power is 3dBm (2mW), and your loss is about 10 dB (i.e. 90% loss), the EDFA needs to add in 1.8mW of power per channel. EDFAs have a wall-plug efficiency of about 10%, so the EDFA will need to be supplied with 18mW per channel. For 15 channels, this comes to 270 mW.

So, you will need to include EDFA power into your power consumption calculations, which will increase your power consumption by a factor of about 2x. For a 51 channel system, as you propose later on, then your power consumption is about 1W total (ring heaters + EDFA).

15) The authors use a generic 20% SD-FEC and 7% HD-FEC threshold for their work. These are thresholds that are often associated with coherent systems, and are not optimized for energy consumption. There are, however, a range of FECs that are optimised for energy consumption in the 400G ZR standard (and some details for the proposed next-gen 800G ZR). E.g. the concatenated staircase FEC with a threshold of BER = $1.25e-2$, or KP-4 FEC with a threshold at BER = $2.2e-4$. These are probably more appropriate or relevant thresholds to consider if thinking about minimising energy consumption in the receiver side. OFEC is being considered for the future 800G ZR standard, and you can have a look at that in W. Wang et al., JLT, v41 p926, 2023. The present a "simplified OFEC" with 3x the power consumption of standard 400G ZR FEC, but with an increase pre-FEC BER threshold of $1.8e-2$.

I would suggest that the authors look at these options. Note that R. Nagarajan (Marvell), JLT, v39 p5221, 2021 state that KP4 FEC uses approximately 100x less power than the 400G ZR CFEC, and that the 400G ZR CFEC is stated to have a power consumption of 560 mW (similar to your dispersion compensator, including EDFA power to compensate for insertion loss).

16) In figure 3, the caption for part f should state the received optical power at the photodiode.

17) The main manuscript includes these references to the power consumption of the dispersion compensator:

"The maximum power consumption of our dispersion compensator is as low as 160 mW. See Supplementary Note 6 for the detailed calculation of the bit rate and Supplementary Note 3 for the consumption."

As above, I don't think that this is sufficient analysis in the main manuscript, considering that the power consumption of your chromatic dispersion compensator is a really key point to what I see as the impact of this paper.

18) Ok, given that I've pointed out things to do with power consumption a few times, here's how I'd provide a back of the envelope calculation for the power consumption of CD compensation for 400G ZR ... and you need about 3 of these to get up to the data rate of your PAM4 system.

From R. Nagarajan (Marvell), JLT, v39 p5221, 2021: About 50% of the total power consumption for 400G ZR transceivers is the DSP. They also state that a 400G ZR module eats up about 16W worth of power. So, this gives about 8W per 400G ZR module dedicated to DSP.

Then, let's assume that the equalizer only eats up 10% of this power (this is an estimated proportion that I think might provide a lower bound for the DSP power consumption estimate), then the equalizer (that achieves CD compensation) eats up 0.8W per 400G, so 2.4W for 1.2 Tb/s. This would tend to indicate that your IM/DD based CD compensation approach (at about 500mW, including EDFA for insertion loss mitigation) is lower power by a significant margin than an equivalent system using 400G ZR.

Please note that 800G ZR is expected to use about half the power per 100 Gb/s (27W / 800 G vs 24W / 400 G ... this is as per Nokia).

I provide this to show the authors how feasible it is to make the power comparisons that I suggest.

19) In the discussion, the authors claim that they could with a matched microcomb/dispersion compensator FSR hit 51 wavelengths. Given that this is the basis for claiming scalability up to 10 Tbit/s, there needs to be some more details as to this claim (especially given that there will be less power per line available when going to wavelengths further from the pump ...)

20) In the discussion, the authors claim that they can go to a 50GHz spacing. Given the edges of their dispersion compensator system, I'm not sure this is going to be entirely true? At least, one would expect that the symbol rate per line would need to be reduced? There needs to be a bit more backing for this "scalable to beyond 10 Tb/s" claim ...

21) The authors quite rightly begin to critique the conversion efficiency and power consumption of

the microcomb. They focus on the dual-ring bright solitons from the Chalmers group (they should also note that that work is now out in Nature Photonics?). The authors might also like to consider the 75% efficiencies shown for laser cavity soliton microcombs (H. Bao et al., Nature Photonics, v13, p384, 2019).

22) When the authors are considering the dual ring bright combs, and state:

“resulting in increased power levels for the majority of the comb lines to -10 dBm, ...”

... this is maybe not quite true, as the high efficiency states occur for low pump powers, and so per line comb power goes down.

Please reconsider this statement ... it's not as straightforward as you present it, I think.

23) The authors state:

“To further enhance the overall performance of the system, it is plausible to minimize the fiber-to-fiber loss associated with the dispersion compensator by introducing low-loss and broadband edge couplers [59-61].”

This would seem to be something you could put a number to fairly easily by looking at the out-of-band insertion loss vs in-band insertion loss for your device? So, if you took a state-of-the-art number for coupling loss, and then took the difference between that and the approx. 6dB out-of-band insertion loss you measure, then you would get a measure of the minimum expected insertion loss (instead of a vague promise for lower losses)?

Reviewer #3 (Remarks to the Author):

In this manuscript, the authors demonstrate a parallel WDM data transmission using a silicon photonic coupled microring resonators to compensate for fibre dispersion. Parallel transmission with an integrated soliton microcomb source was also demonstrated. Although dispersion compensation based on single or cascaded microrings, as well as massive parallel data transmission using a microcomb as a multi-wavelength light source have been demonstrated previously, it appears that in this work the authors are able to combine both microcomb technology and on-chip dispersion compensator for parallel WDM transmission for the first time, demonstrating the potentials of on-chip solutions.

The following points need to be addressed before the manuscript can be considered for publication in Nature Communications.

1. The dispersion of the microrings was designed and calibrated at one wavelength channel. Over a wide wavelength range, the waveguide in the rings will have different dispersion and the couplers will have wavelength dependent coupling coefficients. How do these factors impact the dispersion of the dispersion compensator on the wavelength channels far away from the designed/calibrated channel? Please include the simulated and measured group delay for different wavelength channels.

2. The dispersion compensator shows a non-uniform transmission with more than 10 dB variation within the operational bandwidth. Please discuss the impact of this non-uniform loss on the transmission penalty.

3. In this work, the authors demonstrate the maximum operational bandwidth of 50 GHz with 8 microrings. While the FSR (or WDM channel spacing) is 99.5 GHz. The spectral utilisation is just above 50%, resulting in low link spectral efficiency. Increasing spectral utilisation would require more rings, hence increasing power consumption. This is also true for increasing fibre length. How does the number of rings required and power consumption increase with operational bandwidth and fibre length? What is maximum operational bandwidth that can be achieved with 40 km fibre length and the power consumption? How many rings are required to compensate for 40 km fibre with a 50 GHz bandwidth?

4. In Fig. 4(h), the average dispersion for 40 km fibre increases for channels close to 1540 nm.

This behaviour does not present for other fibre lengths. What is the cause of this behaviour?

5. In the demonstrations, 4-PAM and DMT with 16-QAM modulation formats were used. Can the number of levels in the multi-level modulation be increased to further increase the data rate? What is the limiting factor in the current demonstration? Is it the dispersion compensation or others?

6. In the discussion, the authors suggested to reduce the spacing between adjacent comb lines to 50 GHz to increase the total data rate exceeding 10 Tbit/s. To do this, the FSR of the dispersion compensator also need to be halved to 50 GHz. As a consequence, the operational bandwidth of the dispersion compensator will be reduced. In order to maintain the same operational bandwidth and per channel data rate, the number of rings in the dispersion compensator need to be significantly increased (see point 3 above). If keeping the number of rings unchanged, will half the ring FSR also cause the operational bandwidth reduce by half?

7. Please comment on the long-term stability of the dispersion compensation performance and thermal crosstalk compensation as well as system BER.

8. Please benchmark the performance of the on-chip dispersion compensator used in this work with other dispersion compensation techniques for parallel WDM transmission (such as DCF).

Other minor comments:

In the first paragraph of the Introduction section, the unit for data capacity was incorrectly used (as GHz).

Responses to the reviewers

Reviewer 1

>>The authors claim to demonstrate the first on-chip parallel signal transmission and dispersion compensation using a single-soliton microcomb and a dispersion compensator based on microring resonators (MRRs). The experimental results are impressive. Although the concept of using MRRs for dispersion compensation has been previously reported and there is nothing unique about the design and fabrication of the Soliton source, the integration of both devices in a single experimental setup is novel and highlights the scalability silicon photonics offers.

We thank the reviewer for the insightful summary and evaluation of our work.

Main comments:

Comment #1-1

Despite the significance of the IMDD results presented in this work, it is well-known that the industry is accepting coherent transmission in a shorter reach. Currently, the 800 CWDM4 IMDD systems are standardized for a maximum of 10 km. Hence, I disagree with the overall claim of the manuscript that C-band IMDD systems are an attractive solution for DCIs. Can the author comment on this with supporting references from the industry? Yet, I still see some potential applications for C-band IMDD, but no way it will go into the DCIs market.

Response:

We thank the reviewer for the comment. We agree with the reviewer that for data-center interconnects (DCIs) based on the intensity-modulation direct-detection (IM-DD) scheme, most of them are realized by O-band CWDM technology. On the other hand, although DCIs based on C-band transmission suffer from large chromatic dispersion (CD), some industrial optical transceivers are designed for the C-band. We discuss this issue in the following parts:

Usually, the IM-DD scheme is utilized in short-distance interconnects, especially in the intra-DCIs within 2-km fiber transmission, due to its simple structure, low power consumption, and low cost. The IEEE 802.3df task force has standardized the 800G-DR4 and 800G-FR4 baseline based on the PAM4 IM-DD system¹. As the reviewer mentioned in the comment, an 800G LR4 transmission system (for a maximum length of 10 km) based on the CWDM4 IM-DD scheme has been standardized recently². Among these standards, O-band transmissions are preferred and utilized due to their small CD. Besides, FS has proposed a PAM4-based 400G ER8 optical transceiver for 40-km fiber transmission at the wavelength around 1310 nm³.

Yet, the inter-DCI is accepting a combination of IM-DD and coherent technology⁴. And we find some optical transceivers that are designed for C-band. It is noted that the Colorz 100G DWDM optical platform proposed by Marvell Technology⁵ utilizes the PAM4 modulation format on 100-GHz DWDM wavelength grids to transmit data across a length

of 80 km. In addition, FS proposes a DWDM-compatible PAM4-based 80 km 100G optical transceiver module for the C-band transmission⁶. The two optical transceivers are all based on the quad small form-factor pluggable 28 (QSFP28) form factor. Although these optical modules need extra dispersion compensation modules (DCMs) to compensate for the large CD in the C-band, a recent analysis showed that compared to the use of dedicated transport solutions for data centers situated at a distance of up to 80 km, users can save 58% to 67% on the total cost of ownership (TCO) of transceivers when they utilize the QSFP28-DWDM-based optical transceivers that are installed directly in network devices⁷.

We do not intend to compare these two technologies and say which is better. We mainly focus on the IM-DD scheme in our work, so as to prove the capability and performance of our device for broad application potentials in future optical communication systems. We would like to modify the statement of the introduction of our work on C-band-based DCI as follows:

Revision:

In the Introduction of the manuscript (from line 44 to line 60):

“There are mainly two technical approaches to the DCIs, namely the intensity modulation and direct detection (IM-DD) and the coherent schemes. Until recently, the IM-DD technology has been the clear choice for intra-DCIs (<2 km), owing to its low energy consumption and easy implementation³⁻⁵. However, the scaling of data rate beyond 1.6 Tbit/s is challenging even for the O-band intra-datacenter scenarios as the IM-DD system is less tolerant to optical impairments such as the fiber chromatic dispersion (CD)³. The system suffers from an even larger CD in the C-band, which also limits the scaling of the data rate of IM-DD systems working in the C-band. For the coherent technology, it has been widely used in long-haul and metro optical networks owing to its high receiver power sensitivity and high spectral efficiency³. Due to the considerable reduction of cost and power consumption, the coherent technology has shown competitive performances for transmission within 100 km single-mode fiber (SMF)³. Recently, data transmission using coherent technology has already reached the fiber length of less than 10 km⁶⁻⁸, some of which have even been successfully demonstrated for intra-datacenter applications^{7, 8}.

As the typical distance between two data centers is within 80 km and the majority (~83%) of the DCIs are within 40 km⁹, we mainly focus on the short-reach inter-DCIs with a fiber length of up to 40 km and the utilization of IM-DD scheme for this scenario due to the inherent advantages IM-DD offers, including cost-effectiveness, low power consumption, and a compact physical footprint.”

References

1. <https://ieeexplore.ieee.org/abstract/document/8205220>
2. https://grouper.ieee.org/groups/802/3/dj/public/23_07/rodes_3dj_01_2307.pdf
3. <https://www.fs.com/products/179940.html?attribute=62033&id=1874948>
4. Nagarajan R, Lyubomirsky I, and Agazzi O, Low power DSP-based transceivers for data center optical fiber communications (invited tutorial). *Journal of Lightwave*

Technology, **39**, 5221-5231 (2021).

5. <https://www.marvell.com/content/dam/marvell/en/public-collateral/optical-modules/marvell-optical-module-in-q2ay2-xx-product-brief.pdf>

6. <https://www.fs.com/products/138167.html>

7. <https://datacentreview.com/2021/09/connecting-modern-data-centres-with-dwdm-technology/>

Comment #1-2

Regarding the MRR CDC, would it be beneficial to design it for a fixed dispersion of 20 km and tune it for ± 20 km; instead of going all the way from 0 to 40 km? Would it reduce the insertion loss?

Response:

We thank the reviewer for the comment. This is a very good point. Figure R1a shows the schematic of the MRR CDC realized by a combination of a fixed CD compensation of 20 km and a tunable CD compensation of ± 20 km. The CD compensation of the negative length of SMFs (e.g., -10 km and -20 km) indicates a positive CD is realized by the CDC. The fixed and tunable CD compensation can be both realized by the cascade of 5 MRRs. All the MRRs are capable of resonant wavelength tuning. The difference between these two groups of cascaded MRRs is that the coupling coefficients of MRRs are fixed and tunable for the fixed and tunable CD compensation, respectively. We will discuss various issues of this scheme in the following sections:

Fig. R1 a Schematic of the scheme that combines the fixed and tunable CD compensation. **b, c, d** Simulated transmissions of **(b)** the fixed CD compensation for 20 km SMF using 5 MRRs, **(c)** the tunable CD compensation for 20, 10, -10, and -20 km SMFs, and **(d)** the overall CD compensation for 10, 20, 30, and 40 km SMFs.

(i) Insertion loss. The simulated insertion loss of the fixed CD compensation of 20 km SMF, of the tunable CD compensation of 20, 10, -10, and -20 km SMF, and the overall CD compensation of 10, 20, 30, and 40 km SMFs are shown in Figs. R1b to R1d, respectively. In the simulation, we set the propagation loss of the waveguide to 4 dB/cm and neglected the loss of beam splitters (i.e. MMI in our work). Compared with our proposed structure which consists of 8 MRRs, the overall insertion loss of the CD compensation within the operation bandwidth of 20, 30, and 40 km SMF are comparable, while it is larger for the 10

km condition realized by the combination scheme.

(ii) Control complexity. Although 15 phase shifters (10 for resonant wavelength tuning and 5 for coupling coefficient tuning) are included, which is slightly less than the 16 phase shifters in the proposed 8 MRRs scheme, the control complexity of the scheme shown in Fig. R1a will be comparable or even higher than that of the 8 MRRs scheme because more MRRs are utilized. We should note that the phase shifters for the resonant wavelength tuning of the fixed CD compensation cannot be taken out because it is hard to realize the desired resonant wavelengths of MRRs by passive design.

(iii) Power consumption. Based on the measured power consumption of the 8 MRRs (<160 mW), we assume that the average power consumption of each MRR is ~20 mW, leading to ~10 mW power consumption of each phase shifter. In this case, the power consumption of the combination scheme will be ~150 mW, which is slightly smaller than that of our proposed 8 MRRs.

Therefore, compared with the 8 MRRs scheme, the combination scheme has comparable insertion loss and control complexity, and a slightly lower power consumption.

Comment #1-3

What limited the transmission performance of PAM4 to 80 Gbps? If the spacing between the lines is > 90 GHz, then, you should be able to transmit up to 90 Gbaud (180 Gbps) PAM4. If it is the bandwidth of the MRMs, do not you think it is better to design the comb source with narrower spacing? It might have implications on the reliability of the MRM tunability, please comment on this in the supplementary files.

Response:

We thank the reviewer for the comment. In our experiment, we measured the eye diagrams of a 45 Gbaud PAM4 signal, and they were blurred even under the BtB condition (with FFE) using a CW laser as the light source. We think the transmission performance is limited by the intensity modulator, as it has a 3-dB bandwidth of ~32 GHz. And we finally set the baud rate to 40 Gbaud. Besides, the operation bandwidth of our CDC is only 50 GHz for 20 km CDC, which can be further increased by cascading more MRRs.

We think it is better to design the microcomb with a narrower spacing. We assume a channel spacing of 50 GHz for both the microcomb and the CDC. In principle, if the FSR is halved and we focus on a fixed wavelength span (e.g., 1540 nm to 1580 nm), we can double the total data rate providing that there is a perfect match of the FSR between the microcomb and the CDC. For example, in our experiment, the bandwidth of the 112 Gbit/s DMT modulation is around 40GHz, which means that if we want to maintain the 112 Gbit/s data rate per line, the CDC should have an operation bandwidth of at least 40 GHz when the FSR is 50 GHz. This is feasible with 8 MRRs and the transmission and group delay responses are depicted in Fig. R2. In this case, the total data rate can be doubled. If the microcomb and the MRRs in the CDC have the same FSR, the reliability is the same for a wider wavelength spacing. Once one of the wavelength channels is aligned, the rest of the wavelength channels are aligned together.

Fig. R2 a, b Simulated (a) transmission and (b) group delay responses for the CD compensation of 20 km SMF for MRRs with a 50-GHz FSR.

As the required bandwidth is still 40 GHz, there is no higher requirement on the bandwidth of the transmission system. For the MRM, the laser wavelength should be precisely aligned with the operation wavelength of the MRM, because the MRM is only working for a narrow bandwidth. Recently, Ayarlabs demonstrated a fully-integrated optical I/O system that utilizes multiples of MRMs in the WDM system to transmit terabit-level data rate¹, which verifies the feasibility of MRMs for WDM transmission.

Based on the discussion above, we would like to add the statements to Supplementary Note 8.

Revision:

“Supplementary Note 8: Scalability of the transmission system

In the parallel data transmission experiment, the operation bandwidth of our CDC is tuned to 50 GHz, while the transmitted data rate per channel is limited to 80 Gbps for the PAM4 modulation format, which is due to the limited bandwidth of the intensity modulator (~32 GHz 3-dB bandwidth). We measured eye diagrams of a 45 Gbaud PAM4 signal, and they were blurred even under the BtB condition using a CW laser as the light source. Thus, to scale the transmission system to achieve a total bit rate exceeding 10 Tbit/s, we prefer to design the microcomb and the CDC with a narrower spacing of 50 GHz.

As illustrated in the discussion in the manuscript, the number of the available comb lines is 51 for our existing microcomb, and it can be doubled if the microcomb has an FSR of 50 GHz. To achieve a total bit rate of more than ten terabits, each channel should at least transmit a data rate of 100 Gbit/s. Compared with the PAM4, the modulation format of DMT is preferred as it has a higher spectral efficiency. In our experiment, the utilized bandwidth of the 112 Gbit/s DMT signal is around 40 GHz, indicating that the operation bandwidth of the CDC should be at least 40 GHz. We note that this operation bandwidth can be achieved with 8 cascaded MRRs. Supplementary Fig. 14 shows the simulated transmission and group delay responses of the 8 MRRs for the CD compensation of 20 km SMF. A waveguide loss of 4 dB/cm is assumed. Therefore, we can achieve a total data rate of 11.424 Tbit/s by employing 102 comb lines.

Supplementary Fig. 14. Transmission and group delay characteristics of the MRRs with a 50-GHz FSR. a, b Simulated (a) transmission and (b) group delay responses for the CD compensation of 20 km SMF for MRRs with a 50-GHz FSR.”

Reference:

1. Wade M *et al.*, A bandwidth-dense, low power electronic-photonic platform and architecture for multi-Tbps optical I/O. *European Conference on Optical Communication (ECOC)*, IEEE (2018).

Comment #1-4

In line 287, the authors used the CEI-112G standard to justify employing DMT modulation; however, this standard is based on PAM4. Please revise it.

Response:

We thank the reviewer for the comment. The authors agree with the reviewer that CEI-112G is a standard developed for PAM4 in the industry. Here, we would like to revise the statement of the DMT modulation in the manuscript as follows:

Revision:

In the Results of the manuscript (from lines 288 to 291):

“DMT is an IM-DD modulation technique that enables higher transmission rates to be achieved. In addition, DMT can also be used for frequency domain analysis and characterization of CD performance. Therefore, we further carried out high-speed data transmission using DMT modulation format.”

Minor suggestions:

Comment #1-5

I suggest replacing ref. 4 with 10.1109/LPT.2021.3124196, as it is an updated version of the ref. 4.

Response:

We thank the reviewer for the comment. We have updated the previous Ref 4 as the new Ref 4 in the manuscript.

Revision:

References

4. Berikaa E, *et al.* TFLN MZMs and Next-Gen DACs: Enabling Beyond 400 Gbps IMDD O-Band and C-Band Transmission. *IEEE Photonics Technology Letters* **35**, 850-853 (2023).”

Comment #1-6

Also, I suggest adding ref. 10.1109/LPT.2023.3285881 to the manuscript, as these are the most recent IMDD results to the reviewer's best knowledge.

Response:

We thank the reviewer for the comment. We have added the reference as Ref 5 in the manuscript.

Revision:

References

5. Alam MS, Li X, Jacques M, Berikaa E, Koh PC, Plant DV. Net 300 Gbps/λ Transmission Over 2 km of SMF With a Silicon Photonic Mach-Zehnder Modulator. *IEEE Photonics Technology Letters* **33**, 1391-1394 (2021).”

Comment #1-7

In Fig. 2 (i-j), I prefer using CDC for chromatic dispersion compensation instead of just DC (Just for clarity).

Response:

We thank the reviewer for the comment. We have updated the Fig. 2i and 2j and the corresponding caption.

Revision:

Caption of the Fig. 2i and 2j (Line 209):

“CDC: chromatic dispersion compensator”

Reviewer 2

>>The manuscript “Parallel Wavelength-Division-Multiplexed Signal Transmission and Dispersion Compensation Enabled by Soliton Microcombs and Microrings” by Y. Liu et al. presents a study on chromatic dispersion compensation over inter-DC (i.e. DCI) relevant distances using a microring-based dispersion compensator, leveraging the link between frequency comb line spacing and dispersion compensation band spacing to enable parallel dispersion compensation.

The work is very well done on a technical level, and so I have attempted to try to understand the impact of this work. I am not sure that the potential for impact is well presented by the authors in the current manuscript.

We thank the reviewer’s insightful evaluation of our work and appreciate his/her efforts in providing us with very helpful comments in the following sections.

>>I think the impactful parts of this study are to show that aligned dispersion compensation between a microcomb-supported set of signals and the dispersion compensator can be made to work well over multiple channels, and that the power consumption of this stage can be low. I think both need to be well described and shown in the manuscript for impact.

Given that these are the important aspects, the major gap in this publication is a proper comparison of energy consumption of the presented system and the (ballpark) energy consumption of coherent DCI links. I’ll detail this further below.

We thank the reviewer for pointing out the significant importance of power consumption in our work. We agree with the reviewer’s suggestions and we have made a detailed discussion and a wholistic comparison of the power consumption of various chromatic dispersion (CD) compensation approaches in the following responses.

>>There is little novelty in the use of a microcomb to support an IM/DD system, it is an open question as to whether IM/DD is actually a compelling option for inter-DC (i.e. DCI) applications, against the coherent 400G ZR (and soon, 800G ZR) products are available now.

As such, for this work to have the potential for impact matching expectations for Nature Communications, I think the power consumption issue needs to be a defensible focus of this work. Otherwise, all this shows is a relatively straightforward proof-of-concept demonstration of a concatenation of known technologies, which I would suggest is not at the level of impact desired for Nature Communications.

I’ve put down some specific points (moving in order of where points appear in the manuscript) to help the authors revise their manuscript to address these concerns.

Comment #2-1

The authors state, in the abstract, that dispersion compensation is of high power consumption or lack reconfigurability. These are the points that need to be quantified, for a quantitative analysis to really prove the point that the authors have solved this issue. How high is the power consumption of current methods? How much reconfigurability is needed for inter-DC (DCI) links?

Response:

We thank the reviewer for the comment and the kind remind. This is a rather important point. We quantify the power consumption and the reconfigurability as follows:

For the power consumption issue, we focus on the transmission distance of up to 120 km for the inter-DCIs. Generally, the CD of single-mode fibers (SMFs) can be compensated through several methods, including the dispersion compensating fibers (DCFs) and the digital signal processing (DSP)-based equalization.

(i) DCFs. As DCFs are passive components and do not require any power for CD tuning, the power consumption of using DCFs for CD compensation mainly comes from the power amplifiers (typically EDFAs) that are used to compensate for the loss induced by the DCFs. The attenuation of the DCF by Thorlabs is less than 0.265 dB/km, and the dispersion ranges from -49 ps/(nm·km) to -30 ps/(nm·km)¹. We assume that the dispersion of SMF and DCF is 18 ps/(nm·km) and -36 ps/(nm·km), respectively, which means that a maximum of 60-km-long DCF is required for the compensation of CD of up to 120 km SMF. In this case, the DCF introduces a maximum insertion loss of 15.9 dB. We assume the 60 km DCF is followed by a 120 km SMF. The length of 120 km is consistent with the reach of the ZR coherent transceiver modules. We consider the output of the SMF to be connected with a receiver, and we assume a -10 dBm received optical power at the receiver end (In Ref 2, the author mentioned an around -10 dBm received optical power can achieve a BER below 1e-3). If we consider a 0.2 dB/km attenuation of the SMF, we can obtain the optical power at the input of the SMF to be $-10 + 0.2 \times 120 = 14 \text{ dBm}$ ($\sim 25 \text{ mW}$). After the introduction of DCF, the optical power was reduced to $P_{DCF} = 14 - 15.9 = -1.9 \text{ dBm}$ ($\sim 0.65 \text{ mW}$). We use the EDFA to compensate for the optical power from $P_{DCF} = -1.9 \text{ dBm}$ back to $P_{out} = 14 \text{ dBm}$. Then the power consumption induced by the EDFA can be calculated by:

$$PC_{EDFA} = (P_{out} - P_{DCF})/\eta_{wp} \quad (1)$$

where η_{wp} is the wall-plug efficiency of the EDFA. A 16% wall-plug efficiency is assumed here (We will discuss this efficiency in detail in part (vi) of the response to comment #2-2). In this case, the power consumption of the EDFA is $\sim 152 \text{ mW}$. The power consumption of using DCF increases exponentially with the length of SMF. To be mentioned, although the DCF can also be used for parallel CD compensation due to their broad optical bandwidth, the lack of reconfigurability makes it impractical for inter-DCI applications.

(ii) DSP equalization. The DSP equalization is commonly used in coherent transmission systems to compensate for the CD. Various companies have demonstrated the optical transceiver modules for 400G ZR applications (<120 km). Inphi⁶ (now Marvell) demonstrated a 400G ZR module with a power consumption of 16 W/400G, and NeoPhotonics⁷ (now Lumentum) demonstrated a 400G ZR module that consumes 18 W

power in total. In addition, InnoLight⁸ demonstrated a 400G ZR module with a total power consumption of 18 W. As the companies generally provide a total power consumption of the entire module, we need to first consider a power breakdown of the power consumption to get the approximate power consumed for the CD compensation. We assume that the DSP consumes about 50% power of the module⁶ and the CD compensation accounts for ~25% of the DSP power consumption⁹. In this case, the normalized power consumption (normalized to every 100 Gbit/s data transmission) of the modules in Ref 6, 7, and 8 is 0.5 W/100G, 0.5625 W/100G, and 0.5625 W/100G, respectively, corresponding to a power consumption of 5 pJ/bit, 5.625 pJ/bit, and 5.625 pJ/bit.

For the reconfigurability issue, we mainly focus on the IMDD transmission system. In Ref 10, the authors mentioned that from the latency perspective, the maximum distance between two data centers should be restricted to less than 120 km to ensure an identical customer experience. In addition, they provided a distribution of fiber distances in Microsoft's network for DCI applications, in which they illustrated that the typical distance between two data centers is within 80 km and the majority (~83%) of the DCIs are within 40 km.

We would like to mention the power consumption of CD compensation of 0.5 W/100G for the Marvell 400G ZR coherent module as a brief introduction to the power consumption of the module. And we put a detailed discussion about power consumption in the Discussion of the manuscript and in the Supplementary Information (Supplementary Note 7). Please see the response to the comment #2-2.

Revision:

For the issue of reconfigurability, in the Introduction of the manuscript (from lines 57 to 60):

"As the typical distance between two data centers is within 80 km and the majority (~83%) of the DCIs are within 40 km⁹, we mainly focus on the short-reach inter-DCIs with a fiber length of up to 40 km and the utilization of IM-DD scheme for this scenario due to the inherent advantages IM-DD offers, including cost-effectiveness, low power consumption, and a compact physical footprint."

For the issue of power consumption of the coherent transceiver modules, in the Introduction of the manuscript (from lines 73 to 77):

"However, the DSP-based coherent transceiver module suffers from a relatively high energy consumption of about 0.5 W/100G (5 pJ/bit) for 120-km-SMF CD compensation²⁵. Dispersion-compensating fibers (DCFs) are also commonly used to compensate for the CD of SMF. Nevertheless, using DCFs is burdened with a relatively high associated cost and low reconfigurability for flexible CD compensation of different fiber lengths."

References

1. <https://www.thorlabs.com/catalogpages/obsolete/2017/DCF38.pdf>

2. Zhou X, Urata R, Liu H. Beyond 1 Tb/s intra-data center interconnect technology: IM-DD or coherent? *Journal of Lightwave Technology*, **38**: 475-484 (2019).
3. <https://www.teraxion.com/en/products/optical-communications/slope-matched-tunable-dispersion-compensator/>
4. <https://www.fs.com/products/72427.html>
5. <https://www.optical-sintai.com/products/tunable-dispersion-compensation-card.html>
6. Nagarajan R, Lyubomirsky I, and Agazzi O, Low power DSP-based transceivers for data center optical fiber communications (invited tutorial). *Journal of Lightwave Technology*, **39**, 5221-5231 (2021).
7. <https://www.lumentum.com/en/products/400g-zr-zr-qsfp-dd-dco>
8. <https://www.innolight.com/en/goods/solution/cid/16.html>
9. Seiler P M, *et al.* Toward coherent O-band data center interconnects. *Front. Optoelectron.*, **14**, 414-425 (2021).
10. Nagarajan R, *et al.* Silicon photonics-based 100 Gbit/s, PAM4, DWDM data center interconnects. *J. Opt. Commun. Netw.*, **10**, B25-B36 (2018).

Comment #2-2

Again, in the abstract, the authors state a power consumption of below 160mW. This is then an important point to really analyze in the main manuscript. It also needs to be a wholistic comparison – what bounds do the authors put on their power consumption analysis, to make sure that there is a fair comparison to other dispersion compensation techniques?

Response:

We thank the reviewer very much for the comment. We appreciate the reviewer's efforts in pointing out the importance of power consumption of the CD compensation and discussing how to analyze it and make a fair comparison in comments #2-2, #2-14b, #2-17, and #2-18. We agree with the reviewer that power consumption is a very important point and it should be fairly discussed to show the performance of our CD compensator (CDC). In the following parts, we talk about the power consumption of different methods first, and then we make a wholistic comparison and discussion about the normalized power consumption.

(i) The power consumption of our CDC. As our CDC brings in an extra loss that should be compensated by an EDFA, we also consider the power consumption of the EDFA. For the parallel data transmission, we assume a uniform transmission loss of 18 dB of different channels for 20 km SMF transmission (see Supplementary Fig. 8i. A ~10 dB insertion loss is obtained at the center of the operation bandwidth. A ~8-dB fiber coupling loss from the grating couplers is also added onto it). In the experiment, the total optical power of the 15 comb lines before our CDC was maintained at 17.5 dBm (~56.23 mW). After the introduction of our CDC, the total optical power decreased to 17.5 – 18 = –0.5 dBm (~0.89 mW). We used an EDFA to compensate the total optical power back to 17.5 dBm after the CDC. The input optical power P_{in} and output optical power P_{out} of the

EDFA is -0.5 dBm and 17.5 dBm , respectively. The power consumption of the EDFA PC_{EDFA} can then be calculated by

$$PC_{EDFA} = (P_{out} - P_{in})/\eta_{wp} \quad (2)$$

where η_{wp} is the wall-plug efficiency of the EDFA. We estimate the wall-plug efficiency to be around 16% (We will discuss this later in part (vi)). The PC_{EDFA} can then be estimated to be $\sim 346 \text{ mW}$, leading to a total power consumption of less than 506 mW for our CDC. The power consumption of our CDC per 100 Gbit/s is $\sim 0.0301 \text{ W/100G}$ ($\sim 0.301 \text{ pJ/bit}$).

(ii) The power consumption of CD compensation for existing coherent optical transceiver modules. Inphi¹ (now Marvell) demonstrated a 400G ZR transceiver module with a power consumption of 4 W/100G . NeoPhotonics² (now Lumentum) and InnoLight³ demonstrated 400G ZR modules that consume 18 W power for 400G transmission, respectively. The transmission length is less than 120 km and these power consumptions are generally for the entire module. To get the power consumption of the CD compensation, we assume that the DSP consumes about 50% of the module¹ and the CD compensation accounts for $\sim 25\%$ of the DSP power⁴. In this case, the normalized power consumptions of CD compensation of these three modules are 0.5 W/100G , 0.5625 W/100G , and 0.5625 W/100G , corresponding to a power consumption of 5 pJ/bit , 5.625 pJ/bit , and 5.625 pJ/bit .

(iii) The power consumption of CD compensation for next-generation coherent transceiver modules. The maximum total power consumption allowed for 800G ZR using Quad Small Form Factor Pluggable-Double Density (QSFP-DD) and Octal Small Form Factor Pluggable (OSFP) form factors is 25 W ⁵ and 30 W ⁶, respectively. Maximum total power consumption of 33 W is allowed for 1.6T ZR using the OSFP-Extra Dense (OSFP-XD) form factor⁶. All these modules are designed for transmission less than 120 km . Using a similar calculation method, we can obtain that the power consumption per 100 Gbit/s of CD compensation for 800G ZR coherent using QSDP-DD and OSFP is $\sim 0.39 \text{ W/100G}$ ($\sim 3.9 \text{ pJ/bit}$) and $\sim 0.47 \text{ W/100G}$ ($\sim 4.7 \text{ pJ/bit}$), respectively. The power consumption of CD compensation for 1.6T ZR coherent is $\sim 0.26 \text{ W/100G}$ ($\sim 2.6 \text{ pJ/bit}$).

(v) The normalized power consumption of CD compensation of various methods. As the mentioned CD compensation methods have different transmission lengths and the power consumption is larger for longer SMF transmission, we thus normalize the length of SMF to 40 km for a fair comparison. **(I)** For our own CDC, the increase in power consumption mainly originates from the power consumption of the EDFA, as the power consumption of our chip remains nearly unchanged for CD compensation of different lengths of SMFs in our work. There is an extra loss of $\sim 5 \text{ dB}$ between the CD compensation of 40 km and 20 km SMF (see Fig. 2d in the manuscript, the loss is obtained at the center of the operation bandwidth), and the total power consumption of EDFA is calculated by

$$\frac{10^{\frac{17.5}{10}} - 10^{\frac{17.5-18-5}{10}}}{0.16} = \sim 350 \text{ mW.}$$

The total power consumption is then $\sim 510 \text{ mW}$, leading to a normalized power consumption of $\sim 0.0304 \text{ W/100G}$ ($\sim 0.304 \text{ pJ/bit}$, 1.68 Tbit/s data transmitted). **(II)** For the coherent transceiver modules, we assume that the power consumption of CD compensation varies almost linearly with the SMF length of less than 120 km according to the data in Fig. 12 in Ref 1. Therefore, the power consumption for CD

compensation of 40 km SMF is one-third of that for 120 km transmission, resulting in normalized power consumption of ~0.17 W/100G (~1.7 pJ/bit), ~0.19 W/100G (~1.9 pJ/bit), and ~0.19 W/100G (~1.9 pJ/bit) for the Inphi, NeoPhotonics, and InnoLight transceiver module, respectively. The normalized power consumption is ~0.13 W/100G (~1.3 pJ/bit, 800G ZR QSFP-DD), ~0.16 W/100G (~1.6 pJ/bit, 800G ZR OSFP), and ~0.086 W/100G (~0.86 pJ/bit, 1.6T ZR OSFP-XD) for the next generation DCI applications.

(vi) Wall-plug efficiency. A power efficiency of more than 80% from the pump laser diodes to the EDFA output has been demonstrated decades ago⁷. In addition, many laser diodes pumped at 1480 nm can achieve a power conversion efficiency of more than 20%, like the one from SemiNex⁸. Therefore, we conservatively estimate that the wall-plug efficiency of EDFA is 16% (80%×20%), and the power consumption of EDFA only includes the power consumption of the pump laser diodes.

We list the performances and the normalized power consumptions of different CD compensation methods in Table R1 in this response. We can see that the normalized power consumption of our CDC is about 6 times less than that of the existing 400G ZR coherent transceiver modules, and is even about 3 times less than that of the next generation 1.6T ZR coherent modules. Based on the analysis, we can see that our work is remarkably competitive with the existing or even next-generation commercial modules in terms of power consumption.

Table R1. Performance comparison of various CD compensation methods

Method/ Company	CBG ⁹	MRR ¹⁰	MZI ¹¹	Marvell ¹	Lumentum ² InnoLight ³	Next generation coherent ^{5, 6}			This work
Approach	CBG ^a	MRRs	MZIs	DSP					MRRs
Platform/Node	SOI	Si ₃ N ₄		7 nm		5 nm	3 or 2 nm		SOI
Insertion loss (dB)	~6	0.8	20	/					10
CD (ps/nm)	480	-170	-500	Max. -2400					-355 (Max. -720)
FSR (GHz)	/	100	100	/					99.5
Bandwidth (GHz)	~110	50	15	/					50
SMF length (km)	28	10	40	Max. 120					20 (Max. 40)
Max. data line rate (Gbit/s)	60	240	53.125	400		800	1600		1680
Power consumption ^b (pJ/bit)	/			~1.7	~1.9	~1.3	~1.6	~0.86	~0.304
Footprint ^c	5 mm	2.2 × 0.56 mm ²	9.89 × 22.5 mm ²	89.4 × 18.35 × 8.5 mm ³			100.4 × 22.58 × 13.0 mm ³	/	2 × 1 mm ²

^aCBG stands for chirped Bragg grating.

^bThis power consumption has been normalized to the CD compensation of 40 km SMF.

^cFor the on-chip optical CDCs, the footprint refers to one of the chips. For the commercial modules, the footprint is the one for

the entire module.

Fig. R1. a, b Calculated relationship between the power consumption for CD compensation and **(a)** the SMF length and **(b)** the total data rate (compensation for 40 km SMF) of our CDC and the existing 400G ZR coherent DSP module, respectively. CL: coupling loss; IL: insertion loss.

We further show how the power consumption increases with the length of SMF in Fig. R1a in this response. For the CD compensation of 40 km SMF, we consider the power consumption to be consistent with the previous analysis, in which 160 mW is from our CDC, and 350 mW is from EDFA. An extra 7 MRRs should be introduced for the CD compensation of an additional 40 km SMF, leading to a 140 mW power consumption increment from the CDC, assuming that each MRR consumes an average power consumption of 20 mW. Thus, the power consumption of CDC for the CD compensation of 40 km, 80 km, and 120 km SMF is 160 mW, 300 mW, and 440 mW, respectively. In addition, we assume a 17.5 dBm total optical input power of our CDC, and a 10-dB extra loss (due to the additional MRRs, obtained from measurement results in Fig. 2d in the manuscript) is introduced for every extra 40 km SMF. We consider using an additional EDFA to compensate for the loss for every extra 40 km SMF transmission, similar to the schematic shown in Fig. 1 in Ref 12. Then the power consumption of EDFA for CD compensation of 40 km, 80 km, and 120 km SMF is 350 mW, 666 mW, and 982 mW, respectively. In this case, the power consumption of our CDC is ~ 0.304 pJ/bit, ~ 0.575 pJ/bit, and ~ 0.846 pJ/bit. For the existing 400G ZR coherent, we also assume the power consumption of CD compensation varies almost linearly with the length of SMF, and the corresponding power consumption of 40 km, 80 km, and 120 km CD compensation is ~ 1.7 pJ/bit, ~ 3.3 pJ/bit, and 5 pJ/bit, respectively. We can see that the power consumption of our CDC is around 6 times less and increases much slower than the coherent DSP with an increased SMF length.

The relationship between the power consumption and the total data rate is shown in Fig. R1b in this response. We consider a total data rate of 400 Gbit/s, 800 Gbit/s, 1.6 Tbit/s, 3.2 Tbit/s, 6.4 Tbit/s, 11.424 Tbit/s, and 12.8 Tbit/s. The total input optical power in our work is 17.5 dBm, and we used 15 comb lines, resulting in an average optical power of ~ 5.74 dBm (~ 3.75 mW). The number of the used comb lines is 4, 8, 15, 29, 58, 102, and 115 for 400 Gbit/s, 800 Gbit/s, 1.6 Tbit/s, 3.2 Tbit/s, 6.4 Tbit/s, 11.424 Tbit/s, and 12.8 Tbit/s data rate, leading to a total optical input power of ~ 11.76 dBm, ~ 14.77 dBm, 17.50 dBm, ~ 20.36 dBm, ~ 23.37 dBm, ~ 25.83 dBm, and ~ 26.35 dBm, respectively. We also consider an 18 dB loss

of our CDC (20 km) and consider an extra 5 dB loss of CD compensation of 40 km SMF compared to the one of 20 km. The power consumption of our chip is fixed at 160 mW. We take the calculation of the power consumption for 400 Gbit/s data rate as an example. The

power consumption can be calculated by $\frac{10^{\frac{11.76}{10}} - 10^{\frac{11.76-18-5}{10}}}{0.16} + 160 = \sim 253$ mW. Using a

similar method, the total power consumption of our CDC under the seven scenarios is 0.253 W, 0.347 W, 0.510 W, 0.836 W, 1.512 W, 2.538 W, and 2.841 W, respectively, while the one of coherent DSP is 0.68 W, 1.36 W, 2.72 W, 5.44 W, 10.88 W, ~ 19.42 W, and 21.76 W, respectively. The normalized power consumption of the coherent modules maintains at 1.7 pJ/bit, while it is ~ 0.633 pJ/bit, ~ 0.434 pJ/bit, ~ 0.304 pJ/bit, ~ 0.257 pJ/bit, ~ 0.233 pJ/bit, ~ 0.222 pJ/bit, and ~ 0.221 pJ/bit for our CDC. Our CDC exhibits a much smaller total power consumption when the total transmitted data rate is getting larger, and the normalized power consumption is even lower for higher data rates.

Fig. R2. Transmission and group delay characteristics of the CD compensation with improved propagation loss. a, b Simulated (a) transmission and (b) group delay response of the CD compensation of 40 km SMF, assuming a 0.5 dB/cm propagation loss of the MRR.

Furthermore, we consider an optimized coupling loss of our CDC and a low insertion loss of our chip, and then further analyze the power consumption. The authors in Ref 13 experimentally demonstrated a coupling loss of ~ 0.32 dB/facet in 2015, which is relatively the smallest one for coupling. The total loss of our CDC for the CD compensation of 20 km SMF is then 10.64 dB (10 dB insertion loss and 0.64 dB coupling loss). We calculate the power consumption using a similar method. The power consumptions of CD compensation for various SMF lengths and various data rates are listed in Table R2 and shown in Fig. R1 in this response. In addition, we assume a low waveguide loss of 0.5 dB/cm and a ~ 0.1 dB insertion loss for each MMI. Figs. R2a and R2b show the simulated transmission and group delay response of the CD compensation for 40 km SMF. We can obtain an insertion loss of ~ 2.2 dB from the center of the operation bandwidth. We then consider an insertion loss of ~ 2.2 dB and a coupling loss of ~ 0.64 dB. An extra insertion loss of 2.2 dB is considered for an additional 40 km SMF. Extra 7 MRRs are also considered for the CD compensation of an additional 40 km SMF.

We obtain the normalized power consumption for the CD compensation of various SMF lengths and various data rates, as shown in Table R2 and Fig. R1 in this response. We can see that the normalized power consumptions of low coupling loss (CL) are almost identical

to the ones in our experiment. There is a power consumption reduction of 0.108 pJ/bit, 0.213 pJ/bit, and 0.317 pJ/bit for 40 km, 80 km, and 120 km SMF transmissions if low CL and low insertion loss (IL) are assumed, respectively, corresponding to a reduction of 35.53%, 37.04%, and 37.47%. The power consumption of our CDC for higher data rate is even lower, and under the 11.424 Tbit/s data transmission rate, it is around 15 times less than that of the coherent DSP if low CL and low IL are assumed.

Table R2. Calculated power consumption of various scenarios

Scenarios		Coherent DSP (pJ/bit)	MRR-based CDC (pJ/bit)	MRR-based CDC with low CL (pJ/bit)	MRR-based CDC with low CL and IL (pJ/bit)
Various SMF lengths (km)	40	1.7	0.304	0.299	0.196
	80	3.3	0.575	0.570	0.362
	120	5	0.846	0.841	0.529
Various data rate (Gbit/s)	400	1.7	0.633	0.628	0.513
	800		0.434	0.429	0.313
	1600		0.258	0.299	0.196
	3200		0.257	0.253	0.150
	6400		0.233	0.228	0.125
	11424		0.222	0.218	0.114
	12800		0.221	0.216	0.113

Based on the analysis above, we would like to add the statement of power consumption in the Abstract, Introduction, and Conclusion of the manuscript. We add the detailed discussion of the power consumption into the Discussion part of our manuscript and the Supplementary Note 7:

Revision:

In the Abstract of the manuscript (from lines 30 to 32):

“Our method features a remarkably low energy consumption of ~0.304 pJ/bit for CD compensation of 40-km-SMFs, which is around 6 times less than that of the commercially available 400G ZR coherent transceiver modules.”

In the Abstract of the manuscript (from lines 34 to 36):

“Our approach holds significant promise for achieving data communications at a scale exceeding 10 terabits, and ~0.114 pJ/bit for CD compensation, making it highly valuable for future hyper-scale DCIs.”

In the Introduction of the manuscript (from lines 113 to 115):

“Importantly, we assessed the power consumption for the CD compensation to be as low as 0.304 pJ/bit for 40-km SMFs. This remarkably low power consumption is around six times less than that of the commercially available 400G ZR coherent transceiver modules.”

In the Introduction of the manuscript (from lines 117 to 119):

“As a result, the power consumption can even be reduced to 0.114 pJ/bit for the 40-km SMF transmission with an optimized coupling loss and insertion loss of our CDC.”

In the Discussion part of the manuscript (from lines 342 to 386):

“Since the power consumption is of vital importance in the inter-DCI applications, we calculate the power consumption of CD compensation for our CDC and other commercial modules and make a fair comparison by normalizing the calculated ones to pJ/bit for every 40 km SMF transmission. To be noted, the power consumption of our parallel scheme includes both the tuning power of the MRRs-based CDC and the power consumption of the EDFA used for loss compensation of the CDC. See Supplementary Note 7 for a detailed calculation of the power consumption. Table 1 gives a comparison of the performance of various CD compensation methods, including the on-chip optical CDCs, the existing commercially available 400G ZR coherent transceiver modules, and the next-generation coherent modules. We can see that our CDC outperforms the other integrated optical CDCs in terms of reconfigurability, and the supported total data capacity. Besides, our optical CDC outperforms electronic CD compensation approaches based on DSP algorithms, especially in power efficiency, which necessitates individual processing for each wavelength channel. As seen from Table 1, the normalized power consumption of CD compensation for our CDC is ~0.304 pJ/bit, which is about 6 times, 4 to 5 times, and even 3 times less than that of the existing 400G ZR coherent modules, the next-generation 800G ZR coherent modules, and the next generation 1.6T ZR coherent modules, respectively, due to the parallel CD compensation capabilities of our optical CDC.

The power consumption for CD compensation of our parallel compensation scheme can be further improved by reducing the loss of our CDC chip and also using more wavelength channels. The coupling loss in our work is ~4 dB per grating coupler. It is plausible to minimize the fiber-to-fiber loss associated with the CDC to ~0.32 dB/facet by introducing low-loss and broadband edge coupler⁷⁰. Additionally, MRRs made of widened waveguides can be incorporated to mitigate on-chip losses. Fig. 5a shows the normalized power consumption of our CDC and the Marvell 400G ZR coherent transceiver with various SMF lengths from 40 km to 120 km. We also calculate the power consumption assuming an optimized coupling loss (CL, ~0.64 dB in total) and an improved insertion loss (IL, 0.5 dB/cm waveguide loss, and ~0.1 dB for each MMI) of our CDC. Compared with the 400G coherent transceiver, the power consumption of our CDC increases much slower with the SMF length. For a 120-km-SMF transmission, the power consumption of our CD compensation scheme is less than 0.846 pJ/bit. By considering an optimized coupling loss and insertion loss of our CDC, we can further get a lower power consumption of 0.529 pJ/bit, which is almost 1/10 of that of the DSP power consumption. Fig. 5b depicts the power consumption with different total supported data rates from 400 Gbit/s to 12.8 Tbit/s over 40 km SMF transmission. Due to the parallel compensation, the power consumption of our CDC is even lower for higher data rates. It is approximately 15 times less than that

of the DSP when optimized CL and IL are considered at a total data rate of 11.424 Tbit/s (112 Gbit/s×102). See Supplementary Note 7 for a detailed calculation.

Table 1. Performance comparison of various CD compensation methods

Method/ Company	CBG ⁷¹	MRR ³⁹	MZI ³⁴	Marvell ²⁵	Lumentum ²⁶ InnoLight ²⁷	Next generation coherent ^{72, 73}	This work
Approach	CBG ^a	MRRs	MZIs	DSP			MRRs
Platform/Node	SOI	Si ₃ N ₄		7 nm		5 nm	3 or 2 nm SOI
Insertion loss (dB)	~6	0.8	20	/			10
CD (ps/nm)	480	-170	-500	Max. -2400			-355 (Max. -720)
FSR (GHz)	/	100	100	/			99.5
Bandwidth (GHz)	~110	50	15	/			50
SMF length (km)	28	10	40	Max. 120			20 (Max. 40)
Max. data line rate (Gbit/s)	60	240	53.125	400		800	1600 1680
Power consumption ^b (pJ/bit)	/			~1.7	~1.9	~1.3	~1.6 ~0.86 ~0.304
Footprint ^c	5 mm	2.2 × 0.56 mm ²	9.89 × 22.5 mm ²	89.4 × 18.35 × 8.5 mm ³		100.4 × 22.58 × 13.0 mm ³	/ 2 × 1 mm ²

^aCBG stands for chirped Bragg grating.

^bThis power consumption has been normalized to the CD compensation of 40 km SMF.

^cFor the on-chip optical CDCs, the footprint refers to the one of the chips. For the commercial modules, the footprint is the one for the entire module.

Fig. 5. Power consumption with the SMF length and total supported data rate. a, b Calculated power consumption for CD compensation of (a) the SMF length and (b) the total data rate (compensation for 40 km SMF) of our CDC and the existing Marvell 400G ZR coherent transceiver module, respectively. CDC: chromatic dispersion compensator; CL: coupling loss; IL: insertion loss."

In the Conclusion of the manuscript (from line 418 to line 423):

“The power consumption of the CD compensation of our CDC is ~0.304 pJ/bit for 40-km-SMFs, which is ~1/6 and ~1/3 of that of the existing commercially available 400G ZR coherent transceiver modules and the next generation 1.6T ZR coherent transceiver modules, respectively. Particularly, when the optimized CL and IL are considered, the power consumption of our CDC can be further reduced to ~0.114 pJ/bit, which is ~15 times less than that of the 400G ZR coherent transceiver modules for a total data rate of 11.424 Tbit/s.”

At the end of Results of the manuscript, we delete:

“The maximum power consumption of our dispersion compensator is as low as 160 mW. See Supplementary Note 6 for the detailed calculation of the bit rate and Supplementary Note 3 for the power consumption.”

In the Supplementary Note 7:

“Supplementary Note 7: Power consumption calculation of CD compensation

In the following parts, we talk about the power consumptions of different CD compensation methods first (original means power consumptions without normalization), and then we make a wholistic comparison and discussion about the normalized power consumptions with the unit of pJ/bit for CDC of 40-km SMFs.

(i) The power consumption for DCFs. As DCFs are passive components and do not require any power for CD tuning, the power consumption of using DCFs for CD compensation mainly comes from the power amplifiers (typically EDFAs) that are used to compensate for the loss induced by the DCFs. The attenuation of the DCF by Thorlabs⁵ is less than 0.265 dB/km, and the dispersion ranges from -49 ps/(nm·km) to -30 ps/(nm·km). We assume that the dispersion of SMF and DCF is 18 ps/(nm·km) and -36 ps/(nm·km), respectively, which means that a maximum of 60-km-long DCF is required for the compensation of CD of up to 120 km SMF. In this case, the DCF introduces a maximum insertion loss of 15.9 dB. We assume the 60 km DCF is followed by a 120 km SMF. We consider the output of the SMF to be connected with a receiver, and we assume a -10 dBm received optical power at the receiver end (In Ref 6, the author mentioned an around -10 dBm received optical power can achieve a BER below 1e-3). If we consider a 0.2 dB/km attenuation of the SMF, we can obtain the optical power at the input of the SMF to be $-10 + 0.2 \times 120 = 14 \text{ dBm}$ (~25 mW). After the introduction of DCF, the optical power reduces to $P_{DCF} = 14 - 15.9 = -1.9 \text{ dBm}$ (~0.65 mW). We use the EDFA to compensate the optical power from $P_{DCF} = -1.9 \text{ dBm}$ back to $P_{out} = 14 \text{ dBm}$. Then the power consumption induced by the EDFA can be calculated by:

$$PC_{EDFA} = (P_{out} - P_{DCF})/\eta_{wp} \quad (10)$$

where η_{wp} is the wall-plug efficiency of the EDFA. A 16% wall-plug efficiency is assumed here (We will discuss this efficiency in part (vi)). In this case, the power consumption of the EDFA is ~152 mW. The power consumption of using DCF increases exponentially with the length of SMF. To be mentioned, although the DCF can also be used for parallel CD

compensation due to their broad optical bandwidth, the lack of reconfigurability makes it less favorable for inter-DCI applications.

(ii) The original power consumption of our CDC. The power consumption for controlling the states of CDC is below 160 mW, which can be calculated by referring to the applied voltages shown in Supplementary Table 3. As our CDC brings in an extra loss that should be compensated by an EDFA, we also consider the power consumption of the EDFA. For the parallel data transmission, we assume a uniform transmission loss of 18 dB of different channels for 20 km SMF transmission, including ~10 dB on-chip insertion loss at the center of the operation bandwidth (see Supplementary Fig. 8i) and ~8-dB fiber coupling loss from the grating couplers. In the experiment, the total optical power of the 15 comb lines before our CDC was maintained at 17.5 dBm (~56.23 mW). After the introduction of our CDC, the total optical power decreased to $17.5 - 18 = -0.5$ dBm (~0.89 mW). We used an EDFA to compensate the total optical power back to 17.5 dBm after the CDC. The input optical power P_{in} and output optical power P_{out} of the EDFA is -0.5 dBm and 17.5 dBm, respectively. The power consumption of the EDFA PC_{EDFA} can be then calculated by

$$PC_{EDFA} = (P_{out} - P_{in})/\eta_{wp} \quad (11)$$

The PC_{EDFA} can then be estimated to be ~346 mW, leading to a total power consumption of less than 506 mW for our CDC. Thus, the power consumption of our CDC is ~0.301 pJ/bit (1.68 Tbit/s data transmitted).

(iii) The original power consumption of CD compensation with existing coherent optical transceiver modules. Inphi⁷ (now Marvell) demonstrated a 400G ZR transceiver module with a power consumption of 4 W/100G. NeoPhotonics⁸ (now Lumentum) and InnoLight⁹ demonstrated 400G ZR modules that consume 18 W power for 400G transmission, respectively. The transmission length is less than 120 km and these power consumptions are generally for the entire module. To get the power consumption of the CD compensation, we assume that the DSP consumes about 50% of the module⁷ and the CD compensation accounts for ~25% of the DSP power¹⁰. In this case, the power consumption of the DSP-based CD compensation of Inphi's 400 ZR modules is calculated as $4 \times 50\% \times 25\% = 0.5$ W/100G. Thus, the power consumption calculated in units of pJ/bit of these three modules are 5 pJ/bit, 5.625 pJ/bit, and 5.625 pJ/bit for 120-km-SMFs CDC, respectively.

(iv) The original power consumption of CD compensation with next-generation coherent transceiver modules. The maximum total power consumptions allowed for 800G ZR using Quad Small Form Factor Pluggable-Double Density (QSFP-DD)^{11, 12} and Octal Small Form Factor Pluggable (OSFP)¹² form factors are 25 W and 30 W, respectively. Maximum total power consumption of 33 W is allowed for 1.6T ZR using OSFP-Extra Dense (OSFP-XD) form factor¹². These modules are designed for the SMF transmission of less than 120 km. Using a similar calculation method, we can obtain that the power consumption of CD compensation for 800G ZR coherent using QSDP-DD and OSFP is ~3.9 pJ/bit and ~4.7 pJ/bit, respectively. The power consumption of CD compensation for 1.6T ZR coherent is ~2.6 pJ/bit.

(v) The normalized power consumption of CD compensation of various methods. As the mentioned CD compensation methods have different transmission lengths and the

power consumption is larger for longer SMF transmission, we thus normalize the length of SMF to 40 km for a fair comparison. **(I)** For our own CDC, the increase in power consumption mainly originates from the power consumption of the EDFA, as the power consumption of our chip remains nearly unchanged for compensation of different SMF lengths. There is an extra loss of ~5 dB between the CD compensation of 40 km and 20 km SMF (see Fig. 2d in the manuscript, the loss is obtained at the center of the operation

bandwidth), and the total power consumption of EDFA is calculated by $\frac{10^{\frac{17.5}{10}} - 10^{\frac{17.5-18-5}{10}}}{0.16} =$

~350 mW. The total power consumption is then ~510 mW, leading to a normalized one of ~0.304 pJ/bit. **(II)** For the coherent transceiver modules, we assume that the power consumption of CD compensation varies almost linearly with the SMF length of less than 120 km according to the data in Fig. 12 of Ref 7. Therefore, the power consumption for CD compensation of 40 km SMF is one-third of that for 120 km transmission, resulting in a normalized power consumption of ~1.7 pJ/bit, ~1.9 pJ/bit, and ~1.9 pJ/bit for the Inphi, NeoPhotonics, and InnoLight transceiver modules, respectively. The normalized power consumptions are ~1.3 pJ/bit (800G ZR QSFP-DD), ~1.6 pJ/bit (800G ZR OSFP), and ~0.86 pJ/bit (1.6T ZR OSFP-XD) for the next generation DCI applications.

(vi) Wall-plug efficiency. A power efficiency of more than 80% from the pump laser diodes to the EDFA output has been demonstrated decades ago¹³. In addition, many laser diodes pumped at 1480 nm can achieve a power conversion efficiency of more than 20%, like the one from SemiNex¹⁴. Therefore, we conservatively estimate that the wall-plug efficiency of EDFA is 16% (80%×20%), and the power consumption of EDFA only includes the one of the pump laser diodes.

We list the performances and the normalized power consumptions of different CD compensation methods in Supplementary Table 4. We can see that the normalized power consumption of our CDC is about 6 times less than that of the existing 400G ZR coherent transceiver modules, and is even about 3 times less than that of the next generation 1.6T ZR coherent modules. Based on the analysis, we can see that our work is remarkably competitive with the existing or even next-generation commercial modules in terms of power consumption, and is suitable for the DCI applications where the power consumption is of vital importance.

Supplementary Table 4. Comparison of various CD compensation methods

Methods	This work	Coherent (400G ZR)			Coherent (800G/1.6T ZR, Next Generation)		
Company/ Form factor	/	Marvell ⁷	Lumentum ⁸	InnoLight ⁹	QSFP-DD ^{11, 12}	OSFP ¹²	OSFP-XD ¹²
DSP node	/	7 nm			5 nm	3 or 2 nm	
Transmission range (km)	0~40	<120					
Total data rate (Gbit/s)	1680	400			800	1600	
Power consumption (pJ/bit)	~0.304	~1.7	~1.9	~1.9	1.3	1.6	0.86

We further show how the power consumption increases with the length of SMF in Fig.

5a in the manuscript. For the CD compensation of 40 km SMF, we consider the power consumption to be consistent with the previous analysis, in which 160 mW is from our CDC, and 350 mW is from EDFA. An extra 7 MRRs should be introduced for the CD compensation of an additional 40 km SMF, leading to a 140 mW power consumption increment from the CDC, assuming that each MRR consumes an average power consumption of 20 mW. Thus, the power consumption of CDC for the CD compensation of 40 km, 80 km, and 120 km SMF is 160 mW, 300 mW, and 440 mW, respectively. In addition, we assume a 17.5 dBm total optical input power of our CDC, and a 10-dB extra loss (due to the additional MRRs, obtained from measurement results in Fig. 2d in the manuscript) is introduced for every extra 40 km SMF. We consider using an additional EDFA to compensate for the loss for every extra 40 km SMF transmission, similar to the schematic shown in Fig. 1 in Ref 15. Then the power consumption of EDFA for CD compensation of 40 km, 80 km, and 120 km SMF is 350 mW, 666 mW, and 982 mW, respectively. In this case, the power consumption of our CDC is ~ 0.304 pJ/bit, ~ 0.575 pJ/bit, and ~ 0.846 pJ/bit. For the existing 400G ZR coherent, we also assume the power consumption of CD compensation varies almost linearly with the length of SMF, and the corresponding power consumption of 40 km, 80 km, and 120 km CD compensation is ~ 1.7 pJ/bit, ~ 3.3 pJ/bit, and 5 pJ/bit, respectively. We can see that the power consumption of our CDC is around 6 times less and increases much slower than the coherent DSP with an increased SMF length.

The relationship between the power consumption and the total data rate is shown in Fig. 5b in the manuscript. We consider a total data rate of 400 Gbit/s, 800 Gbit/s, 1.6 Tbit/s, 3.2 Tbit/s, 6.4 Tbit/s, 11.424 Tbit/s, and 12.8 Tbit/s. The total input optical power in our work is 17.5 dBm, and we used 15 comb lines, resulting in an average optical power of ~ 5.74 dBm (~ 3.75 mW). The number of the used comb lines is 4, 8, 15, 29, 58, 102, and 115 for 400 Gbit/s, 800 Gbit/s, 1.6 Tbit/s, 3.2 Tbit/s, 6.4 Tbit/s, 11.424 Tbit/s, and 12.8 Tbit/s data rate, leading to a total optical input power of ~ 11.76 dBm, ~ 14.77 dBm, 17.50 dBm, ~ 20.36 dBm, ~ 23.37 dBm, ~ 25.83 dBm, and ~ 26.35 dBm, respectively. We also consider an 18 dB loss of our CDC (20 km) and consider an extra 5 dB loss of CD compensation of 40 km SMF compared to the one of 20 km. The power consumption of our chip is fixed at 160 mW. We take the calculation of the power consumption for 400 Gbit/s data rate as an example. The

power consumption can be calculated by $\frac{10^{\frac{11.76}{10}} - 10^{\frac{11.76-18-5}{10}}}{0.16} + 160 = \sim 253$ mW. Using a similar method, the total power consumption of our CDC under the seven scenarios is 0.253 W, 0.347 W, 0.510 W, 0.836 W, 1.512 W, 2.538 W, and 2.841 W, respectively, while the one of coherent DSP is 0.68 W, 1.36 W, 2.72 W, 5.44 W, 10.88 W, ~ 19.41 W, and 21.76 W, respectively. The normalized power consumption of the coherent modules maintains at 1.7 pJ/bit, while it is ~ 0.633 pJ/bit, ~ 0.434 pJ/bit, ~ 0.304 pJ/bit, ~ 0.257 pJ/bit, ~ 0.233 pJ/bit, 0.222 pJ/bit, and ~ 0.221 pJ/bit for our CDC. Our CDC exhibits a much smaller total power consumption when the total transmitted data rate is getting larger, and the normalized power consumption is even lower for higher data rates.

Supplementary Fig. 13. Transmission and group delay characteristics of the CD compensation with improved propagation loss. a, b Simulated (a) transmission and (b) group delay response of the CD compensation of 40 km SMF, assuming a 0.5 dB/cm propagation loss of the MRR.

Furthermore, we consider an optimized coupling loss of our CDC and a low insertion loss of our chip (~ 0.5 dB/cm), and then further analyze the power consumption. The authors in Ref 16 experimentally demonstrated a coupling loss of ~ 0.32 dB/facet in 2015, which is relatively the smallest one for coupling. The total loss of our CDC for the CD compensation of 20 km SMF is then 10.64 dB (10 dB insertion loss and 0.64 dB coupling loss). We calculate the power consumption using a similar method. The power consumptions of CD compensation for various SMF lengths and various data rates are listed in Supplementary Table 5 and shown in Fig. 5 in the manuscript. In addition, we assume a low waveguide loss of 0.5 dB/cm and a ~ 0.1 dB insertion loss for each MMI. Supplementary Figs. 13a and 13b show the simulated transmission and group delay response of the CD compensation for 40 km SMF. We can obtain an insertion loss of ~ 2.2 dB from the center of the operation bandwidth. We then consider an insertion loss of ~ 2.2 dB and a coupling loss of ~ 0.64 dB. An extra insertion loss of 2.2 dB is considered for an additional 40 km SMF. Extra 7 MRRs are also considered for the CD compensation of an additional 40 km SMF.

We obtain the normalized power consumption for the CD compensation of various SMF lengths and various data rates, as shown in Supplementary Table 5 and Fig. 5 in the manuscript. We can see that the normalized power consumptions of low coupling loss (CL) are almost identical to the ones in our experiment. There is a power consumption reduction of 0.108 pJ/bit, 0.213 pJ/bit, and 0.317 pJ/bit for 40 km, 80 km, and 120 km SMF transmissions if low CL and low insertion loss (IL) are assumed, respectively, corresponding to a reduction of 35.53%, 37.04%, and 37.47%. The power consumption of our CDC for higher data rate is even lower, and under the 11.424 Tbit/s data transmission rate, it is around 15 times less than that of the coherent DSP if low CL and low IL are assumed.

Supplementary Table 5. Calculated power consumption of various scenarios

Scenarios		Coherent DSP (pJ/bit)	MRR-based CDC (pJ/bit)	MRR-based CDC with low CL (pJ/bit)	MRR-based CDC with low CL and IL (pJ/bit)
Various SMF lengths (km)	40	1.7	0.304	0.299	0.196

	80	3.3	0.575	0.570	0.362
	120	5	0.846	0.841	0.529
Various data rate (Gbit/s)	400	1.7	0.633	0.628	0.513
	800		0.434	0.429	0.313
	1600		0.258	0.299	0.196
	3200		0.257	0.253	0.150
	6400		0.233	0.228	0.125
	11424		0.222	0.218	0.114
	12800		0.221	0.216	0.113

”

References

1. Nagarajan R, Lyubomirsky I, and Agazzi O, Low power DSP-based transceivers for data center optical fiber communications (invited tutorial). *Journal of Lightwave Technology*, **39**, 5221-5231 (2021).
2. <https://www.lumentum.com/en/products/400g-zr-zr-qsf-p-dd-dco>
3. <https://www.innolight.com/en/goods/solution/cid/16.html>
4. Seiler P M, *et al.* Toward coherent O-band data center interconnects. *Front. Optoelectron.*, **14**, 414-425 (2021).
5. <http://www.qsf-p-dd.com/wp-content/uploads/2021/01/2021-QSFP-DD-MSA-Thermal-Whitepaper-Final.pdf>
6. Tauber D *et al.*, Role of coherent systems in the next DCI generation. *Journal of Lightwave Technology*, **41**: 1139-1151 (2023).
7. Massicott J F *et al.*, Efficient, high power, high gain, Er³⁺ doped silica fibre amplifier. *Electronics Letters*, **26**: 1038-1039 (1990).
8. <https://seminex.com/product/chip-laser-diode-8/>
9. Giuntoni I *et al.*, Integrated Dispersion Compensator Based on Apodized SOI Bragg Gratings. *IEEE Photonics Technology Letters* **25**, 1313-1316 (2013).
10. Alam MS, *et al.* 224 Gb/s Transmission over 10 km of SMF at 1550 nm Enabled by a SiN Optical Dispersion Compensator and Stokes Vector Direct Detect Receiver. In: *OSA Advanced Photonics Congress (APC), paper SpM4I.5* (2020).
11. Brodnik GM *et al.*, Extended Reach 40km Transmission of C-Band Real-Time 53.125 Gbps PAM-4 Enabled with a Photonic Integrated Tunable Lattice Filter Dispersion Compensator. In: *Optical Fiber Communications Conference and Exposition (OFC), paper W2A.30* (2018).
12. B. S. G. Pillai *et al.*, Chromatic dispersion compensation – An energy consumption perspective. *OFC/NFOEC*, IEEE (2012).
13. Cheben P *et al.*, Broadband polarization independent nanophotonic coupler for silicon

waveguides with ultra-high efficiency. *Optics Express*, **23**: 22553-22563 (2015).

Comment #2-3

In the abstract, the authors state that they present an aggregate bit rate of 1.68 Tbit/s. They need to be clear at this point that they are presenting a number for the line rate (i.e. without including reduction of data rates due to coding overhead)

Response:

We thank the reviewer for the kind remind. The bit rate of 1.68 Tbit/s is the line rate. We would like to revise our manuscript as follows:

Revision:

In the Abstract of the manuscript (from lines 32 to 34):

“Through our experimental validation, we demonstrate an aggregate line rate of 1.68 Tbit/s over a 20-km-long single-mode fiber using 15 independent wavelength channels spaced at ~100 GHz.”

In the Introduction of the manuscript (from lines 109 to 112):

“By employing pulse-amplitude four-level (PAM4) modulation signals at a rate of 80 Gbit/s, we achieved an aggregate data line rate of 1.2 Tbit/s. Additionally, by employing discrete multi-tone (DMT) modulation signals at a rate of 112 Gbit/s, we achieved an aggregate data line rate of 1.68 Tbit/s.”

In the Results of the manuscript (from lines 278 to 281, and from lines 338 to 339):

“Although a minor FSR difference between the comb lines and the channels of the CDC leads to a slightly higher BER level around the left and right sides of the wavelength span, all the BERs are below the 7% HD-FEC threshold, enabling a transmitted line rate of 1.2 Tbit/s in total (1.12 Tbit/s net rate after FEC overhead subtraction, if the FEC overhead is encoded).”

“In this case, an aggregate data line rate of 1.68 Tbit/s (1.46 Tbit/s net rate, if FEC overhead is encoded) was achieved.”

Comment #2-4

I am unaware of the originality of the authors approach. They state in the abstract that they “present an original scalable on-chip parallel IM/DD data transmission system”. Given that all of the building blocks of their system have been demonstrated separately, or in different combinations, and that the authors do not present a fully chip-based demonstration themselves ... I would not agree that this is “original”, which suggests that the novelty in their work primarily lies there.

Response:

We thank the reviewer for the comment. Indeed, the massive parallel data transmission

using a microcomb and the CD compensation based on single or cascaded MRRs have been separately demonstrated previously. Yet, we first proposed the concept in our work that we can use the integrated MRR-based microcomb and MRRs-based CDC together to show the capability and potential of realizing parallel massive data transmission and signal processing. By using these two integrated devices as the key components, we significantly improve the total data rate from 240 Gbit/s (line rate) to 1.68 Tbit/s (line rate) for the transmission system with integrated optical CDCs. We achieve a power consumption for CD compensation of more than six times less than that of the existing commercially available coherent transceiver modules. In addition, by utilizing the silicon nitride on SOI platform, these components can be integrated on a single chip in principle.

We would like to remove the word “original”.

Revision:

In the Abstract of the manuscript (from lines 24 to 26)

“In this study, we present a scalable on-chip parallel IM-DD data transmission system enabled by a single-soliton Kerr microcomb and a reconfigurable microring resonator (MRR)-based CD compensator (CDC).”

Comment #2-5

In the introduction, the authors write:

“The total data capacity grows 80 times from 640 GHz in 2010 to 51.2 THz in 2022 for data switching in datacenter interconnects (DCIs) ...”

This is a problematic sentence in that data carrying capacity is measured in bits/s, while the authors here use units of Hz. This also conflates the concept of switching with data transmission ... the authors should rethink what they are trying to present here and be clear in setting up this key problem statement.

Response:

We thank the reviewer for the kind remind and we are sorry for making such a mistake. What we want to present here is although the data capacity grows dramatically in the 12 years, the total power consumption also increases significantly, which imposes a demand for energy-efficient and cost-effective data transmission.

We would like to revise our manuscript as follows:

Revision:

At the beginning of the Introduction of the manuscript (from lines 41 to 44):

“Although the total data capacity grows 80 times from 640 Gbit/s in 2010 to 51.2 Tbit/s in 2022 in datacenter interconnects (DCIs), the total power consumption also grows dramatically with a 22-fold increase². As a result, there is a critical need for energy-efficient and cost-effective DCIs that can accommodate a substantial capacity.”

Comment #2-6

The authors make the bold claim that:

“For short-reach DCIs spanning distances of less than 40 km, the utilization of intensity modulation and direct detection (IM-DD) has emerged as a practical and preferred solution [3-5].”

This is a rather controversial statement, and is not really sufficiently backed up by the papers cited.

I would point the authors to the development of the 400G ZR standards, and the use of coherent 16-QAM within this for DCI applications (i.e. 2km to 80km or so).

Coherent is also starting to be considered by R&D divisions of major datacentre companies for intra-DC links (i.e. <2km). For example, in X. Zhou et al. (Google), JLT, v38 p475, 2020, they show that coherent looks good at >2km reach for 1.6 Tbit/s links. For DCI, then coherent already looks good at the 400G ZR level for > 2km reach (R. Nagarajan (Marvell), JLT, v39 p5221, 2021).

I think, that for the authors to really make a fair comparison, they need to consider these relatively recent developments in coherent DCI technologies to understand the potential future impact of their work and the comb-based IM/DD approach. Otherwise, their work is not put in the relevant context, where their dispersion compensator may make a compelling case for the use of IM/DD over inter DC distances.

Response:

We thank the reviewer for the comment and the quite helpful suggestions. We are sorry to make the bold claim. We discuss the development of IMDD and coherent as follows:

Until recently, the IMDD technology has been the clear choice for intra-datacenter interconnects (<2 km), owing to its low energy consumption and easy implementation¹. However, the scaling of data rate beyond 1.6 Tbit/s is challenging even for the O-band intra-datacenter scenarios as the IMDD system is less tolerant to optical impairments such as the fiber chromatic dispersion (CD)¹. The system suffers from an even larger CD in the C-band, which also limits the scaling of the data rate of IMDD systems working in the C-band.

For coherent technology, it has been widely used in long-haul and metro optical networks owing to its high receiver power sensitivity and high spectral efficiency¹. Due to the considerable reduction of cost, power consumption, and spatial size of this technology, the coherent technology has shown competitive performances for transmission within 100 km SMF¹. Recently, data transmission using coherent technology has already reached the fiber length of less than 10 km²⁻⁴, some of which have even been successfully demonstrated for intra-datacenter applications^{2, 3}.

There is a combination of IMDD and coherent technology for the inter-DCIs with a fiber length of up to 100 km⁵. In our work, we focus on the transmission of a fiber length of less than 40 km. What we want to show in our work is to demonstrate the feasibility of utilizing the IMDD system, which is based on the integrated microcomb and CDC, to realize the terabit-level data transmission in the C-band over tens of kilometers of SMF, and to show the potential of scaling the data rate to tens of terabits based on our proposed system.

Thus, based on the discussion above, we would like to revise our manuscript as follows:

Revision:

In the Introduction of the manuscript (from lines 44 to 56):

“There are mainly two technical approaches to the DCIs, namely the intensity modulation and direct detection (IM-DD) and the coherent schemes. Until recently, the IM-DD technology has been the clear choice for intra-DCIs (<2 km), owing to its low energy consumption and easy implementation³⁻⁵. However, the scaling of data rate beyond 1.6 Tbit/s is challenging even for the O-band intra-datacenter scenarios as the IM-DD system is less tolerant to optical impairments such as the fiber chromatic dispersion (CD)³. The system suffers from an even larger CD in the C-band, which also limits the scaling of the data rate of IM-DD systems working in the C-band. For the coherent technology, it has been widely used in long-haul and metro optical networks owing to its high receiver power sensitivity and high spectral efficiency³. Due to the considerable reduction of cost, power consumption, and spatial size of this technology, the coherent technology has shown competitive performances for transmission within 100 km single-mode fiber (SMF)³. Recently, the data transmission using coherent technology has already reached the fiber length of less than 10 km⁶⁻⁸, some of which have even been successfully demonstrated for intra-datacenter applications^{7, 8}.”

References

1. Zhou X, Urata R, Liu H, Beyond 1 Tb/s intra-data center interconnect technology: IM-DD OR coherent? *Journal of Lightwave Technology*, **38**: 475-484 (2020).
2. Maharry A *et al.*, First demonstration of an O-band coherent link for intra-data center applications. *Journal of Lightwave Technology*, **41**: 6643-6650 (2023).
3. Zhang T *et al.*, Cost-effective and hardware-efficient coherent scheme for 400g short-reach transmission. *Optical Fiber Communication Conference (OFC)*, paper M5G. 4 (2021).
4. Berikaa E *et al.*, Next-Generation O-band Coherent Transmission for 1.6 Tbps 10 km Intra-Datacenter Interconnects. *Journal of Lightwave Technology*, Early access (2023).
5. Nagarajan R, Lyubomirsky I, and Agazzi O, Low power DSP-based transceivers for data center optical fiber communications (invited tutorial). *Journal of Lightwave Technology*, **39**, 5221-5231 (2021).

Comment #2-7

The authors write in their introductions:

“To address the dispersion effect, various electronic and optical approaches have been employed, such as digital signal processing (DSP) [11-13] and dispersion compensating fibers (DCFs). However, these methods are burdened with limitations, particularly in terms of a high energy consumption and associated costs [14].”

They then switch topics to microcombs.

Comment #2-7a

The authors need to separate their arguments against DSP and optical methods here. DCFs are clearly not high energy consumption, especially compared to the author’s own

chip-based approach. DSP is also fairly clearly not comparatively high cost, given its prevalence in both IM/DD and coherent DCIs.

Response:

We thank the reviewer for the comment. We agree with the reviewer that we should separate these arguments here to make a clear statement. We would like to revise our manuscript as follows:

Revision:

In the Abstract of the manuscript (from lines 22 to 24):

“Electronic dispersion compensation based on digital signal processing algorithms suffers from high power consumption, while the optical approach using dispersion-compensating fibers lacks full reconfigurability.”

In the Introduction of the manuscript (from lines 73 to 77):

“However, the DSP-based coherent transceiver module suffers from a relatively high energy consumption of about 0.5 W/100G (5 pJ/bit) for 120-km-SMF CD compensation²⁵. Dispersion-compensating fibers (DCFs) are also commonly used to compensate for the CD of SMF. Nevertheless, using DCFs is burdened with a relatively high associated cost and low reconfigurability for flexible CD compensation of different fiber lengths.”

Comment #2-7b

The authors should here start to acknowledge the large number of tunable optical chromatic dispersion compensation techniques available, especially before the advent of coherent systems c.a. 2006. This then sets their compact chip-based system up in context

Response:

We thank the reviewer for the comment and the detailed suggestions. We agree that we should include the CD compensation techniques here.

Generally, the CD compensation techniques can be categorized into three types¹, namely (i) the compensation at the transmitter side, (ii) the inline compensation, and (iii) the post-compensation. We discuss these three kinds of CD compensation methods as follows:

(i) The CD compensation at the transmitter side. This kind of CD compensation is also regarded as pre-distortion compensation, in which the characteristics of the light pulse are modified before it transmits into the optical link to pre-compensate the impact of CD¹. One can reduce the penalty of CD by optimizing the frequency chirp of the transmitter^{2, 3}. In addition, the electronic pre-compensation that pre-distorts the amplitude and phase waveforms of the transmitted signal using digital processing can be utilized for the CD compensation⁴.

(ii) The inline compensation. This kind of technique includes but is not limited to the compensation realized by the dispersion compensating fibers (DCFs)⁵ and chirped fiber Bragg gratings (CFBGs)⁶. In Ref 5, the authors first demonstrated the use of DCFs for CD

compensation. The negative CD of DCFs in the C-band can be utilized to compensate for the positive CD induced by the SMF transmission. For the CFBGs, the signals at different wavelengths are reflected at different positions inside the CFBGs, leading to a shorter (longer) delay for the signal at longer (shorter) wavelength and thus the compensation of the CD.

(iii) The post compensation. This kind of technique is usually implemented on the electrical signals at the receiver and is realized by electronic equalization. The digital signal processing (DSP)-based electronic CD compensation can be used in IMDD⁷⁻⁹ and coherent transmission systems^{10, 11}. Particularly, the DSP-based CD compensation has been commonly used in the commercially available coherent transceiver modules¹²⁻¹⁴.

Based on the discussion above, we would like to revise our manuscript as follows:

Revision:

In the Introduction of the manuscript (from line 67 to line 73):

“To address the CD effect, various electronic and optical approaches have been employed, which can be generally categorized into three types¹², the pre-distortion compensation¹³⁻¹⁹, the inline compensation^{20, 21}, and the post compensation²²⁻²⁷, depending on the position of CD compensator (CDC). Digital signal processing (DSP)-based electronic CD compensation is usually applied at the receiver side as post compensation, which has been used in IM-DD²² and coherent^{23, 24} transmission systems. Particularly, it has been commonly used in the commercially available 400G ZR coherent transceiver modules²⁵⁻²⁷.”

References

1. Dar A B and Jha R K, Chromatic dispersion compensation techniques and characterization of fiber Bragg grating for dispersion compensation. *Optical and quantum electronics*, **49**: 1-35 (2017).
2. Gnauck A H *et al.*, Dispersion penalty reduction using an optical modulator with adjustable chirp. *IEEE Photonics Technology Letters*, **3**: 916-918 (1991).
3. Morito K *et al.*, Penalty-free 10 Gb/s NRZ transmission over 100 km of standard fiber at 1.55 μm with a blue-chirp modulator integrated DFB laser. *IEEE Photonics Technology Letters*, **8**: 431-433 (1996).
4. Killey R I *et al.*, Electronic dispersion compensation by signal predistortion using digital processing and a dual-drive Mach-Zehnder modulator. *IEEE Photonics Technology Letters*, **17**: 714-716 (2005).
5. Lin C, Kogelnik H, Cohen L G, Optical-pulse equalization of low-dispersion transmission in single-mode fibers in the 1.3–1.7- μm spectral region. *Optics Letters*, **5**: 476-478 (1980).
6. Hill K O *et al.*, Chirped in-fiber Bragg gratings for compensation of optical-fiber dispersion. *Optics Letters*, **19**: 1314-1316 (1994).
7. Yin M *et al.*, Transmission of a 56 Gbit/s PAM4 signal with low-resolution DAC and pre-equalization only over 80 km fiber in C-band IM/DD systems for optical interconnects.

Optics Letters, **46**: 5615-5618 (2021).

8. Zhong K *et al.*, Digital signal processing for short-reach optical communications: A review of current technologies and future trends. *Journal of Lightwave Technology*, **36**: 377-400 (2018).

9. Wu X *et al.*, C-band 100-GBaud PS-PAM-4 transmission over 50-km SSMF enabled by FIR-filter-based pre-electronic dispersion compensation. *Optics Express*, **31**: 17759-17768 (2023).

10. Taylor M G, Coherent detection method using DSP for demodulation of signal and subsequent equalization of propagation impairments. *IEEE Photonics Technology Letters*, **16**: 674-676 (2004).

11. Tsukamoto S, Kato K, Kikuchi K, Unrepeated transmission of 20-Gb/s optical quadrature phase-shift-keying signal over 200-km standard single-mode fiber based on digital processing of homodyne-detected signal for group-velocity dispersion compensation. *IEEE Photonics Technology Letters*, **18**: 1016-1018 (2006).

12. Nagarajan R, Lyubomirsky I, and Agazzi O, Low power DSP-based transceivers for data center optical fiber communications (invited tutorial). *Journal of Lightwave Technology*, **39**, 5221-5231 (2021).

13. <https://www.lumentum.com/en/products/400g-zr-zr-qsfp-dd-dco>

14. <https://www.innolight.com/en/goods/solution/cid/16.html>

Comment #2-7c

Ref [14] does not really make the point about high energy consumption and cost ...

Response:

We thank the reviewer for the comment and we are sorry for making such a mistake. Ref [14] does not make a point about the energy consumption and cost.

Revision:

We have removed this reference from the manuscript.

Comment #2-8

Ref [26] is claimed to be variously based on “perfect soliton crystals” and “soliton crystals with defects” ... which one is it (I think the “with defects”, looking at [26])?

Response:

We thank the reviewer for the kind reminder and we are sorry for making such mistakes. We have revised the statement as follows:

Revision:

In the Introduction of the manuscript (from line 87 to line 90):

“Recent advancements have demonstrated massively parallel data transmission using various types of microcomb sources, including bright single-solitons⁵², soliton crystals with

defects⁵³, and dark solitons⁵⁴.”

Comment #2-9

Is the rate of 55 Tb/s line rate for [25] (capacity before taking FEC into account), or net rate (capacity after subtracting for the FEC overhead)? (I think line ...)

Response:

We thank the reviewer for the kind reminder and we are sorry for the unclear statement. The 55 Tbit/s in Ref [25] is the line rate and we have revised this in the manuscript:

Revision:

In the Introduction of the manuscript (from line 90 to line 92):

“For example, the utilization of two interleaved single-soliton microcombs in a silicon nitride (Si_3N_4) microresonator enabled coherent data transmission with 179 carriers in the C- and L-bands, achieving a total line rate of 55 Tbit/s⁵².”

Comment #2-10

Is the rate of 40.1 Tb/s for [21] line or net? (I think net ...)

Response:

We thank the reviewer for the kind reminder and we are sorry for the unclear statement. The 40.1 Tbit/s in Ref 26 is the net rate. We would like to use the line rate to describe the data rate in Ref 26 for consistency. We have revised this in the manuscript:

Revision:

In the Introduction of the manuscript (from line 92 to line 94):

“Similarly, the soliton crystal with defects, generated in a doped silica glass microresonator, facilitated a line rate of 44.2 Tbit/s and a spectral efficiency of 10.2 bit/s/Hz⁵³.”

Comment #2-11

Again, the introduction closes with a paragraph that suggest that the authors system is “original”. I would again suggest that this is not really the case, and that that they are looking at one building block in the chain. I would remove this focus, and instead shift to focusing on the power consumption aspect.

Response:

We thank the reviewer for the comment. We agree with the reviewer’s comment and we have removed the word “original”. We would like to revise our manuscript as follows:

Revision:

In the last paragraph of the Introduction of the manuscript (from lines 102 to 103):

“In this paper, we present an approach to short-reach inter-DCIs based on an IM-DD optical communications architecture.”

Comment #2-12

At the end of the introduction, the authors state that they are using PAM4 and DMT as modulation schemes. To start to understand the power consumption of these schemes, the authors can refer to J. Cheng (Alibaba), Proc. OFC, W1F.2, 2019. Here they present graphs that show the breakdown of power consumption of PAM4 and DMT schemes, including DSP. This shows that DMT DSP consumes about as much power as coherent (noting that coherent explicitly includes chromatic dispersion compensation), while PAM4 has about half the power for DSP as coherent (noting that PAM4 DSP explicitly does not include dispersion compensation, just an FFE as the authors here use). It may be that the authors wish to foreshadow this here, or introduce this information later.

Response:

We thank the reviewer for the comment. In our work, we utilized the integrated MRRs-based optical CDC to compensate the CD of various lengths of SMF in different operation bandwidths. We note that the power consumption of our CDC does not vary much with the modulation format.

Comment #2-13

Figure 1 is somewhat misleading about what has been achieved in this demonstration. It would be better to clearly outline, in figure 1a, what has been demonstrated in this paper (i.e. the CD compensator), while clearly showing that this has not been implemented here with the ring-based Tx and Rx

Response:

We thank the reviewer for the comment. In our demonstration, we have achieved the on-chip MRR-based Si₃N₄ microcomb and the on-chip MRRs-based Si CDC. We would like to revise Fig. 1a as follows:

Revision:

Fig. 1a:

In the caption of the Fig. 1a (from line 129 to line 130):

“The red dashed boxes show the on-chip components that have been demonstrated as a proof of concept in this work.”

Comment #2-14

The authors state around line 184, that:

“The insertion loss increases from ~8.4 dB to ~14.8 dB at the wavelength around 1558.7 nm for dispersion compensation of 10-km to 40-km SMFs, respectively.”

Two things to note:

Comment #2-14a

The insertion loss changes significantly over the signal bandwidth, as shown in Fig 2a for example. This then means that the FFE (PAM) or 1-tap eq. (DMT) used in the receiver DSP will need to equalize out this loss, potentially increasing DSP complexity somewhat compared to a system without this eq.

Response:

We thank the reviewer for the comment. In our work, we utilized the double side band intensity modulation scheme, which generates two side bands equally distributed at the two sides of the optical carrier within the operation bandwidth of the CDC with a frequency span of f_{max} , where f_{max} is the maximum frequency of the RF signal that modulates the optical signal. In this case, when the wavelength of the laser source is set at the center of the operation bandwidth of the CDC, there is an inverse variation of the optical loss for the upper and lower sidebands, as shown in Fig. R3a. After the optical-to-electrical conversion, the upper and lower sidebands are shown in Fig. R3b. Due to the complementary power variation of the two sidebands, there is little power variation of the received electrical signal, as shown in Fig. R3c. Thus, the varied insertion loss of the CDC has little negative effect on the system. Meanwhile, this transmission performance of CDC is intrinsically certain kind of frequency domain equalization which is highly beneficial for widespread practical low pass systems, as well as advanced modulations like vestigial sideband (VSB) and single sideband (SSB) systems.

In the measurement, we utilized an offline digital signal processing (DSP) algorithm with a time-domain feed-forward equalization (FFE) at the receiver side to equalize the impairments of the back-to-back (BtB, without the SMFs and the MRR-based CDC) transmission of each wavelength channel. And the FFE taps were fixed for data transmission under different fiber lengths and CD compensation. According to the BER measurement results in Fig. 2(j), there is negligible degradation after CD compensation of different length of SMFs from 10 km to 40 km, which verifies the explanation in Fig. R3.

Fig. R3 **a** Schematic of one channel of the modulated signal after the CDC. **b** The upper and lower bands of the modulated signal after the optical-to-electrical conversion. **c** The received electrical signal.

We would like to revise the manuscript as follows:

Revision:

In the Results of the manuscript (from line 233 to line 236):

“We utilized an offline digital signal processing (DSP) algorithm with a time-domain feed-forward equalization (FFE) at the receiver side to equalize the impairments of the back-to-back (BtB, without the SMFs and the MRR-based CDC) transmission of each wavelength channel.”

In the Results of the manuscript (from line 239 to line 240):

“The FFE taps were identical to the BtB transmission.”

In the Results of the manuscript (from line 253 to line 255):

“Besides, even with a ~10-dB loss variation over the signal bandwidth as shown in Fig. 2a, the BER deviation is negligible compared with the impairments from CD of the SMFs.”

Comment #2-14b

This insertion loss means that the signal power needs to be boosted by an EDFA. To make a fair comparison with other dispersion compensating techniques, you need to then include EDFA power consumption.

A back of the envelope calculation:

Assuming your top received power is 3dBm (2mW), and your loss is about 10 dB (i.e. 90% loss), the EDFA needs to add in 1.8mW of power per channel. EDFAs have a wall-plug efficiency of about 10%, so the EDFA will need to be supplied with 18mW per channel. For 15 channels, this comes to 270 mW.

So, you will need to include EDFA power into your power consumption calculations, which will increase your power consumption by a factor of about 2x. For a 51 channel system, as you propose later on, then your power consumption is about 1W total (ring heaters + EDFA).

Response:

We thank the reviewer for the detailed suggestions for the power consumption calculation. We have discussed the power consumption of the CD compensation of our CDC, the existing 400G ZR coherent transceiver modules, and the next-generation coherent modules in detail in response to comment #2-2. We conclude the performances of different CD compensation methods as follows:

Our CDC has a normalized power consumption (normalized to pJ/bit for every 40 km SMF transmission) that is less than 1/6 of that of the 400G ZR coherent modules. In addition, the power consumption of our CDC is much less than that of the coherent DSP modules when the SMF length and the total data rate are larger.

We have revised the power consumption in the Discussion of the manuscript and

Supplementary Note 7 of the Supplementary Information. Please see the response to the comment #2-2 for the revisions.

Comment #2-15

The authors use a generic 20% SD-FEC and 7% HD-FEC threshold for their work. These are thresholds that are often associated with coherent systems, and are not optimized for energy consumption. There are, however, a range of FECs that are optimised for energy consumption in the 400G ZR standard (and some details for the proposed next-gen 800G ZR). E.g. the concatenated staircase FEC with a threshold of BER = $1.25e-2$, or KP-4 FEC with a threshold at BER = $2.2e-4$. These are probably more appropriate or relevant thresholds to consider if thinking about minimising energy consumption in the receiver side. OFEC is being considered for the future 800G ZR standard, and you can have a look at that in W. Wang et al., JLT, v41 p926, 2023. They present a “simplified OFEC” with 3x the power consumption of standard 400G ZR FEC, but with an increase pre-FEC BER threshold of $1.8e-2$.

I would suggest that the authors look at these options. Note that R. Nagarajan (Marvell), JLT, v39 p5221, 2021 state that KP4 FEC uses approximately 100x less power than the 400G ZR CFEC, and that the 400G ZR CFEC is stated to have a power consumption of 560 mW (similar to your dispersion compensator, including EDFA power to compensate for insertion loss).

Response:

We thank the reviewer for the comment and the kind suggestions. Indeed, the concatenated FEC (CFEC) and the KP-4 FEC are optimized for energy consumption, and the KP-4 FEC consumes approximately a hundred times less power than the 400G ZR CFEC. Utilizing these FEC thresholds can further lower the power consumption. Yet, in our demonstration, we did not encode the overhead into our transmitted data. No extra power consumption was consumed for the FEC at the receiver side of DSP. For the power consumption, we mainly focus on that of our CDC, and we have made a wholistic comparison of the power consumption of our CDC and other modules such as the existing coherent transceiver modules and the next generation coherent transceiver modules. Please see the response to the comment #2-2 for detailed discussion. We utilized the SD-FEC and HD-FEC thresholds to show that the data can be successfully transmitted under these two FEC thresholds. We agree with the reviewer’s comment that we should consider these energy-efficient FECs to further reduce power consumption in future experiments.

Comment #2-16

In figure 3, the caption for part f should state the received optical power at the photodiode.

Response:

We thank the reviewer for the kind reminder. We fully agree with the reviewer that we should state clearly the received optical power is measured at the photodiode. We think what the reviewer tends to point out is the caption for part g, rather than part f, of Fig. 3. We thus would like to revise the caption of part g of Fig. 3 as follows:

Revision:

In the caption of Fig. 3g of the manuscript (from lines 305 to 307):

“g BERs versus received optical power at the photodiode for the 20-km-long SMF transmissions of the 3rd, 6th, and 9th channels with CD compensation using the CW-laser and the microcomb as the light source, respectively.”

Comment #2-17

The main manuscript includes these references to the power consumption of the dispersion compensator:

“The maximum power consumption of our dispersion compensator is as low as 160 mW. See Supplementary Note 6 for the detailed calculation of the bit rate and Supplementary Note 3 for the consumption.”

As above, I don't think that this is sufficient analysis in the main manuscript, considering that the power consumption of your chromatic dispersion compensator is a really key point to what I see as the impact of this paper.

Response:

We thank the reviewer for the quite helpful suggestions. We have discussed the power consumption of CD compensation of our CD, the existing 400G ZR coherent transceiver modules, and the next generation coherent modules in detail in response to comment #2-2. And we have revised our manuscript accordingly. Please see the response to the comment #2-2.

We have revised the discussion of the power consumption in the Discussion of the manuscript and the Supplementary Note 7 in the Supplementary Information. Please see the response to the comment #2-2 for the revisions.

Comment #2-18

Ok, given that I've pointed out things to do with power consumption a few times, here's how I'd provide a back of the envelope calculation for the power consumption of CD compensation for 400G ZR ... and you need about 3 of these to get up to the data rate of your PAM4 system.

From R. Nagarajan (Marvell), JLT, v39 p5221, 2021: About 50% of the total power consumption for 400G ZR transceivers is the DSP. They also state that a 400G ZR module eats up about 16W worth of power. So, this gives about 8W per 400G ZR module dedicated to DSP.

Then, let's assume that the equalizer only eats up 10% of this power (this is an estimated proportion that I think might provide a lower bound for the DSP power consumption estimate), then the equalizer (that achieves CD compensation) eats up 0.8W per 400G, so 2.4W for 1.2 Tb/s.

This would tend to indicate that your IM/DD based CD compensation approach (at about 500mW, including EDFA for insertion loss mitigation) is lower power by a significant margin than an equivalent system using 400G ZR.

Please note that 800G ZR is expected to use about half the power per 100 Gb/s (27W /

800 G vs 24W / 400 G ... this is as per Nokia).

I provide this to show the authors how feasible it is to make the power comparisons that I suggest.

Response:

We thank the reviewer for the detailed analysis of the power consumption. We fully agree with the reviewer's suggestions on the power comparisons. As more power is consumed for the CD compensation of a longer SMF, we further normalize the power consumption to pJ/bit for every 40 km SMF transmission to make a fair comparison. The normalized power consumption of our CD is less than 1/6 of that of the commercially available 400G ZR coherent transceiver modules. Please see the response to the comment #2-2 for the details.

We have revised the discussion of the power consumption in the Discussion of the manuscript and the Supplementary Note 7 in the Supplementary Information. Please see the response to the comment #2-2 for the revisions.

Comment #2-19

In the discussion, the authors claim that they could with a matched microcomb/dispersion compensator FSR hit 51 wavelengths. Given that this is the basis for claiming scalability up to 10 Tbit/s, there needs to be some more details as to this claim (especially given that there will be less power per line available when going to wavelengths further from the pump ...)

Response:

We thank the reviewer for the comment and we are sorry for the vague description. In our experiment, we only utilized 15 comb lines due to the mismatch of FSR between the microcomb and the CDC. In our experiment, the used EDFA has a minimum input power of -30 dBm, which means that the comb lines with a maximum optical power of larger than -30 dBm can be utilized in principle, leading to up to 51 available channels of our existing microcomb.

Furthermore, if we want to scale the total data rate beyond 10 Tbit/s, we should adopt an MRR-based microcomb and an MRR-based CDC with a narrower channel spacing. We can also utilize a more efficient dark soliton microcomb as the light source because the optical power of each comb line of the dark soliton microcomb is much higher than that of the bright soliton microcomb. Petabit-per-second data transmission has been demonstrated in Ref 1 by utilizing the dark soliton microcomb. For example, we assume the channel spacing of the microcomb and the CDC is 50 GHz. In our experiment, when we modulate the comb lines with the 112 Gbit/s DMT format, the utilized bandwidth due to the modulation of each channel is around 40 GHz (the total bandwidth of the lower and upper band, obtained from Fig. 4c in the manuscript), which means our CDC should support an operation bandwidth of at least 40 GHz. We should note that this is feasible with 8 MRRs and the simulated transmission and group delay responses of CD compensation of 20 km SMF are shown in Fig. R4. Therefore, we can achieve a total data rate of 11.424 Tbit/s (102×112 Gbit/s) by using 102 channels.

Fig. R4 a, b Simulated (a) transmission and (b) group delay responses for the CD compensation of 20 km SMF for MRRs with a 50-GHz FSR.

We would like to revise our manuscript as follows:

Revision:

In the Discussion of the manuscript (from line 387 to line 394):

“Under the assumption of a matched FSR between the microcomb and the CDC, we can in principle utilize up to 51 channels with an output power of higher than -30 dBm from the microcomb, which encompasses a wavelength span ranging from 1540 nm to 1580 nm, to expand the total data rate to around 6 Tbit/s. To achieve a total bit rate exceeding 10 Tbit/s, the spacing of adjacent comb lines and the channel spacing of the CDC can be further reduced to 50 GHz. We can use the efficient dark soliton microcomb as the light source, which leads to at least 102 available channels of the microcomb and the CDC. We can still transmit 112 Gbit/s data per channel to achieve a total bit rate of larger than 10 Tbit/s. See Supplementary Note 8 for the detailed discussion.”

We have also added a detailed discussion of the scalability of the transmission system in the Supplementary Note 8:

“Supplementary Note 8: Scalability of the transmission system

In the parallel data transmission experiment, the operation bandwidth of our CDC is tuned to 50 GHz, while the transmitted data rate per channel is limited to 80 Gbps for the PAM4 modulation format, which is due to the limited bandwidth of the whole transmission system. We measured eye diagrams of a 45 Gbaud PAM4 signal, and they were blurred even under the BtB condition using a CW laser as the light source. Thus, to scale the transmission system to achieve a total bit rate exceeding 10 Tbit/s, we prefer to design the microcomb and the CDC with a narrower spacing of 50 GHz.

As illustrated in the discussion in the manuscript, the number of the available comb lines is 51 for our existing microcomb, and it can be doubled if the microcomb has an FSR of 50 GHz. To achieve a total bit rate of more than ten terabits, each channel should at least transmit a data rate of 100 Gbit/s. Compared with the PAM4, the modulation format of DMT is preferred as it has a higher spectral efficiency. In our experiment, the utilized bandwidth of the 112 Gbit/s DMT signal is around 40 GHz, indicating that the operation bandwidth of

the CDC should be at least 40 GHz. We note that this operation bandwidth can be achieved with 8 cascaded MRRs. Supplementary Fig. 14 shows the simulated transmission and group delay responses of the 8 MRRs for the CD compensation of 20 km SMF. A waveguide loss of 4 dB/cm is assumed. Therefore, we can achieve a total data rate of 11.424 Tbit/s by employing 102 comb lines.

Supplementary Fig. 14. Transmission and group delay characteristics of the MRRs with a 50-GHz FSR. a, b Simulated (a) transmission and (b) group delay responses for the CD compensation of 20 km SMF for MRRs with a 50-GHz FSR.”

References

1. Jørgensen A A *et al.*, Petabit-per-second data transmission using a chip-scale microcomb ring resonator source. *Nature Photonics*, **16**: 798-802 (2022).

Comment #2-20

In the discussion, the authors claim that they can go to a 50GHz spacing. Given the edges of their dispersion compensator system, I'm not sure this is going to be entirely true? At least, one would expect that the symbol rate per line would need to be reduced? There needs to be a bit more backing for this “scalable to beyond 10 Tb/s” claim ...

Response:

We thank the reviewer for the comment. We can achieve a data rate exceeding 10 Tbit/s without the reduction of the symbol rate per line. We have analyzed the realization of the beyond 10 Tbit/s data transmission in detail in response to the comment # 2-19. Please refer to that response.

Comment #2-21

The authors quite rightly begin to critique the conversion efficiency and power consumption of the microcomb. They focus on the dual-rind bright solitons from the Chalmers group (they should also note that that work is now out in *Nature Photonics*?). The authors might also like to consider the 75% efficiencies shown for laser cavity soliton microcombs (H. Bao *et al.*, *Nature Photonics*, v13, p384, 2019).

Response:

We thank the reviewer for the kind reminder. We have updated the citation of the dual-ring bright soliton to the latest version in Nature Photonics. Meanwhile, we have extended the discussion of other efficient soliton microcombs, including the laser cavity soliton microcombs and dark soliton microcombs. We would like to revise the Discussion and the References of the manuscript as follows:

Revision:

In the Discussion of the manuscript (from lines 403 to 406):

“Besides, the laser cavity soliton and the dark soliton microcombs have also been demonstrated to have a high conversion efficiency⁷⁵⁻⁷⁷. It is emphasized that the efficient soliton microcomb sources can support comb lines with high power, which reduces the gain required by the EDFA.”

In the References of the manuscript:

“References

75. Bao HL, *et al.* Laser cavity-soliton microcombs. *Nature Photonics* **13**, 384-389 (2019).

76. Helgason ÓB, *et al.* Dissipative solitons in photonic molecules. *Nature Photonics* **15**, 305-310 (2021).

77. Kim BY, *et al.* Turn-key, high-efficiency Kerr comb source. *Optics Letters* **44**, 4475-4478 (2019).”

Comment #2-22

When the authors are considering the dual ring bright combs, and state:

“resulting in increased power levels for the majority of the comb lines to -10 dBm, ...”

... this is maybe not quite true, as the high efficiency states occur for low pump powers, and so per line comb power goes down.

Please reconsider this statement ... it's not a straightforward as you present it, I think.

Response:

We thank the reviewer for the comment. The conversion efficiency is indeed inversely proportional to the pump power according to the conversion efficiency expression in Ref 1 of the response to this comment. However, the reduction of microring loss and a larger coupling coefficient between the bus waveguide and the Kerr microring can increase the conversion efficiency at the same pump laser power level. The dual-ring structure in Ref 58 of the manuscript has a low cavity loss and strong coupling. Besides, by reducing the mode mismatch between the pump laser and the resonance, the conversion efficiency can be significantly improved from 1% to 50%. Although the on-chip pump power of Ref 58 (previous manuscript, now Ref 74) is only near 10 dBm, the comb line can reach the power level of -10 dBm due to the extremely high conversion efficiency. In contrast, the on-chip power is near 23 dBm in our experiment, while the power of the comb line is -20 dBm for our soliton source. Furthermore, we also need to consider that the efficient dark microcombs² can support comb lines with much higher power with a similar pump power

level, compared with the bright single soliton³. Therefore, from a rigorous perspective, we reclaim that the efficient soliton microcomb can support comb lines with higher power which is not specific to the bright single soliton dual-ring structure of Ref 58 (Ref 74 in the current manuscript).

We have revised the manuscript as follows:

Revision:

In the Discussion of the manuscript (from line 404 to line 406):

“It is emphasized that the efficient soliton microcomb sources can support comb lines with high power, which reduces the gain required by the EDFA.”

References

1. Jang, J.K., *et al.*, Conversion efficiency of soliton Kerr combs. *Optics Letters*, **46**: 3657-3660 (2021).
2. Kim, B.Y., *et al.*, Turn-key, high-efficiency Kerr comb source. *Optics Letters*, **44**: 4475-4478 (2019)
3. Zhou, H., *et al.*, Soliton bursts and deterministic dissipative Kerr soliton generation in auxiliary-assisted microcavities. *Light Science & Applications*, **8**: 50 (2019).

Comment #2-23

The authors state:

“To further enhance the overall performance of the system, it is plausible to minimize the fiber-to-fiber loss associated with the dispersion compensator by introducing low-loss and broadband edge couplers [59-61].”

This would seem to be something you could put a number to fairly easily by looking at the out-of-band insertion loss vs in-band insertion loss for your device? So, if you took a state-of-the-art number for coupling loss, and then took the difference between that and the approx. 6dB out-of-band insertion loss you measure, then you would get a measure of the minimum expected insertion loss (instead of a vague promise for lower losses)?

Response:

We thank the reviewer for the comment. We checked the demonstrated coupling loss of various silicon-based edge couplers and found that the measured coupling loss of ~0.32 dB/facet in Ref 1 is relatively the smallest among the proposed edge couplers. In addition, we analyze the corresponding power consumption of our CDC, provided that we utilize the low-loss edge coupler for coupling. Compared with our existing grating coupler-based CDC, we can achieve a lower power consumption for both a longer SMF transmission and a larger data rate transmission. Please see the detailed analysis, calculation, and revisions of the power consumption in the response to comment #2-2.

References

1. Cheben P *et al.*, Broadband polarization independent nanophotonic coupler for silicon waveguides with ultra-high efficiency. *Optics Express*, **23**: 22553-22563 (2015).

Reviewer 3

>>In this manuscript, the authors demonstrate a parallel WDM data transmission using a silicon photonic coupled microring resonators to compensate for fibre dispersion. Parallel transmission with an integrated soliton microcomb source was also demonstrated. Although dispersion compensation based on single or cascaded microrings, as well as massive parallel data transmission using a microcomb as a multi-wavelength light source have been demonstrated previously, it appears that in this work the authors be able to combine both microcomb technology and on-chip dispersion compensator for parallel WDM transmission for the first time, demonstrating the potentials of on-chip solutions.

We thank the reviewer for his/her insightful evaluation of our work.

The following points need to be addressed before the manuscript can be considered for publication in Nature Communications.

Comment #3-1

The dispersion of the microrings was designed and calibrated at one wavelength channel. Over a wide wavelength range, the waveguide in the rings will have different dispersion and the couplers will have wavelength dependent coupling coefficients. How do these factors impact the dispersion of the dispersion compensator on the wavelength channels far away from the designed/calibrated channel? Please include the simulated and measured group delay for different wavelength channels.

Response:

We thank the reviewer for the comment. Indeed, the dispersion of the waveguide and coupling coefficient of MRRs are wavelength-dependent, which will induce the variation of CD over the wide wavelength range. The use of MZI tunable coupler instead of a directional coupler in the MRR has less coupling coefficient variation over the operation bandwidth. However, the insertion loss and phase response of the MMIs, the propagation loss of the waveguides, and the thermo-optic phase shifts are all wavelength-dependent, which results in wavelength-dependent coupling coefficients, and thus affects the envelope of the group delay response. We simulate the group delay responses for CD compensation of 40 km SMF within 32-GHz bandwidth across a large wavelength span ranging from 1540 nm to 1580 nm, using a linear changed coupling coefficient upon different wavelength channels. In the simulation, we also consider the material and structure dispersion (i.e., the effective refractive index of the mode varies with wavelength). Figs. R1a to R1c show the simulated results with the power coupling coefficient variation slope of 0.0005/nm, 0.002/nm, and 0.004/nm, respectively. The power coupling coefficient is larger at longer wavelengths. Fig. R1d shows the extracted average CD of each channel under the three power coupling coefficient variation slopes. The average CD is calculated as $D_{avg} = (\tau_1 - \tau_2)/(\lambda_1 - \lambda_2)$, where λ_1 and λ_2 are the shorter and longer wavelength ends across the operation bandwidth of each channel, τ_1 and τ_2 are the corresponding group delays, respectively. We can see that the average CD vary linearly with the wavelength. The variation of the envelope of the group delay response is larger with a larger power coupling coefficient

variation, which also leads to a larger CD variation across the wavelength span. Under the power coupling coefficient variation slope of 0.004/nm, the average CD ranges from approximately -798 ps/nm to -622 ps/nm within the 40-nm wavelength span.

Fig. R1. a, b, c Simulated group delay responses of the CD compensation of 40 km SMF within a bandwidth of 32 GHz with a power coupling coefficient slope of **(a)** 0.0005/nm, **(b)** 0.002/nm, and **(c)** 0.004/nm. PCC: power coupling coefficient. **d** Extracted average CD of each channel under the three PCC variation slopes.

We have characterized the measured transmission and group delay responses in the identical wavelength span of 1540 nm to 1580 nm in our manuscript, which are shown in Figs. 2c and 2e. In addition, we analyzed the insertion loss at the center of the operation bandwidth and the average CD of each channel of the CD compensator (CDC) across the 40-nm wavelength range. The analysis is shown in Figs. 2g and 2h. As can be seen, due to the existence of the wavelength-dependent group index and coupling coefficient of MRRs, the average CD shown in Fig. 2h shows a linear dependence on the wavelength, which is consistent with our simulations shown in Fig. R1. We should note that the degradation of the average CD of the 40-km-long SMF at the wavelength around 1540 nm is due to the limited dynamic range of our measurement equipment (Agilent 86038B).

Although the group delay and the corresponding CD vary with the wavelength, the measured BER results using the CW laser (Fig. 2j) demonstrate that the BERs are all below the 7% HD-FEC threshold, which indicates the CD variation has a limited influence on the quality of the data transmission.

We would like to revise our Supplementary Information (Supplementary Note 2) as follows:

Revision:

In the Supplementary Fig. 3, we add the four figures (Supplementary Figs. 3a to 3d):

We add the captions of the Supplementary Fig. 3a to 3d (from lines 119 to 122):

“**a, b, c** Simulated group delay responses of the CD compensation of 40 km SMF within a bandwidth of 32 GHz with a power coupling coefficient slope of **(a)** 0.0005/nm, **(b)** 0.002/nm, and **(c)** 0.004/nm. PCC: power coupling coefficient. **d** Extracted average CD of each channel under the three PCC variation slopes.”

We add the following statements in the Supplementary Note 2 (from lines 106 to 117, 127 to 134):

“We simulate the group delay responses for CD compensation of 40 km SMF across a large wavelength span ranging from 1540 nm to 1580 nm, using a linear changed coupling coefficient upon different wavelength channels. In the simulation, we also consider the material and structure dispersion (i.e., the effective refractive index of the mode varies with wavelength). Supplementary Figs. 3a to 3c show the simulated group delay responses with the power coupling coefficient variation slope of 0.0005/nm, 0.002/nm, and 0.004/nm, respectively. The power coupling coefficient is larger at longer wavelengths. The use of MZI tunable coupler instead of a directional coupler in the MRR has less coupling coefficient variation over the operation bandwidth. However, the insertion loss and phase response of the multimode interferometers (MMIs), the propagation loss of the waveguides, and the thermo-optic phase shifts are all wavelength-dependent, which results in wavelength-dependent coupling coefficients, thus affects the envelope of the group delay response. Supplementary Fig. 3d shows the extracted average CD of each channel under the three power coupling coefficient variation slopes. The average CD is calculated as $D_{avg} = (\tau_1 - \tau_2)/(\lambda_1 - \lambda_2)$, where λ_1 and λ_2 are the shorter and longer wavelength ends across the operation bandwidth of each channel, τ_1 and τ_2 are the corresponding group delays, respectively. We can see that the average CD vary linearly with the wavelength. The variation of the envelope of the group delay response is larger with a larger power coupling coefficient variation, which also leads to a larger CD variation across the wavelength span. Under the power coupling coefficient variation slope of 0.004/nm, the

average CD ranges from approximately -798 ps/nm to -622 ps/nm within the 40-nm wavelength span.”

Comment #3-2

The dispersion compensator shows a non-uniform transmission with more than 10 dB variation within the operational bandwidth. Please discuss the impact of this non-uniform loss on the transmission penalty.

Response:

We thank the reviewer for the comment. In our work, we utilized the double side band intensity modulation scheme, which generates two side bands equally distributed at the two sides of the optical carrier within the operation bandwidth of the CDC with a frequency span of f_{max} , where f_{max} is the maximum frequency of the RF signal that modulates the optical signal. In this case, when the wavelength of the laser source is set at the center of the operation bandwidth of the CDC, there is an inverse variation of the optical loss for the upper and lower sidebands, as shown in Fig. R2a. After the optical-to-electrical conversion, the upper and lower sidebands are shown in Fig. R2b. Due to the complementary power variation of the two sidebands, there is little power variation of the received electrical signal, as shown in Fig. R2c. Thus, the varied insertion loss of the CDC has little negative effect on the system. Meanwhile, this transmission performance of CDC is intrinsically certain kind of frequency domain equalization which is highly beneficial for widespread practical low pass systems, as well as advanced modulations like vestigial sideband (VSB) and single sideband (SSB) systems.

In the measurement, we utilized an offline digital signal processing (DSP) algorithm with a time-domain feed-forward equalization (FFE) at the receiver side to equalize the impairments of the back-to-back (BtB, without the SMFs and the MRR-based CDC) transmission of each wavelength channel. And the FFE taps were fixed for data transmission under different fiber lengths and CD compensation. According to the BER measurement results in Fig. 2(j), there is negligible degradation after CD compensation of different length of SMFs from 10 km to 40 km, which verifies the explanation in Fig. R2.

Fig. R2 a Schematic of one channel of the modulated signal after the CDC. b The upper and lower bands of the modulated signal after the optical-to-electrical conversion. c The received electrical signal.

We would like to revise the manuscript as follows:

Revision:

In the Results of the manuscript (from line 233 to line 236):

“We utilized an offline digital signal processing (DSP) algorithm with a time-domain feed-forward equalization (FFE) at the receiver side to equalize the impairments of the back-to-back (BtB, without the SMFs and the MRR-based CDC) transmission of each wavelength channel.”

In the Results of the manuscript (from line 239 to line 240):

“The FFE taps were identical to the BtB transmission.”

In the Results of the manuscript (from line 253 to line 255):

“Besides, even with a ~10-dB loss variation over the signal bandwidth as shown in Fig. 2a, the BER deviation is negligible compared with the impairments from CD of the SMFs.”

Comment #3-3

In this work, the authors demonstrate the maximum operational bandwidth of 50 GHz with 8 mirroring rings. While the FSR (or WDM channel spacing) is 99.5 GHz. The spectral utilization is just above 50%, resulting in low link spectral efficiency. Increasing spectral utilization would require more rings, hence increasing power consumption. This is also true for increasing fibre length. How does the number of rings required and power consumption increase with operational bandwidth and fibre length? What is maximum operational bandwidth that can be achieved with 40 km fibre length and the power consumption? How many rings are required to compensate for 40 km fibre with a 50 GHz bandwidth?

Response:

We thank the reviewer for the comment. We discuss these issues as follows:

(i) Number of MRRs and corresponding power consumptions for various operational bandwidth and fiber length. Table R1 summarizes the required number of MRRs for various operational bandwidths and fiber lengths. Here we consider a fiber length of 40 km, 60 km, and 80 km, and an operational bandwidth of 40 GHz, 50 GHz, and 60 GHz. We assume an 18 ps/(nm·km) dispersion of the SMF. Fig. R3 shows the simulated transmission and group delay responses for the listed conditions of 40 km, 60 km, and 80 km SMF transmission. We can see that 14 MRRs are required for the CD compensation of 40 km SMF within a bandwidth of 50 GHz. In addition, we also simulate the CD compensation of 100 km and 120 km SMF but with a smaller operational bandwidth. We need 15 and 17 MRRs to achieve the CD compensation of 100 km and 120 km SMF within a bandwidth of 32 GHz, respectively. Nevertheless, we can infer from the simulations that extra 4 MRRs, 6 MRRs, and 8 MRRs are required for an extra 10 GHz operational bandwidth for CD compensation of 40 km, 60 km, and 80 km SMF, respectively. We can also deduce the number of MRRs for CD compensation of 100 km SMF under various bandwidths. In this case, we need 26 MRRs to realize the CD compensation for 40 km with a bandwidth of 80 GHz. We should note that we can halve the FSR of the microcomb and the CDC to achieve a total data rate of larger than 10 Tbit/s while maintaining the number

of MRRs at 8, as discussed in the response to comment #3-6. In this case, we do not need to use too many MRRs as shown in Table R1.

Table R1. Number of MRRs required for different bandwidth and fiber length

Number of MRRs		Length of SMF (km)				
		40	60	80	100	120
Operational bandwidth (GHz)	32	/	/	/	15	17
	40	10	13	17	/	
	50	14	19	25		
	60	18	25	33		
	70 (Deduced)	22	31	41		
	80 (Deduced)	26	37	49		

Fig. R3. Simulated transmission and group delay responses for various operational bandwidth and fiber lengths. **a, c, e** Simulated transmission responses for an operational bandwidth of **(a)** 40 GHz, **(c)** 50 GHz, and **(e)** 60 GHz for CD compensation of 40 km, 60 km, and 80 km SMF. **b, d, f** Simulated group delay responses for an operational bandwidth of **(b)** 40 GHz, **(d)** 50 GHz, and **(f)** 60 GHz for CD compensation of 40 km, 60 km, and 80 km SMF.

(ii) Power consumption. In our work, the measured power consumption is 160 mW for 8 MRRs, corresponding to an average power consumption of 20 mW for each MRR. If we only look at the power consumption of the MRRs, Fig. R4 shows how the calculated power consumption increases with the bandwidth and fiber length. The power consumption increases linearly with the operational bandwidth under various lengths of SMF. The increment of power consumption increases linearly with the lengths of SMF.

Fig. R4. Power consumption of the CDC for different operational bandwidth and lengths of SMF. a Power consumption of the CDC for different operational bandwidth under various lengths of SMF. **b** Power consumption for different lengths of SMF under different operational bandwidths.

We would like to add the statements as Supplementary Note 9 in the Supplementary Information:

Revision:

“Supplementary Note 9: The required number of MRRs and the corresponding power consumption for CD compensation of various fiber lengths and operation bandwidth

Supplementary Table 6 summarizes the required number of MRRs for various operational bandwidth and fiber length. We consider a fiber length of 40 km, 60 km, and 80 km, and an operation bandwidth of 40 GHz, 50 GHz, and 60 GHz. We assume an 18 ps/(nm·km) dispersion of the SMF. Supplementary Fig. 15 shows the simulated transmission and group delay responses for the listed conditions of 40 km, 60 km, and 80 km SMF transmission. We can see that 14 MRRs are required for the CD compensation of 40 km SMF within a bandwidth of 50 GHz. In addition, we also simulate the CD compensation of 100 km and 120 km SMF but with a smaller operation bandwidth. We need 15 and 17 MRRs to achieve the CD compensation of 100 km and 120 km SMF within a bandwidth of 32 GHz, respectively. We can infer from the simulations that extra 4 MRRs, 6 MRRs, and 8 MRRs are required for an extra 10 GHz operational bandwidth for CD compensation of 40 km, 60 km, and 80 km SMF, respectively. And we can also deduce the number of MRRs for CD compensation of 100 km SMF under various bandwidth. In this case, we need 26 MRRs to realize the CD compensation for 40 km with a bandwidth of 80 GHz. We should note that we can halve the FSR of the microcomb and the CDC to achieve a total data rate of larger than 10 Tbit/s while maintaining the number of MRRs at 8, as discussed in the Supplementary Note 8. In this case, we do not need to use too many MRRs as shown in the Supplementary Table 6.

Supplementary Table 6. Number of MRRs required for different bandwidth and fiber lengths

Number of MRRs	Length of SMF (km)				
	40	60	80	100	120

Operation bandwidth (GHz)	32	/	/	/	15	17
	40	10	13	17	/	
	50	14	19	25		
	60	18	25	33		
	70 (Deduced)	22	31	41		
	80 (Deduced)	26	37	49		

In our work, the measured power consumption is 160 mW for 8 MRRs, corresponding to an average power consumption of 20 mW for each MRR. If we only consider the power consumption of the MRRs, Supplementary Fig. 16 shows how the calculated power consumption increases with the bandwidth and fiber length. The power consumption increases linearly with the operational bandwidth under various lengths of SMF. And the increment of power consumption increases linearly with lengths of SMF.

Supplementary Fig. 15. Simulated transmission and group delay responses for various operation bandwidth and fiber length. a, c, e Simulated transmission responses for an operational bandwidth of **(a)** 40 GHz, **(c)** 50 GHz, and **(e)** 60 GHz for CD compensation of 40 km, 60 km, and 80 km SMF. **b, d, f** Simulated group delay responses for an operational bandwidth of **(b)** 40 GHz, **(d)** 50 GHz, and **(f)** 60 GHz for CD compensation of 40 km, 60 km, and 80 km SMF.

Supplementary Fig. 16. Power consumption of the CDC for different operational bandwidth and lengths of SMF. a Power consumption of the CDC for different operational bandwidth under various lengths of SMF. **b** Power consumption for different lengths of SMF under different operational bandwidth.”

Comment #3-4

In Fig. 2(h), the average dispersion for 40 km fibre increases for channels close to 1540 nm. This behaviour does not present for other fibre lengths. What is the cause of this behaviour?

Response:

We thank the reviewer for the comment. We have analyzed the cause of this issue in our manuscript (lines 190~192). The group delay and the corresponding CD degradation at the wavelength around 1540 nm are caused by the limited dynamic range of our measurement equipment, which is the Photonic Dispersion and Loss Analyzer (PDLA).

To further confirm and check that this issue results from the limitation of our equipment, we measured the eye diagrams and the corresponding BERs of 64 Gbit/s PAM4 data transmission using a CW laser as a light source, as shown in Figs. 2i and 2j. The measured BER at around 1540 nm for the CD compensation of 40 km SMF is below the 7% HD-FEC threshold, indicating that the data has been successfully transmitted and the degradation is limited by our instrument.

Comment #3-5

In the demonstrations, 4-PAM and DMT with 16-QAM modulation formats were used. Can the number of levels in the multi-level modulation be increased to further increase the data rate? What is the limiting factor in the current demonstration? Is it the dispersion compensation or others?

Response:

We thank the reviewer for the comment. Utilizing higher-order modulation formats can certainly increase the overall transmission rate of the system, but it also imposes higher requirements on the signal-to-noise ratio (SNR) and requires more complex DSP¹. If we utilize the higher order PAM signals, the power consumption of the entire system (including

the one for DSP) is also increased to achieve a higher SNR, which is not desired for the data center interconnect applications where the power consumption is of vital importance. Thus, PAM4 is a good option to reach the compromise between the achievable bit rate and system performance¹. We find that many commercially available 400G, 800G, and 1.6T optical transceiver modules using the IMDD scheme also use the PAM4 modulation format to transmit data²⁻⁷. In addition, the DMT modulation scheme is also an attractive approach to realizing low-cost short-reach systems^{1, 8}. Based on the above considerations, we finally chose the PAM4 and DMT modulation schemes in our demonstrations.

References

1. Zhong K *et al.*, Digital signal processing for short-reach optical communications: A review of current technologies and future trends. *Journal of Lightwave Technology*, **36**: 377-400 (2018).
2. <https://www.fs.com/products/105875.html>
3. <https://www.fs.com/products/106421.html>
4. <https://www.fs.com/products/142441.html>
5. <https://www.fs.com/products/150960.html>
6. https://www.accelink.com/en/lighting_your_dreams/products_optical-transceivers/421.html
7. https://www.accelink.com/en/lighting_your_dreams/products_optical-transceivers/388.html
8. Xie C *et al.*, Single-VCSEL 100-Gb/s short-reach system using discrete multi-tone modulation and direct detection. *Optical Fiber Communication Conference (OFC)*, paper Tu2H. 2 (2015).

Comment #3-6

In the discussion, the authors suggested to reduce the spacing between adjacent comb lines to 50 GHz to increase the total data rate exceeding 10 Tbit/s. To do this, the FSR of the dispersion compensator also need to be halved to 50 GHz. As a consequence, the operational bandwidth of the dispersion compensator will be reduced. In order to maintain the same operational bandwidth and per channel data rate, the number of rings in the dispersion compensator need to be significantly increased (see point 3 above). If keeping the number of rings unchanged, will half the ring FSR also cause the operational bandwidth reduce by half?

Response:

We thank the reviewer for the comment. We assume the channel spacing of the microcomb and the CDC is 50 GHz. In our experiment, when we modulate the comb lines with the 112 Gbit/s DMT format, the utilized bandwidth due to the modulation of each channel is around 40 GHz (the total bandwidth of the lower and upper band, obtained from Fig. 4c in the

manuscript), which means our CDC should support an operation bandwidth of at least 40 GHz. We should note that this is feasible with 8 MRRs and the simulated transmission and group delay responses of CD compensation of 20 km SMF are shown in Fig. R5. Benefit from the periodical response of the MRR-based CDC, it is quite suitable to reduce the FSR of the MRRs to improve the spectral efficiency and total line rate, instead of increasing the operation bandwidth of the CDC, as discussed in the response of comment #3-3.

In our experiment, the used EDFA has a minimum input power of -30 dBm, which means that the comb lines with a maximum optical power larger than -30 dBm can be utilized in principle, leading to up to 51 available channels of our existing microcomb. Therefore, we can achieve a total data rate of 11.424 Tbit/s (102×112 Gbit/s) by using 102 channels if the FSR is halved.

Fig. R5 a, b Simulated (a) transmission and (b) group delay responses for the CD compensation of 20 km SMF for MRRs with a 50-GHz FSR.

We would like to revise our manuscript as follows:

Revision:

In the Discussion of the manuscript (from line 387 to line 394):

“Under the assumption of a matched FSR between the microcomb and the CDC, we can in principle utilize up to 51 channels with an output power of higher than -30 dBm from the microcomb, which encompasses a wavelength span ranging from 1540 nm to 1580 nm, to expand the total data rate to around 6 Tbit/s. To achieve a total bit rate exceeding 10 Tbit/s, the spacing of adjacent comb lines and the channel spacing of the CDC can be further reduced to 50 GHz. We can use the efficient dark soliton microcomb as the light source, which leads to at least 102 available channels of the microcomb and the CDC. We can still transmit 112 Gbit/s data per channel to achieve a total bit rate of larger than 10 Tbit/s. See Supplementary Note 8 for the detailed discussion.”

We have also added a detailed discussion of the scalability of the transmission system in the Supplementary Note 8:

“Supplementary Note 8: Scalability of the transmission system

In the parallel data transmission experiment, the operation bandwidth of our CDC is tuned to 50 GHz, while the transmitted data rate per channel is limited to 80 Gbps for the PAM4 modulation format, which is due to the limited bandwidth of the whole transmission system. We measured eye diagrams of a 45 Gbaud PAM4 signal, and they were blurred even under the BtB condition using a CW laser as the light source. Thus, to scale the transmission system to achieve a total bit rate exceeding 10 Tbit/s, we prefer to design the microcomb and the CDC with a narrower spacing of 50 GHz.

As illustrated in the discussion in the manuscript, the number of the available comb lines is 51 for our existing microcomb, and it can be doubled if the microcomb has an FSR of 50 GHz. To achieve a total bit rate of more than ten terabits, each channel should at least transmit a data rate of 100 Gbit/s. Compared with the PAM4, the modulation format of DMT is preferred as it has a higher spectral efficiency. In our experiment, the utilized bandwidth of the 112 Gbit/s DMT signal is around 40 GHz, indicating that the operation bandwidth of the CDC should be at least 40 GHz. We note that this operation bandwidth can be achieved with 8 cascaded MRRs. Supplementary Fig. 14 shows the simulated transmission and group delay responses of the 8 MRRs for the CD compensation of 20 km SMF. A waveguide loss of 4 dB/cm is assumed. Therefore, we can achieve a total data rate of 11.424 Tbit/s by employing 102 comb lines.

Supplementary Fig. 14. Transmission and group delay characteristics of the MRRs with a 50-GHz FSR. a, b Simulated (a) transmission and (b) group delay responses for the CD compensation of 20 km SMF for MRRs with a 50-GHz FSR.”

Comment #3-7

Please comment on the long-term stability of the dispersion compensation performance and thermal crosstalk compensation as well as system BER.

Response:

We thank the reviewer for the comment. We discuss the long-term stability of our work in the following sections:

(i) The CD compensation performance (thermal crosstalk compensation included).

In our work, we utilized our MRRs-based CDC in the data transmission system after the calibrations of MRRs and the thermal crosstalk compensation (TCC). The TCC is

implemented as part of the calibration process. The main issue that influences the long-term performance of our CDC is the ambient temperature variance, which may mainly affect the resonances of MRRs and thus the overall performance of the CDC. In our parallel data transmission experiment, we measured the data transmission performance successively from channel 1 to 15 with CD compensation for each comb line, and this is already a long-term measurement that lasts for more than 3 hours. During this measurement, TECs placed underneath the CDC and the microcomb were used to stabilize the temperature of the chips. In addition, we note that after the calibration we recorded the required voltages that should be applied to the 8 MRRs. We used these voltages as the applied ones each time we implemented the measurements. During the entire measurement period (at least one month, including the measurements for the CW laser source and the comb laser source), we did not observe a noticeable performance deterioration, and the measured eye diagrams and constellations shown in Figs. 3e and 4f in the manuscript and Supplementary Figs. 10 and 11 verify that the CD compensation, including the TCC, works well in a long-term measurement.

(ii) The system performance. As can be seen from Fig. 3f, the measured BERs for data transmission based on the PAM4 modulation format did not deteriorate significantly for different channels. The differences between the levels of BERs result from the free spectral range (FSR) of the microcomb and the CDC. The case is similar to the experimental results shown in Fig. 4e. These measured BERs demonstrate that the BERs of the system did not vary considerably during the long-period measurement, indicating that the system performance is relatively stable for the long-term experiment.

Comment #3-8

Please benchmark the performance of the on-chip dispersion compensator used in this work with other dispersion compensation techniques for parallel WDM transmission (such as DCF).

Response:

We thank the reviewer for the comment. We make a comparison of the CD compensation realized by various approaches, including the integrated optical devices, the existing 400G ZR coherent transceiver modules, and next-generation 800G and 1.6T coherent transceiver modules. As the data rate of the data transmission with CD compensation based on DCFs is relatively difficult to quantify, we did not include the comparison of CD compensation based on the DCFs. The integrated optical devices-based works that report the highest data line rate are chosen and listed in Table 1 in the manuscript. Please see the comparison as follows:

Revision:

In the Discussion of the manuscript (Table 1, line 377):

Table 1. Performance comparison of various CD compensation methods

Method/ Company	CBG ⁷¹	MRR ³⁹	MZI ³⁴	Marvell ²⁵	Lumentum ²⁶ InnoLight ²⁷	Next generation coherent ^{72, 73}		This work	
Approach	CBG ^a	MRRs	MZIs	DSP				MRRs	
Platform/Node	SOI	Si ₃ N ₄		7 nm		5 nm	3 or 2 nm	SOI	
Insertion loss (dB)	~6	0.8	20	/				10	
CD (ps/nm)	480	-170	-500	Max. -2400				-355 (Max. -720)	
FSR (GHz)	/	100	100	/				99.5	
Bandwidth (GHz)	~110	50	15	/				50	
SMF length (km)	28	10	40	Max. 120				20 (Max. 40)	
Max. data line rate (Gbit/s)	60	240	53.125	400		800	1600	1680	
Power consumption ^b (pJ/bit)	/			~1.7	~1.9	~1.3	~1.6	~0.86	~0.304
Footprint ^c	5 mm	2.2 × 0.56 mm ²	9.89 × 22.5 mm ²	89.4 × 18.35 × 8.5 mm ³		100.4 × 22.58 × 13.0 mm ³	/	2 × 1 mm ²	

^aCBG stands for chirped Bragg grating.

^bThis power consumption has been normalized to the CD compensation of 40 km SMF.

^cFor the on-chip optical CDCs, the footprint refers to the one of the chips. For the commercial modules, the footprint is the one for the entire module.

We have also added Supplementary Note 7 for a detailed discussion of the power consumption.

References

1. Giuntoni I *et al.*, Integrated dispersion compensator based on apodized SOI Bragg gratings. *IEEE Photonics Technology Letters*, **25**: 1313-1316 (2013).
2. Alam M S *et al.*, 224 Gb/s transmission over 10 km of SMF at 1550 nm enabled by a SiN optical dispersion compensator and stokes vector direct detect receiver. *Signal Processing in Photonic Communications*, paper SpM4I. 5 (2020).
3. Brodnik G M *et al.*, Extended reach 40km transmission of C-band real-time 53.125 Gbps PAM-4 enabled with a photonic integrated tunable lattice filter dispersion compensator. *Optical Fiber Communications Conference and Exposition (OFC)*. IEEE (2018).
4. <https://www.teraxion.com/en/products/optical-communications/slope-matched-tunable-dispersion-compensator/>

5. Nagarajan R, Lyubomirsky I, and Agazzi O, Low power DSP-based transceivers for data center optical fiber communications (invited tutorial). *Journal of Lightwave Technology*, **39**, 5221-5231 (2021).

6. <https://www.lumentum.com/en/products/400g-zr-zr-qsfp-dd-dco>

7. <https://www.innolight.com/en/goods/solution/cid/16.html>

8. <http://www.qsfp-dd.com/wp-content/uploads/2021/01/2021-QSFP-DD-MSA-Thermal-Whitepaper-Final.pdf>

9. Tauber D *et al.*, Role of coherent systems in the next DCI generation. *Journal of Lightwave Technology*, **41**: 1139-1151 (2023).

Other minor comments:

Comment #3-9

In the first paragraph of the Introduction section, the unit for data capacity was incorrectly used (as GHz).

Response:

We thank the reviewer for the kind reminder and we are sorry for making such a mistake. What we want to present here is although the data capacity grows dramatically in the 12 years, the total power consumption also increases significantly, which imposes a demand for energy-efficient and cost-effective data transmission.

We would like to revise our manuscript as follows:

Revision:

In the Introduction of the manuscript (from lines 41 to 44)

“Although the total data capacity grows 80 times from 640 Gbit/s in 2010 to 51.2 Tbit/s in 2022 in datacenter interconnects (DCIs), the total power consumption also grows dramatically with a 22-fold increase². As a result, there is a critical need for energy-efficient and cost-effective DCIs that can accommodate a substantial capacity.”

REVIEWER COMMENTS

Reviewer #1 (Remarks to the Author):

The authors claim to demonstrate the first on-chip parallel signal transmission and dispersion compensation using a single-soliton microcomb and a dispersion compensator based on microring resonators (MRRs). The experimental results are impressive. Although the concept of using MRRs for dispersion compensation has been previously reported and there is nothing unique about the design and fabrication of the Soliton source, the integration of both devices in a single experimental setup is novel and highlights the scalability silicon photonics offers.

Comments:

The authors have addressed all my comments and questions. I have only one extra comment that arose from the power consumption calculations listed in Table 1. Performance comparison of various CD compensation methods. The authors assumed that the power consumption of the DSP is 50% of the overall pluggable module, which is a reasonable assumption. However, they assumed that 25% of the DSP power is dedicated for CD compensation, which is exaggerated. The DSP (ASIC engine) power consumption is dominated by the digital-to-analog converters (DACs), analog-to-digital converters (ADCs), and the FEC engines, which leaves a very small portion for the actual clock and data recovery (CDR) module. The chromatic dispersion compensation is a power power-hungry component of the CDR, but I suspect that it consumes 25% of the whole ASIC engine power. Can the authors either support their assumption or correct it to better reflect the practical implementation?

Reviewer #2 (Remarks to the Author):

My initial comments were as Reviewer 2.

The Authors have provided a good response to the points I raised in my previous review. I hope that they agree that their manuscript is much improved over their initial submission – I appreciate the work that has gone into their changes.

I have only a few minor further points that I think need to be considered / acknowledged in the text.

1) When quoting numbers for power consumption, it is really important to be clear that these numbers are extrapolated.

E.g. in the abstract, where you state that “our method features a remarkably low energy consumption of ~ 0.304 pJ/bit”

a) This number is extrapolated from an assumed EDFA wall-plug efficiency of 16%, if I have read your manuscript correctly. So, your method does not “feature” this power consumption, but instead you extrapolate this number. Similarly, you are extrapolating an energy consumption figure for the coherent systems you compare to.

You might instead reasonably state:

“From measurements, we extrapolate an energy efficiency of 0.3pJ/bit for CD compensation of 40-km SSMF, which we calculate as being around 6 times less than that of commercially available 400G ZR coherent transceivers modules”.

Please be careful to write inn caveats each time you mention energy efficiency numbers that you have calculated with some assumptions.

E.g., where you write “power consumption is...”, replace with “power consumption is estimated to be ...”

b) If you are suggesting that this measure is approximate (i.e. ~ 0.304), then you should not be quoting a number to 3 significant figures. Use ~ 0.3 pJ/bit instead ...

Please truncate energy consumption figures throughout the manuscript to this level of precision where you provide them, as your estimates are not accurate to within $<1\%$ of the nominal value ...

2) Similarly, in the discussion, please place in a clear description of the key assumptions you have made in your energy efficiency calculation, in addition to pointing the reader toward your supplementary materials.

E.g., where you have:

“See Supplementary Note 7 ...power consumption.”

Please add in, such that this reads something like:

“See Supplementary Note 7 ...power consumption. The key assumptions we make are a wall-plug efficiency for EDFAs of around 16%, and that CDC consumes approximately 12.5% of the power used in a coherent transceiver.”

3) The authors still do not provide a justification for the degree of reconfigurability they think is needed for DCI links. This I think is required so that the reader can understand the context of the work presented.

I imagine that the authors could be thinking about shipping transceiver units with this compensator included, that then need to be tuned to the correct dispersion over a broad range? They might be thinking that the DCI network is reconfigurable (e.g. through WSS/ROADM), and therefore dispersion compensation needs to be tunable?

Otherwise, it would seem to be a trivial exercise to add in a fixed dispersion compensator.

4) When comparing your result to the energy consumption of DCF, it should be acknowledged that DCF is colourless, and so does not require temperature control to operate (unlike wavelength selective approaches such as your own, or other channelized dispersion compensators).

Please be careful to include this caveat when comparing your work against DCF in terms of energy consumption.

5) Can the authors label where the demonstration that they provide correlates with the curves in Fig 5, please?

6) Table 1 compares integrated CDC approaches only. The rest of the text focuses only on DCF as a non-integrated approach. I agree that keeping table 1 to integrated approaches is a good idea, but other tuneable compensators should also be acknowledged in the text (even if only to then establish why your approach might have advantages).

I would encourage the authors to also compare against established channelized dispersion compensators, such as cascaded chirped fibre Bragg gratings (e.g.

teraxion.com/en/products/optical-communications/tunable-dispersion-compensation-module-

benchtop/) and others (I suspect that this is a Gires-Tournois interferometer:
fiberroad.com/products/dcm/50ghz-100ghz-tunable-dispersion-compensator-tdcm/)
Here, I think it makes sense to then talk about how your approach is integrable, as opposed to
suggesting that any comparison should only be made in terms of power consumption.
In the case of DSP, clearly a head-to-head comparison was needed, as you are comparing chips with
chips. For physically larger alternatives, I think that is your major point of difference.

Reviewer #3 (Remarks to the Author):

Thank you the authors for the detailed responses to my comments. My comments have been
adequately addressed in the response letter and in the revised manuscript.

Reviewer #3 (Remarks on code availability):

N/A

Responses to the reviewers

Reviewer 1

>>The authors claim to demonstrate the first on-chip parallel signal transmission and dispersion compensation using a single-soliton microcomb and a dispersion compensator based on microring resonators (MRRs). The experimental results are impressive. Although the concept of using MRRs for dispersion compensation has been previously reported and there is nothing unique about the design and fabrication of the Soliton source, the integration of both devices in a single experimental setup is novel and highlights the scalability silicon photonics offers.

Comment #1-1:

The authors have addressed all my comments and questions. I have only one extra comment that arose from the power consumption calculations listed in Table 1. Performance comparison of various CD compensation methods. The authors assumed that the power consumption of the DSP is 50% of the overall pluggable module, which is a reasonable assumption. However, they assumed that 25% of the DSP power is dedicated for CD compensation, which is exaggerated. The DSP (ASIC engine) power consumption is dominated by the digital-to-analog converters (DACs), analog-to-digital converters (ADCs), and the FEC engines, which leaves a very small portion for the actual clock and data recovery (CDR) module. The chromatic dispersion compensation is a power power-hungry component of the CDR, but I suspect that it consumes 25% of the whole ASIC engine power. Can the authors either support their assumption or correct it to better reflect the practical implementation?

Response:

We thank the reviewer for the comment. In our manuscript and supplementary information, we assume the claim of “25% of the DSP power is dedicated to CD compensation” by referring to Ref. [Seiler PM, et al. *Toward coherent O-band data center interconnects. Frontiers of Optoelectronics* 14, 414-425 (2021)], in which they claim that “Here, we assume that the DSP accounts for approx. 50% of the total power consumption [41], and that the CD compensation accounts for approx. 25% of the DSP power consumption [40].” This reference has already been added to the Supplementary Information as Ref. 10.

We look further into Ref. 40 of the cited Ref. 10 in the Supplementary Information [*End-to-end energy modeling and analysis of long-haul coherent transmission systems. Journal of Lightwave Technology* 32, 3093–3111 (2014)]. In this literature, the authors note the CD compensation is implemented through a frequency domain filter, which involves a fast Fourier transform (FFT) followed by single-tap multiplication and inverse fast Fourier transform (IFFT). Fig. 15 of this literature shows the breakdown of the power consumptions of various modules of the receiver DSP under various modulation formats of long-haul transmission. The FEC implementation is independent of pre-FEC BER. We can see that the CD compensation consumes about 25% or even more than 25% of the total power of the DSP. Thus, we think the assumption that ~25% power of the DSP is consumed by the

CD compensation is reasonable.

Reviewer 2

>>The Authors have provided a good response to the points I raised in my previous review. I hope that they agree that their manuscript is much improved over their initial submission. I appreciate the work that has gone into their changes.

We thank the reviewer for the approval of our efforts spent on the revisions. Indeed, we agree that our manuscript is much improved after the revision.

I have only a few minor further points that I think need to be considered / acknowledged in the text.

Comment #2-1:

When quoting numbers for power consumption, it is really important to be clear that these numbers are extrapolated.

E.g. in the abstract, where you state that “our method features a remarkably low energy consumption of ~0.304 pJ/bit”

Comment #2-1a:

This number is extrapolated from an assumed EDFA wall-plug efficiency of 16%, if I have read your manuscript correctly. So, your method does not “feature” this power consumption, but instead you extrapolate this number. Similarly, you are extrapolating an energy consumption figure for the coherent systems you compare to.

You might instead reasonably state:

“ From measurements, we extrapolate an energy efficiency of 0.3pJ/bit for CD compensation of 40-km SSMF, which we calculate as being around 6 times less than that of commercially available 400G ZR coherent transceivers modules”.

Please be careful to write in caveats each time you mention energy efficiency numbers that you have calculated with some assumptions.

E.g., where you write “power consumption is...”, replace with “power consumption is estimated to be ...”

Response:

We thank the reviewer for the comment. We are sorry to be less careful when making these statements, and we appreciate that the reviewer points these out. We agree with the reviewer that the power consumption is an estimated one.

We have revised our manuscript as follows:

Revisions:

In the Abstract (from line 30 to line 32):

“From measurements, we extrapolate an energy consumption of ~0.3 pJ/bit for CD compensation of 40-km-SMFs, which we calculate as being around 6 times less than that of the commercially available 400G ZR coherent transceiver modules.”

At the end of the Introduction (from line 121 to line 123, line 125 to line 127):

“Importantly, we assessed the power consumption for the CD compensation to be as low as ~ 0.3 pJ/bit for 40-km SMFs. This remarkably low power consumption is estimated to be around six times less than that of the commercially available 400G ZR coherent transceiver modules.”

“As a result, the estimated power consumption can even be reduced to ~ 0.1 pJ/bit for the 40-km SMF transmission with an optimized coupling loss and insertion loss of our CDC.”

In the Discussion (from line 369 to line 370, line 379, line 386 to line 387, line 389 to line 392):

“As seen from Table 1, the normalized power consumption of CD compensation for our CDC is estimated to be ~ 0.3 pJ/bit, ...”

“Fig. 5a shows the estimated normalized power consumption of our CDC and ...”

“For a 120-km-SMF transmission, the power consumption of our CD compensation scheme is estimated to be less than ~ 0.8 pJ/bit.”

“Fig. 5b depicts the extrapolated power consumption with different total supported data rates from 400 Gbit/s to 12.8 Tbit/s over 40 km SMF transmission. Due to the parallel compensation, the estimated power consumption of our CDC is even lower for higher data rates.”

In the Conclusion (from line 436 to line 437, line 439 to line 440):

“The power consumption of the CD compensation of our CDC is estimated to be ~ 0.3 pJ/bit for 40-km-SMFs, ...”

“Particularly, when the optimized CL and IL are considered, the estimated power consumption of our CDC can be further reduced to ~ 0.1 pJ/bit, ...”

In Supplementary Note 7 (from line 431 to line 432, line 461 to line 462, line 469, line 470 to line 471, line 486 to line 487, line 495, line 514 to line 515, line 518, line 532 to line 533, line 535, line 562 to line 563):

“In this case, the estimated power consumption of the EDFA is ~ 152 mW.”

“Thus, the estimated power consumption in units of pJ/bit of these three modules are 5 pJ/bit, ~ 5.6 pJ/bit, and ~ 5.6 pJ/bit for 120-km-SMFs CDC, respectively.”

“... we can obtain that the estimated power consumption of CD compensation ...”

“The power consumption of CD compensation for 1.6T ZR coherent is estimated to be ~ 2.6 pJ/bit.”

“The normalized power consumptions are estimated to be ...”

“We can see that the estimated power consumption ...”

“In this case, the power consumption of our CDC is estimated to be ...”

“We can see that the estimated power ...”

“... the total power consumption of our CDC under the seven scenarios is estimated to be ...”

“The estimated power consumption of the coherent modules ...”

“The estimated power consumption of our CDC ...”

Comment #2-1b:

If you are suggesting that this measure is approximate (i.e. ~ 0.304), then you should not be quoting a number to 3 significant figures. Use ~ 0.3 pJ/bit instead ...

Please truncate energy consumption figures throughout the manuscript to this level of precision where you provide them, as your estimates are not accurate to within $<1\%$ of the nominal value ...

Response:

We thank the reviewer for the comment. We agree with the reviewer that the figures should be approximated to one decimal place. We have revised the energy consumption figures in our manuscript and supplementary information.

We do not tend to list the revisions of the figures here as there are many of them in the manuscript and the Supplementary Information. We have revised them all to make sure they are rounded to the nearest tenth. And we put a “ \sim ” before each figure. Each revised figure has been highlighted in red.

Comment #2-2:

Similarly, in the discussion, please place in a clear description of the key assumptions you have made in your energy efficiency calculation, in addition to pointing the reader toward your supplementary materials.

E.g., where you have:

“See Supplementary Note 7 ...power consumption.”

Please add in, such that this reads something like:

“See Supplementary Note 7 ...power consumption. The key assumptions we make are a wall-plug efficiency for EDFAs of around 16%, and that CDC consumes approximately 12.5% of the power used in a coherent transceiver.”

Response:

We thank the reviewer for the comment. We have added a description of the key assumptions in the Discussion.

Revision:

In the Discussion (from line 361 to line 363):

“The key assumptions we make are a $\sim 16\%$ wall-plug efficiency for EDFAs^{75, 76}, and that

CDC consumes approximately 12.5% of the power used in a coherent transceiver^{25, 77}.”

We have added some related literatures in References:

“75. Massicott JF, Wyatt R, Ainslie BJ, Crag-Ryan SP. Efficient, high power, high gain, Er³⁺-doped silica fibre amplifier. *Electronics letters* **26**, 1038-1039 (1990).

76. <https://semix.com/product/chip-laser-diode-8/>.

77. Seiler PM, et al. Toward coherent O-band data center interconnects. *Frontiers of Optoelectronics* **14**, 414-425 (2021).”

Comment #2-3:

The authors still do not provide a justification for the degree of reconfigurability they think is needed for DCI links. This I think is required so that the reader can understand the context of the work presented.

I imagine that the authors could be thinking about shipping transceiver units with this compensator included, that then need to be tuned to the correct dispersion over a broad range? They might be thinking that the DCI network is reconfigurable (e.g., through WSS/ROADM), and therefore dispersion compensation needs to be tunable?

Otherwise, it would seem to be a trivial exercise to add in a fixed dispersion compensator.

Response:

We thank the reviewer for the comment. We justified the needed reconfigurability of the DCI links in our revised manuscript by referring to the literature [Nagarajan R, et al. *Silicon photonics-based 100 Gbit/s, PAM4, DWDM data center interconnects. J. Opt. Commun. Netw., 10, B25-B36 (2018)*].

In the previously revised manuscript (from lines 57 to 60), we have the following statement:

“As the typical distance between two data centers is within 80 km and the majority (~83%) of the DCIs are within 40 km⁹, we mainly focus on the short-reach inter-DCIs with a fiber length of up to 40 km and the utilization of IM-DD scheme for this scenario due to the inherent advantages IM-DD offers, including cost-effectiveness, low power consumption, and a compact physical footprint.”

We believe the tunable dispersion compensation is suitable for both scenarios (i.e. I. shipping transceiver units with this tunable CDC tuned to the correct dispersion, and II. applications in reconfigurable DCI networks). Since ~83% of the DCIs are within 40 km, the tunable CDC can satisfy the need for CD compensation of the majority of the DCI links by using only one chip or module. In addition, as mentioned in Ref. 1, the reconfigurable optical networks may be utilized in next-generation systems to improve the system performance and efficiency by making the network demand aware. Therefore, the tunable CDC is also suitable for applications in the reconfigurable optical networks.

Besides, we note that the tunable CDC, rather than the fixed CDC, is desired for the optical communication system, as the amount of dispersion may vary in time due to several possible impairments such as the optical power variations and environmental-variation

induced transmission condition changes².

References:

1. Hall, M. N., Foerster, K. T., Schmid, S., & Durairajan, R. A survey of reconfigurable optical networks. *Optical Switching and Networking* **41**, 100621 (2021).
2. Eggleton, B. J., Ahuja, A., Westbrook, P. S., Rogers, J. A., Kuo, P., Nielsen, T. N., & Mikkelsen, B. Integrated tunable fiber gratings for dispersion management in high-bit rate systems. *Journal of Lightwave Technology* **18**, 1418-1432 (2000).

To point out the importance of the reconfigurability of CDCs, we have revised our manuscript as follows:

Revision:

In the Introduction (from line 78 to line 85):

“The tunability of CDCs is important, as the amount of CD may vary in time due to some possible impairments such as the optical power variations and environmental-variation-induced transmission condition changes²⁸. Besides, one reconfigurable CDC capable of compensating the CD of different lengths of SMFs up to 40 km can meet the demand of CD compensation for ~83% of DCI links, which can greatly reduce the overall costs. In addition, reconfigurable optical networks that may be deployed in next-generation DCI networks to improve the system performance and efficiency²⁹ also need tunable CDCs.”

Comment #2-4:

When comparing your result to the energy consumption of DCF, it should be acknowledged that DCF is colourless, and so does not require temperature control to operate (unlike wavelength selective approaches such as your own, or other channelized dispersion compensators).

Please be careful to include this caveat when comparing your work against DCF in terms of energy consumption.

Response:

We thank the reviewer for the comment. Indeed, the operation of DCFs is different from that of wavelength selective approaches. In our Supplementary Information, we include the power consumption of CD compensation using DCFs. However, we did not tend to make the comparison between using DCFs against using our approach, because DCFs are bulky and the dispersion compensation is not reconfigurable, which cannot satisfy the requirement of future DCI applications. We did not further calculate the normalized power consumption of CD compensation using DCFs, and we also did not include the DCFs in the comparison shown in Table 1 in the manuscript.

We have revised the statements of the CD compensation using DCFs in the Supplementary Note 7 as follows to make them clearer:

Revision:

In the (i) part of the Supplementary Note 7 (from lines 432 to 436):

“To be mentioned, the DCF can also be used for parallel CD compensation due to their broad optical bandwidth. And its colorless operation requires no temperature control. However, the DCF is bulky and can only compensate for the CD of a fixed length of the SMF. Therefore, we will not include the DCF in the power consumption comparison.”

Comment #2-5:

Can the authors label where the demonstration that they provide correlates with the curves in Fig 5, please?

Response:

We thank the reviewer for the comment. The key demonstrations that are correlated with Fig. 5 are shown in the second paragraph of the Discussion section in the manuscript. The detailed analysis of the data shown in Fig. 5 is put into the four paragraphs after Supplementary Table 4 in Supplementary Note 7.

To make the power consumption calculation clear in the manuscript, we have added some statements of the utilized key assumptions and methods into the Discussion:

Revision:

In the Discussion (from line 382 to line 384):

“In the calculation, 7 extra MRRs are needed for the CD compensation of an additional 40 km SMF, and every extra 40 km SMF transmission introduces a 10-dB extra loss.”

Comment #2-6:

Table 1 compares integrated CDC approaches only. The rest of the text focuses only on DCF as a non-integrated approach. I agree that keeping table 1 to integrated approaches is a good idea, but other tunable compensators should also be acknowledged in the text (even if only to then establish why your approach might have advantages).

I would encourage the authors to also compare against established channelized dispersion compensators, such as cascaded chirped fibre Bragg gratings (e.g. teraxion.com/en/products/optical-communications/tunable-dispersion-compensation-module-benchtop/) and others (I suspect that this is a Gires-Tournois interferometer: fiberroad.com/products/dcm/50ghz-100ghz-tunable-dispersion-compensator-tdcm/)

Here, I think it makes sense to then talk about how your approach is integrable, as opposed to suggesting that any comparison should only be made in terms of power consumption. In the case of DSP, clearly a head-to-head comparison was needed, as you are comparing chips with chips. For physically larger alternatives, I think that is your major point of difference.

Response:

We thank the reviewer for the comment. Indeed, we mainly include the integrated CDC approaches in Table 1. We agree with the reviewer that we should further include the

tunable dispersion compensation modules (TDCMs) in the comparison in terms of the footprint, rather than the power consumption, as it is not that trivial to get the normalized power consumption of CD compensation of TDCMs.

We have mentioned DCFs and TDCMs in the Discussion of the manuscript, and we mainly compare the performances of the integrated CD compensation methods in Table 1 of the manuscript.

We have revised our manuscript as follows:

Revisions:

In the Discussion (from line 350 to line 355, line 363):

“While several commercially available optical CDCs, including DCFs⁷² and tunable dispersion compensation modules (TDCMs)^{73, 74}, can be utilized for parallel CD compensation in WDM systems, DCFs can only compensate for the CD of a fixed length of SMF, and both DCFs and TDCMs are bulky and lack compactness. In contrast, our approach shows the merits of compact size and reconfigurability, which are suitable for inter-DCI applications.”

“Table 1 gives a comparison of the performance of various integrated CD compensation methods, ...”

The title of Table 1:

“Table 1. Performance comparison of various integrated CD compensation methods”

We have added one reference on DCF and two more references on TDCMs:

“72. Thorlabs. Dispersion compensating fibers,
<https://www.thorlabs.com/catalogpages/obsolete/2017/DCF38.pdf>.

73. TeraXion. Tunable dispersion compensation module,
<https://www.teraxion.com/en/products/optical-communications/tunable-dispersion-compensation-module-benchtop/>.

74. Fiberroad. Tunable dispersion compensation module,
<https://fiberroad.com/products/dcm/50ghz-100ghz-tunable-dispersion-compensator-tdcm/>.”

Reviewer 3

>>Thank you the authors for the detailed responses to my comments. My comments have been adequately addressed in the response letter and in the revised manuscript.

>>Remarks on code availability: N/A

We thank the reviewer for the approval of our revised manuscript.

Other revisions

We have updated Ref. 6 in the manuscript from the early access version to the latest one:

Revision:

“6. Berikaa E, *et al.* Next-Generation O-band Coherent Transmission for 1.6 Tbps 10 km Intra-Datacenter Interconnects. *Journal of Lightwave Technology* **42**, 1126-1135 (2024).”

REVIEWERS' COMMENTS

Reviewer #1 (Remarks to the Author):

The authors have addressed my comments, and I do recommend the rapid publication of the work in NC.

Reviewer #2 (Remarks to the Author):

I appreciate the work further work that the authors have done to address my additional comments from the re-review. I have no further comments or suggestions.

I am happy to recommend this manuscript for publication in Nature Communications.